# Prior-Informed Flow Matching for Graph Reconstruction

## Abstract

We introduce *Prior-Informed Flow Matching (PIFM)*, a conditional flow model for graph reconstruction. Reconstructing graphs from partial observations remains a key challenge; classical embedding methods often lack global consistency, while modern generative models struggle to incorporate structural priors. PIFM bridges this gap by integrating embedding-based priors with continuous-time flow matching. Motivated by a permutation equivariant version of the distortion-perception theory, our method first uses a prior, such as GraphSAGE or node2vec, to form an informed initial estimate of the adjacency matrix based on local information. It then applies rectified flow matching to refine this estimate, transporting it toward the true distribution of clean graphs and learning a global coupling. Experiments on different datasets demonstrate that PIFM consistently enhances classical embeddings, outperforming them and state-of-the-art generative baselines in reconstruction accuracy.

## 1 Introduction

Graph generative models have seen remarkable progress in recent years, enabling the synthesis of realistic graph structures in domains such as drug design (Yang et al., 2024) and social networks (Grover et al., 2019). In particular, diffusion-based (Niu et al., 2020; Jo et al., 2022; Vignac et al., 2023) and flow-based (Qin et al., 2025; Eijkelboom et al., 2024) approaches have emerged as state-of-the-art. While these models excel at *unconditional* generation and property-controlled generation, their application to inverse problems, and in particular, the reconstruction of a graph from partial observations, remains a fundamental open problem.

Conditional graph reconstruction aims to recover the topology of a graph from a partially observed set of edges while ensuring that the reconstructed graph aligns with the global distribution of valid graphs. The most common formulation of this problem is link prediction, which typically assumes that the locations of the missing edges are known. Although modern link-prediction methods can incorporate global context, such as edge features, they typically return edge-wise predictions. Classical embedding techniques, such as Node2Vec (Grover & Leskovec, 2016; Perozzi et al., 2014), and inductive architectures, like GraphSAGE (Hamilton et al., 2017), learn expressive local representations for this task but do not explicitly capture dependencies between edge predictions or preserve global structural properties.

More recently, diffusion and flow-based generative models have been adapted for graph inpainting and posterior-guided sampling (Vignac et al., 2023; Trivedi et al., 2024; Sharma et al., 2024; Tenorio et al., 2025). While these approaches leverage powerful graph priors to generate globally consistent samples, they are primarily designed for constrained generation rather than fidelity and observation-driven reconstruction. Consequently, a gap remains: link-prediction methods generates edge-wise, independent scores without an explicit joint output model, and generative approaches produce structurally coherent samples but may lack reconstruction fidelity.

We introduce **Prior-Informed Flow Matching (PIFM)**, a flow matching-based model that interpolates between a structural edge predictor as an informative source and learns a coupled correction toward the clean-graph distribution. The distortion–perception perspective (Blau & Michaeli, 2018; Freirich et al., 2021; Ohayon et al., 2025) motivates a two stage design: estimate marginal edge probabilities, then refine them with a rectified flow (Liu et al., 2023; Albergo et al., 2023). The first stage can use diverse structural estimators, while the second stage learns dependencies across their edge predictions. The resulting architecture is

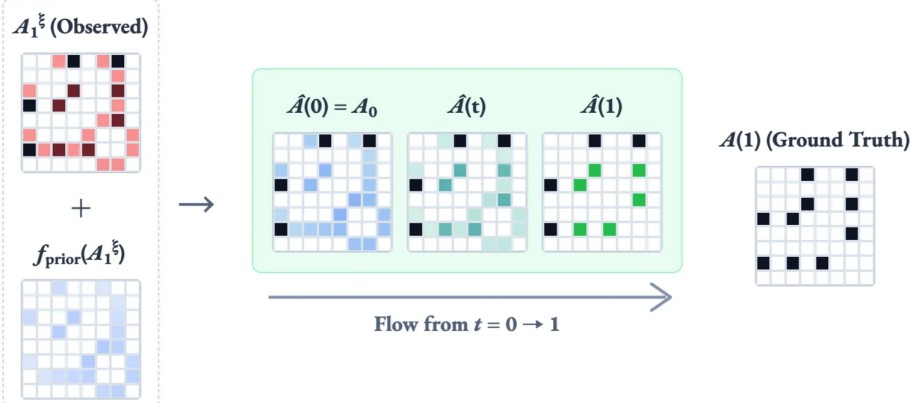

Figure 1: Overview of the **Prior-Informed Flow Matching (PIFM)** graph reconstruction framework. Starting from a partially observed adjacency matrix $\mathbf{A}^{\mathcal{O}} = \xi \odot \mathbf{A}$, where $\xi$ denotes a mask, we form an initialization $\mathbf{A}_0$ by combining the observed entries with prior predictions $f_{\text{prior}}(\mathbf{A}^{\mathcal{O}})$ obtained with an element-wise predictor. In dark red we denote the true edges that are masked, while in light red those masked positions where there is no edge between nodes. A rectified flow then interpolates linearly from $\mathbf{A}_0$ to the ground-truth graph $\mathbf{A}_1 = \mathbf{A}$, learning global structural information from a coupling of all the edges. The intermediate states $\mathbf{A}_t$ improve on the prior-informed initialization, enabling recovery of the missing edges.

permutation equivariant. We evaluate PIFM on dense and sparse graph datasets using embedding, GNN, autoencoder, and Transformer priors. PIFM adapts standard rectified flow matching (Liu et al., 2023; Albergo et al., 2023) to graph reconstruction; what is graph-specific is the observation-dependent source, the within-instance coupling, and the equivariance guarantee of Theorem 1.

Our contributions are as follows:

- We formulate conditional graph reconstruction through the lens of the permutation-equivariant distortion–perception trade-off.

- We propose PIFM, a two-stage reconstruction framework that first constructs a prior-informed source distribution using embeddings from latent graph models, and then refines these predictions with a flow-matching model.

- We show that existing link prediction methods primarily rely on local edge-wise predictions and lack mechanisms to enforce global structural consistency. PIFM addresses this limitation by learning a generative model that integrates information across the full graph during reconstruction.

- We empirically validate our approach on link prediction and two blind reconstruction settings, termed *expansion* (recovering missing edges) and *denoising* (removing spurious edges), showing consistent improvements over predictors based solely on local information.

## 2 Related Work

**Flow/Diffusion models on graphs.** Diffusion and flow-based graph generative models have shown impressive performance in recent years. Early models, namely EDP-GNN (Niu et al., 2020) and GDSS (Jo et al., 2022), employ score-based *continuous* diffusion over a relaxation of the graph structure. However, given that graphs are inherently discrete, subsequent work has explored discrete diffusion processes (Austin et al., 2021). Models like DiGress (Vignac et al., 2023) demonstrated the effectiveness of this approach, which has been further advanced by discrete flow-based models like DeFoG (Qin et al., 2025) and variational approaches like CatFlow (Eijkelboom et al., 2024). A common point of these models is their reliance on a simple source distribution, such as Gaussian (continuous) or uniform (discrete) noise. While effective for unconstrained generation, recent work on image-based inverse problems demonstrates the advantages of learning a data-dependent flow, using a prior-informed source distribution (Albergo et al., 2023; Delbracio

& Milanfar, 2024; Ohayon et al., 2025). Concurrently to our work, Flowette (Wijesinghe et al., 2026) uses domain-specific graphette priors and flow matching for graph generation. In contrast, PIFM is conditioned on a partially observed graph and mask, pairing each prior-informed source with the corresponding clean graph for reconstruction. While both methods exploit informative structural priors, they address different tasks and differ in conditioning, coupling, architecture, and evaluation, making them complementary rather than competing approaches.

**Graph topology inference via flow/diffusion-based solvers.** Graph topology inference – the task of recovering hidden edges from a partially observed graph – is a long-standing inverse problem (Segarra et al., 2017; Dong et al., 2016). Several methods adapt diffusion for constrained graph generation, which is related to but distinct from topology inference. DiGress (Vignac et al., 2023) introduced an inpainting mechanism, inspired by Repaint (Lugmayr et al., 2022), to generate graph structures consistent with a partial observation. Similarly, in Trivedi et al. (2024) a similar mechanism is used for completing partially observed graphs. PRODIGY (Sharma et al., 2024) enforces hard constraints by projecting the graph estimate onto a feasible set at each sampling step. More recently, GGDiff (Tenorio et al., 2025) incorporates a guidance mechanism as a flexible alternative to inpainting. However, all these methods are designed for *constrained generation* (e.g., molecule generation with a given scaffold) rather than *recovering masked edges from a partially observed graph.* Hence, to the best of our knowledge, designing a diffusion-based model explicitly for graph topology inference remains an open problem.

**Link prediction and Graph Autoencoders.** Graph reconstruction from partial observations is closely related to link prediction, which aims to infer whether unobserved edges should exist in a partially observed graph (Newman, 2001; Adamic & Adar, 2003; Zhou et al., 2009). Classical approaches rely on topology based heuristics, while embedding based methods such as DeepWalk (Perozzi et al., 2014) and node2vec (Grover & Leskovec, 2016) learn node representations from random walks and score candidate edges from pairwise embedding features. Graph neural network (GNN) methods further improve link prediction by learning node representations through message passing. For example, GraphSAGE learns inductive neighborhood aggregation functions that can generalize across nodes or graphs (Hamilton et al., 2017), while variational graph autoencoders (VGAE) encode the observed graph into latent node embeddings and decode pairwise edge probabilities (Kipf & Welling, 2016). Other methods model pairwise or path based structure more directly, such as NBFNet, which formulates link prediction as neural path aggregation through a Bellman Ford style recurrence (Zhu et al., 2021). LPFormer (Shomer et al., 2024) combines structural encodings with attention for link prediction.

## 3 Background

We represent an undirected graph $\mathcal{G} = \{\mathcal{V}, \mathcal{E}\}$, where $\mathcal{V}$ denotes the nodes and $\mathcal{E}$ the edges, by its binary symmetric adjacency matrix $\mathbf{A} \in \mathbb{R}^{N \times N}$. Table 1 summarizes the main notation used throughout the method.

Table 1: Notation used in PIFM.

| Symbol | Meaning |
|--------|---------|
| $\mathbf{A}_1$ | Clean ground-truth adjacency matrix |
| $\mathbf{A}^{\mathcal{O}}$ | Observed partial adjacency matrix |
| $\xi$ | Binary observation mask |
| $\hat{\mathbf{A}}^*$ | Prior estimate or approximate MMSE estimate |
| $\mathbf{A}_0$ | Prior-informed source sample for flow matching |
| $\mathbf{A}_t$ | Intermediate state along the flow path |
| $\widehat{\mathbf{A}}$ | Final reconstructed adjacency matrix |

### 3.1 Continuous flow matching for graph generation

Flow matching (Albergo et al., 2023; Lipman et al., 2023) is a family of generative models that defines a continuous-time transport map from samples $\mathbf{A}_0$ drawn from a source distribution $p_0$ to samples $\mathbf{A}_1$ from a target distribution $p_1$. It is governed by the ODE

$$d\mathbf{A}_t = v(\mathbf{A}_t, t)\, dt, \tag{1}$$

where $v(\cdot, t)$ is a velocity field and $\mathbf{A}_t$ denotes a forward process, also known as stochastic interpolant, for $t \in [0, 1]$. Typically, $p_0$ is a tractable distribution (e.g., a Gaussian distribution), while $p_1$ corresponds to the data distribution. To generate new samples, one must specify both $\mathbf{A}_t$ and $v$. A common choice for the forward process is $\mathbf{A}_t = \alpha_t \mathbf{A}_0 + \beta_t \mathbf{A}_1$, where $\alpha_t$ and $\beta_t$ are differentiable functions such that $\alpha_0 = 1$, $\beta_0 = 0$ and $\alpha_1 = 0$, $\beta_1 = 1$. Differentiating this path gives a velocity $v(\mathbf{A}_t, t) = \dot{\alpha}_t \mathbf{A}_0 + \dot{\beta}_t \mathbf{A}_1$. Despite its closed-form, this expression depends explicitly on $\mathbf{A}_1$, making it impractical since the target is unknown at inference/sampling. To circumvent this, we instead consider $v(\mathbf{A}_t, t) = \mathbb{E}_{\mathbf{A}_0, \mathbf{A}_1}[\dot{\alpha}_t \mathbf{A}_0 + \dot{\beta}_t \mathbf{A}_1 \mid \mathbf{A}_t]$, the conditional expectation of the velocity given $\mathbf{A}_t$ (Albergo et al., 2023), which is then approximated with a neural network $v_\theta$. The network is trained using a mean squared error loss:

$$\mathcal{L}_{\mathrm{FM}} = \mathbb{E}_{t, \mathbf{A}_0, \mathbf{A}_1} \left[ \left\| v_\theta(\mathbf{A}_t, t) - (\dot{\alpha}_t \mathbf{A}_0 + \dot{\beta}_t \mathbf{A}_1) \right\|_2^2 \right]. \tag{2}$$

In particular, this formulation does not require $\mathbf{A}_0$ and $\mathbf{A}_1$ to be independent; in fact, they might be sampled from a joint distribution, allowing for richer transport plans in cases where paired data is available. This has been exploited to solve inverse problems on images (Ohayon et al., 2025; Albergo et al., 2024; Delbracio & Milanfar, 2024), and is directly related to our proposed method, as described later.

Throughout this work, we consider the *rectified flow* case (Liu et al., 2023), where $\alpha_t = 1 - t$ and $\beta_t = t$. As shown in Tong et al. (2024), the velocity field associated with this linear path approximates the optimal transport vector field when the joint distribution $p(\mathbf{A}_0, \mathbf{A}_1)$ closely resembles the optimal coupling between the marginals $p(\mathbf{A}_0)$ and $p(\mathbf{A}_1)$. We defer to Appendix B.5.1 a more detailed background on generative models on graphs beyond continuous flow matching, including diffusion-based models, as well as related works.

### 3.2 Distortion–perception formulation for conditional graph reconstruction

Our goal is to reconstruct the ground-truth adjacency matrix $\mathbf{A}$ of a graph $\mathcal{G}$ from a partially observed version, denoted by $\mathbf{A}^{\mathcal{O}}$. Similarly to image restoration, there are two complementary criteria to assess the quality of reconstructed graphs: ($i$) the average closeness of the reconstruction to the ground truth, measured through a distortion metric, and ($ii$) the similarity in distribution between reconstructed and real graphs, namely, how well the reconstructed graph resembles a sample from the underlying graph distribution.

These two objectives are generally at odds and can be formalized through the perception–distortion trade-off introduced in Blau & Michaeli (2018):

$$D(P) = \min_{p(\hat{\mathbf{A}} | \mathbf{A}^{\mathcal{O}})} \left\{ \mathbb{E}_{p(\mathbf{A}, \hat{\mathbf{A}})} \left[ \Delta(\mathbf{A}, \hat{\mathbf{A}}) \right] \; : \; d(p_{\mathbf{A}}, p_{\hat{\mathbf{A}}}) \leq P \right\}, \tag{3}$$

where $\hat{\mathbf{A}}$ denotes an estimator of $\mathbf{A}$ conditioned on the observation $\mathbf{A}^{\mathcal{O}}$, $\Delta(\cdot, \cdot)$ is a distortion metric (e.g., MSE), and $d(p_{\mathbf{A}}, p_{\hat{\mathbf{A}}})$ measures the divergence between the distribution of clean graphs $p_{\mathbf{A}}$ and the distribution of reconstructed graphs $p_{\hat{\mathbf{A}}}$.

The function in (3) has been extensively studied in the image domain, where different values of $P$ yield estimators with different trade-offs between reconstruction fidelity (distortion) and perceptual realism. We adapt this framework to graph reconstruction by estimating the adjacency matrix $\mathbf{A}$ while accounting for the symmetry constraints induced by the permutation invariance of graph representations.

**MSE distortion.** We first consider the distortion metric $\Delta(\mathbf{A}, \hat{\mathbf{A}}) = \|\mathbf{A} - \hat{\mathbf{A}}\|_F^2$, corresponding to the mean squared error (MSE). This setting was studied in Freirich et al. (2021), particularly for the two extreme

cases $P = \infty$ and $P = 0$. When $P = \infty$, the problem reduces to minimizing distortion without imposing perceptual constraints. The optimal estimator is the posterior mean

$$\hat{\mathbf{A}}^* = \mathbb{E}[\mathbf{A} \mid \mathbf{A}^{\mathcal{O}}],$$

which minimizes the expected MSE. Although optimal in terms of distortion, this estimator may produce unrealistic graphs whose structural properties deviate from those of the underlying graph distribution.

At the other extreme, $P = 0$ enforces perfect perceptual consistency, meaning that the reconstructed graphs follow exactly the same distribution as the ground-truth graphs, i.e., $p_{\hat{\mathbf{A}}} = p_{\mathbf{A}}$. As shown in Freirich et al. (2021), the corresponding estimator can be characterized through the optimal transport problem

$$p^*_{\hat{\mathbf{A}}, \hat{\mathbf{A}}^*} = \underset{p \in \Pi(p_{\mathbf{A}}, p_{\hat{\mathbf{A}}^*})}{\operatorname{argmin}} \mathbb{E}\left[\|\hat{\mathbf{A}} - \hat{\mathbf{A}}^*\|_F^2\right], \tag{4}$$

where $\Pi(p_{\mathbf{A}}, p_{\hat{\mathbf{A}}^*})$ denotes the set of couplings with marginals $p_{\mathbf{A}}$ and $p_{\hat{\mathbf{A}}^*}$.

For the special case of squared-error distortion with the Wasserstein-2 perception index, Freirich et al. (2021) show that the estimator at any point on the distortion–perception curve can be obtained by interpolating the two extremes, $\hat{\mathbf{A}}_p = (1 - \frac{P}{P_\infty})\hat{\mathbf{A}} + \frac{P}{P_\infty}\hat{\mathbf{A}}^*$, where $\hat{\mathbf{A}}$ is the perfect-perception ($P = 0$) estimator, $\hat{\mathbf{A}}^*$ is the MMSE estimator, and $P_\infty$ is the perception index attained by $\hat{\mathbf{A}}^*$. Therefore, obtaining the estimator associated with $D(0)$ amounts to solving an optimal transport problem between the distribution of clean graphs $p_{\mathbf{A}}$ and the distribution of the MMSE estimator $p_{\hat{\mathbf{A}}^*}$. In practice, we approximate a $P = 0$ estimator by *(i)* using an element-wise structural predictor $\hat{\mathbf{A}}^*$ as a approximation for the posterior mean $\mathbb{E}[\mathbf{A} \mid \mathbf{A}^{\mathcal{O}}]$, and *(ii)* learning a transport from $p_{\hat{\mathbf{A}}^*}$ to $p_{\mathbf{A}}$ with the rectified flow described below. We do not establish that these predictors recover $\mathbb{E}[\mathbf{A} \mid \mathbf{A}^{\mathcal{O}}]$, so the posterior-mean view motivates the choice of source rather than characterizing it. The learned map is trained to push $p_{\hat{\mathbf{A}}^*}$ toward $p_{\mathbf{A}}$, i.e., to approach perfect perception ($P = 0$); this is only approximate, both because of finite model capacity and because a continuous, invertible flow cannot exactly match the discrete distribution of binary adjacency matrices. We likewise do not claim it recovers the exact optimal-transport coupling of (4). Intuitively, this procedure refines the coarse estimate $\hat{\mathbf{A}}^*$ into a graph that is consistent with the underlying data distribution.

We focus on the $P = 0$ case: while $P = \infty$ is MSE-optimal, it may produce graphs that violate structural constraints (e.g., chemical validity in molecule generation), whereas $P = 0$ guarantees distributional consistency.

**Cross-entropy distortion.** We also consider a cross-entropy-based distortion, which is particularly natural for graph reconstruction since adjacency matrices are discrete-valued objects. Specifically, letting $p_\theta^d(k \mid \mathbf{A}_t, t)$ denote the categorical distribution induced by the flow model for edge variable $d$, we define

$$\Delta(\mathbf{A}, \hat{\mathbf{A}}) = -\sum_{d=1}^{D}\sum_{k=1}^{K^d} \mathbb{I}[A^d = k] \log p_\theta^d(k \mid \mathbf{A}_t, t),$$

which corresponds to the cross-entropy between the ground-truth adjacency matrix and the predicted edge distributions, and where $D = \frac{N(N-1)}{2}$ denotes the number of edge variables. The expression above is the generic form. In the binary instantiation we actually optimize (Eq. (8)), the sum is restricted to the supervised node pairs rather than taken over all $D$ of them, and a positive-class weight is applied to counteract edge sparsity. Unlike the MSE case, this formulation does not admit the same closed-form characterization of the perception–distortion trade-off. Nevertheless, as we demonstrate experimentally, it exhibits qualitatively similar behavior, yielding a comparable trade-off between reconstruction fidelity and perceptual consistency.

## 4 Method

PIFM can be viewed as a two-stage refinement framework for conditional graph reconstruction, motivated by the distortion–perception formulation introduced in Section 3.2. First, a structural prior estimates the

conditional mean $\mathbb{E}[\mathbf{A} \mid \mathbf{A}^{\mathcal{O}}]$ from the observed subgraph $\mathbf{A}^{\mathcal{O}}$ by predicting marginal probabilities for the masked node pairs. While this estimate minimizes distortion locally, it does not necessarily produce a globally consistent graph. To address this, PIFM uses the resulting adjacency estimate as the source state of a rectified flow that learns to refine it toward the true graph distribution, recovering higher-order structural dependencies not captured by edge-wise predictions alone. The flow is trained on paired samples $(\mathbf{A}_0, \mathbf{A}_1)$ from the same graph, where $\mathbf{A}_0$ combines observed adjacency entries with prior predictions on the masked region. At inference time, only $\mathbf{A}^{\mathcal{O}}$ and $\xi$ are available: the prior produces the initial estimate, and the flow integrates the learned velocity field to generate the final graph reconstruction.

### 4.1 Approximating the posterior mean

As discussed in Section 3.2, our goal is to approximate the conditional mean $\mathbb{E}[\mathbf{A} \mid \mathbf{A}^{\mathcal{O}}]$ with a *permutation equivariant* estimator. Before moving to particular parameterizations of the conditional mean, we introduce two assumptions.

**AS 1.** *We assume each edge in $\mathbf{A} \in \{0,1\}^{n \times n}$ follows a Bernoulli distribution whose probabilities depend on latent node variables $\mathbf{z}_1, \ldots, \mathbf{z}_n \in \mathcal{Z}$ such that:*

$$A_{ij} \sim Bernoulli\big(f(\mathbf{z}_i, \mathbf{z}_j)\big), \qquad 1 \le i < j \le n. \tag{5}$$

*The function $f$ maps pairs of latent variables to edge probabilities, i.e., $f(\mathbf{z}_i, \mathbf{z}_j) = P(A_{ij}|\mathbf{z}_i, \mathbf{z}_j) = p_{ij}$.*

**AS 2.** *We assume that the edges are conditionally independent given the latent structure, i.e., given the latent structure $Z = \{\mathbf{z}_1, \ldots, \mathbf{z}_n\}$, we have:*

$$P(\mathbf{A} \mid Z) = \prod_{1 \le i < j \le n} P(A_{ij} \mid \mathbf{z}_i, \mathbf{z}_j). \tag{6}$$

Under these two assumptions, and assuming access to the mapping $\mathbf{z}^{-1} : \mathbf{A} \to \mathcal{Z}$ that recovers the latent coordinates $Z = \mathbf{z}^{-1}(\mathbf{A}^{\mathcal{O}})$, the posterior mean can be computed element-wise: on observed entries it equals the known value $A_{ij}^{\mathcal{O}}$, while on masked entries $\left[\mathbb{E}[\mathbf{A} \mid \mathbf{A}^{\mathcal{O}}]\right]_{ij} = P(A_{ij} = 1 \mid \mathbf{z}_i, \mathbf{z}_j) = f(\mathbf{z}_i, \mathbf{z}_j)$.

We adopt two different types of priors: ($i$) inductive methods, represented by *GraphSAGE* (Hamilton et al., 2017), a GNN-based estimator, and ($ii$) transductive ones, obtained from *node2vec* (Grover & Leskovec, 2016), which provides an instance-level learned probabilistic model.

**Posterior mean using inductive methods (dataset-informed).** We approximate the posterior mean using a dataset-informed, inductive approach based on *GraphSAGE* (Hamilton et al., 2017). We train the model on the partially observed graphs in the dataset to produce node embeddings $\{\mathbf{z}_i\}_{i=1}^N$. From these embeddings, we train a single logistic predictor on Hadamard edge features $(\mathbf{z}_i \odot \mathbf{z}_j)$ to estimate edge probabilities. The resulting conditional mean is parameterized as $\left[\mathbb{E}[\mathbf{A} \mid \mathbf{A}^{\mathcal{O}}]\right]_{ij} \approx f_\phi(\mathbf{z}_i \odot \mathbf{z}_j)$.

**Posterior mean using transductive methods (instance-specific).** We use four per-graph priors that fit an embedding and predictor to each partially observed graph $\mathbf{A}^{\mathcal{O}}$. *node2vec* (Grover & Leskovec, 2016) learns embeddings $\{\mathbf{z}_i\}_{i=1}^N$ from biased random walks; a per-graph logistic predictor on Hadamard edge features then gives $\left[\mathbb{E}[\mathbf{A} \mid \mathbf{A}^{\mathcal{O}}]\right]_{ij} \approx f_\phi(\mathbf{z}_i \odot \mathbf{z}_j)$. *VGAE* (Kipf & Welling, 2016) uses a GCN encoder to produce a Gaussian posterior over per-node latents $q(\mathbf{z}_i \mid \mathbf{A}^{\mathcal{O}}) = \mathcal{N}(\boldsymbol{\mu}_i, \mathrm{diag}(\boldsymbol{\sigma}_i^2))$ and an inner-product decoder, yielding $\left[\mathbb{E}[\mathbf{A} \mid \mathbf{A}^{\mathcal{O}}]\right]_{ij} \approx \sigma(\boldsymbol{\mu}_i^\top \boldsymbol{\mu}_j)$, trained by ELBO maximization with a standard-normal prior. *NCNC* (Wang et al., 2024) is a neural common-neighbor link predictor that combines a GCN encoder with a pooled sum over softly-completed common-neighbor representations, giving $\left[\mathbb{E}[\mathbf{A} \mid \mathbf{A}^{\mathcal{O}}]\right]_{ij} \approx \sigma(\mathrm{NCNC}(i, j; \mathbf{A}^{\mathcal{O}}, \mathbf{X}))$. *LPFormer* (Shomer et al., 2024) combines a GCN node encoder with pair features, PPR-conditioned attention, and structural counts. We train a fresh model per graph using observed edges and observed non-edges, then score every masked pair.

## 4.2 Learning the flow model

We now learn a flow model that approximates the joint distribution (coupling) $p(\mathbf{A}, \hat{\mathbf{A}}^*)$ of the clean graph and its MMSE estimate. As explained in Section 3, we need to specify the forward path $\mathbf{A}_t$ and the velocity field $v$. Inspired by Ohayon et al. (2025), we incorporate prior information as the initialization of the forward path; with slight abuse of notation, we denote $f_{\text{prior}}$ as the prediction of the full graph (i.e., $f_{\text{prior}}(\mathbf{A}^{\mathcal{O}}) \approx \mathbb{E}[\mathbf{A} \mid \mathbf{A}^{\mathcal{O}}]$). Specifically, we compute the sample $\mathbf{A}_0$ from the source distribution as follows:

$$\mathbf{A}_0 = \mathbf{A}^{\mathcal{O}} + (1 - \xi) \odot \left( f_{\text{prior}}(\mathbf{A}^{\mathcal{O}}) + \boldsymbol{\epsilon}_s \right), \tag{7}$$

where $\mathbf{A}$ is the ground-truth graph, $\xi$ is the corresponding mask (taking value 1 for the observed pairs of nodes and 0 otherwise), $\mathbf{A}^{\mathcal{O}} = \xi \odot \mathbf{A}$ is the observed graph, and $f_{\text{prior}}(\mathbf{A}^{\mathcal{O}})$ is our approximate MMSE estimator for the masked node pairs. We also add a small amount of noise $\boldsymbol{\epsilon}_s \sim \mathcal{N}(0, \sigma_s^2)$ following Albergo et al. (2024). We define $\mathbf{A}_1 = \mathbf{A}$ for the target distribution.

**Distortion instantiations.** Given the source distribution, we can learn the flow by minimizing a certain distortion $\Delta(\hat{\mathbf{A}}_\theta, \mathbf{A})$. We consider two instances following the description in Section 3.2, which differ in what the network predicts, $\hat{y}_\theta$, and in the target $y$ it is regressed against (the two quantities minimized in Alg. 1).

($i$) **MSE.** The network is the velocity field: $\hat{y}_\theta = v_\theta(\mathbf{A}_t, t)$, $y = \mathbf{A}_1 - \mathbf{A}_0$, and $\Delta(\hat{y}_\theta, y) = \|\hat{y}_\theta - y\|_F^2$, recovering the rectified flow-matching loss in (2).

($ii$) **CE.** The network is instead an *endpoint predictor* $g_\theta(\mathbf{A}_t, t) \in \mathbb{R}^{N \times N}$ emitting per-pair logits for the clean adjacency, so that $\hat{y}_\theta = \sigma(g_\theta(\mathbf{A}_t, t))$ with $\sigma$ the logistic sigmoid, and the target is the endpoint $y = \mathbf{A}_1$ rather than the residual (Eijkelboom et al., 2024; Dieleman et al., 2022; Lee et al., 2026; Roos et al., 2026). The distortion is a class-weighted binary cross-entropy over the supervised pairs $\mathcal{M} = \{(i, j) : i < j, \xi_{ij} = 0\}$:

$$\Delta(\hat{y}_\theta, \mathbf{A}_1) = -\frac{1}{|\mathcal{M}|} \sum_{(i,j) \in \mathcal{M}} \left[ w\, A_{1,ij} \log \sigma(g_{\theta,ij}) + (1 - A_{1,ij}) \log\left(1 - \sigma(g_{\theta,ij})\right) \right], \qquad w = \min\left( \frac{n_-}{n_+}, 50 \right), \tag{8}$$

averaged per graph and then over the batch, where $n_+$ and $n_-$ count positives and negatives among the supervised pairs. The weight $w$ prevents the minimizer on sparse graphs from collapsing to predicting no edge. For the CE experiments we set $\sigma_s = 0$.

**Training.** For the velocity field $v$, we use the architecture from Jo et al. (2022), a GNN-based network that yields a permutation-equivariant parameterization (see Appendix D for details). For each training graph $\mathbf{A}_1$, we sample a mask $\xi$ and time $t \sim U[0, 1]$, construct $\mathbf{A}_0$ via Eq. (7), and minimize the flow-matching objective with the chosen distortion $\Delta$ over the resulting paired samples $(\mathbf{A}_0, \mathbf{A}_1)$. Both instantiations add explicit training weight at $t = 0$, the only point the $K = 1$ sampler evaluates: an endpoint anchor for MSE (Eq. (14), $\lambda_0 = 5$) and a boundary-aware time law for CE. Both act only during training, leaving the sampler of Alg. 1 unchanged; Appendix D.4 defines them, records which configuration produced each result, and ablates them.

**Inference.** Given an observed graph $\mathbf{A}^{\mathcal{O}}$ and mask $\xi$, we initialize $\hat{\mathbf{A}}$ from the prior prediction and integrate the learned velocity field using $K$ Euler steps, as shown in the sampling procedure of Alg. 1. For a fixed learned field and terminal time, $K$ determines the discretization of the Euler integration. Different finite-step choices therefore produce different estimators. Although $K$ is not theoretically a distortion–perception control parameter, it has been observed empirically to correlate with this trade-off Delbracio & Milanfar (2024); Ohayon et al. (2025): smaller values of $K$ produce estimates closer to the posterior mean, whereas larger values of $K$ behave more like posterior sampling. Under the MSE objective, the network directly predicts the velocity field. Under the CE objective, it predicts edge-value distributions, from which the velocity field is computed before each update:

$$v_\theta(\mathbf{A}_t, t) = \frac{\sigma\left(g_\theta(\mathbf{A}_t, t)\right) - \mathbf{A}_t}{1 - t + \varepsilon}, \qquad \varepsilon = 10^{-5}, \tag{9}$$

which is the velocity that carries the current state $\mathbf{A}_t$ to the predicted endpoint over the remaining time $1 - t$; the constant $\varepsilon$ keeps the denominator bounded as $t \to 1$.

**Permutation invariance.** Combined with the posterior-mean parameterization of Section 4.1, the permutation-equivariant velocity field endows PIFM with two related symmetry guarantees. First, the reconstruction map is permutation-equivariant: relabeling the observation and mask $(\mathbf{A}^{\mathcal{O}}, \xi)$, with $\mathbf{A}^{\mathcal{O}} = \xi \odot \mathbf{A}$ the observed part of the graph, produces a correspondingly relabeled reconstruction $\tilde{\mathbf{A}}_1$ (the sampler output at $t = 1$). This holds for the full sampler of Alg. 1 (both the MSE and CE branches, for any number of steps $K$) and needs only equivariance and well-posedness of the dynamics; it is formalized in Theorem 1. Second, in the idealized continuous-time flow the conditional density of the reconstruction is itself invariant under simultaneous relabeling of $(\tilde{\mathbf{A}}_1, \mathbf{A}^{\mathcal{O}}, \xi)$; this stronger statement is formalized in Proposition 1. Since graphs are exchangeable, the model's output should not depend on node ordering, making these symmetries a desirable inductive bias.

**Theorem 1** (Permutation equivariance of the reconstruction). *Assume:*

(a) *the prior $f_{prior} : \mathbb{R}^{N \times N} \to \mathbb{R}^{N \times N}$ and the velocity field $v_\theta : \mathbb{R}^{N \times N} \times [0, 1] \to \mathbb{R}^{N \times N}$ are permutation-equivariant;*

(b) *$v_\theta$ is locally Lipschitz in its first argument (e.g., a smooth neural network on a bounded domain), so the sampling dynamics are well-posed.*

*For an observed graph $\mathbf{A}^{\mathcal{O}}$ supported on the observed entries (i.e., $(1 - \xi) \odot \mathbf{A}^{\mathcal{O}} = 0$) and mask $\xi$, let $\mathrm{PIFM}(\mathbf{A}^{\mathcal{O}}, \xi)$ be the reconstruction returned by Alg. 1 under either the MSE or the CE branch, for any number of Euler steps $K \geq 1$. Then, for every permutation matrix $\mathbf{P}$, the reconstruction is permutation-equivariant in distribution,*

$$\mathrm{PIFM}(\mathbf{P}^\top \mathbf{A}^{\mathcal{O}} \mathbf{P}, \mathbf{P}^\top \xi \mathbf{P}) \stackrel{d}{=} \mathbf{P}^\top \mathrm{PIFM}(\mathbf{A}^{\mathcal{O}}, \xi) \mathbf{P}, \tag{10}$$

*with equality holding pointwise either under the coupled noise draw $\mathbf{P}^\top \boldsymbol{\epsilon}_s \mathbf{P}$ or when $\sigma_s = 0$ (as in our CE experiments).*

Theorem 1 establishes the symmetry property of the practical PIFM sampler. It applies to the algorithm exactly as implemented, for both MSE and CE objectives (in fact, is independent of the distortion) and any number of Euler steps $K$, including masked Euler updates, sigmoid activation (when applicable), and re-imposition of observed entries.

Theorem 1 provides a closure guarantee for the overall reconstruction map, assuming an equivariant prior and velocity field. It complements, rather than replaces, standard equivariance results for the individual GNN components. The guarantee is exact for GraphSAGE, VGAE, and NCNC (up to coupled randomness in stochastic inference). For Node2vec, the guarantee is only approximate, since the learned embeddings depend on stochastic random walks and optimization and therefore do not define a deterministic equivariant map. A detailed discussion is provided in Appendix C.1.

The proof is in Appendix C.1. Since the reconstruction task is exchangeable in node ordering (relabeling the input observation and mask should produce a correspondingly relabeled output), Theorem 1 guarantees that PIFM has this inductive bias built in.

**Final algorithm.** In Alg. 1, we describe our training and sampling algorithms. In essence, PIFM is a general framework that learns a global graph structure to enhance simple, conditionally independent edge-wise priors.

To illustrate what we mean by *learning a global and dependent predictor*, we now describe a toy experiment. Consider a four-node graph $\mathcal{G}$ (see Fig. 2 (a)) where the goal is to predict the diagonal edges under a specific constraint: the only valid outcomes are either both edges are present or both are absent, i.e., $\mathcal{E} = \{[e_{02} = 1, e_{13} = 1], [e_{02} = 0, e_{13} = 0]\}$. Moreover, we assume that the probability of observing the first case is 0.6, while the second one is 0.4.

We first train an edge-wise prior using node2vec, which yields a probability of 0.6 for each diagonal edge. Crucially, because node2vec models each edge prediction independently, this prior is misspecified. A standard predictor based on this prior would always predict $[1, 1]$ if used as conditional mean or, if sampling were to be performed, could generate invalid predictions such as $[1, 0]$.

---

**Algorithm 1** Training and Sampling

---

1: **Training**
2: Sample $\mathbf{A}_1 \sim p(\mathbf{A})$, a mask $\xi$, and time $t \sim U[0, 1]$ (the CE instantiation instead draws $t$ from $p\,\delta_0 + (1 - p)\,U[0, 1]$ with $p = 0.25$; see Appendix D.4).
3: Train approximate MMSE estimator: $f_{\text{prior}}(\mathbf{A}^{\mathcal{O}})$
4: Compute
$\mathbf{A}_0 = \mathbf{A}^{\mathcal{O}} + (1 - \xi) \odot \left( f_{\text{prior}}(\mathbf{A}^{\mathcal{O}}) + \boldsymbol{\epsilon}_s \right), \quad \boldsymbol{\epsilon}_s \sim \mathcal{N}(0, \sigma_s^2)$
5: Compute $\mathbf{A}_t = (1 - t)\mathbf{A}_0 + t\mathbf{A}_1$.
6: Train flow model:
$\theta^* = \operatorname{argmin}_\theta \mathbb{E}_{\mathbf{A}_1, \mathbf{A}_0, \xi, t}\left[ \Delta\left( \hat{y}_\theta(\mathbf{A}_t, t), \ y(\mathbf{A}_0, \mathbf{A}_1) \right) \right]$ (the MSE instantiation adds the endpoint-anchor term of Eq. (14) with $\lambda_0 = 5$; see Appendix D.4)
7: **Sampling (Reconstruction)**
8: Initialize $\hat{\mathbf{A}} \leftarrow \xi \odot \mathbf{A}^{\mathcal{O}} + (1 - \xi) \odot f_{\text{prior}}(\mathbf{A}^{\mathcal{O}}) + (1 - \xi) \odot \boldsymbol{\epsilon}_s, \quad \boldsymbol{\epsilon}_s \sim \mathcal{N}(0, \sigma_{\text{samp}}^2)$.
9: **for** $i \leftarrow 0, \ldots, K - 1$ **do**
10:     **if** MSE distortion **then**
11:         $\hat{\mathbf{A}} \leftarrow \hat{\mathbf{A}} + \frac{1}{K}\, v_{\theta^*}\left( \hat{\mathbf{A}}, \frac{i}{K} \right)$
12:         $\hat{\mathbf{A}} \leftarrow \xi \odot \mathbf{A}^{\mathcal{O}} + (1 - \xi) \odot \hat{\mathbf{A}}$
13:     **else**
14:         *// CE distortion*
15:         $\mu \leftarrow \sigma\left( v_{\theta^*}(\hat{\mathbf{A}}, \frac{i}{K}) \right)$
16:         $\hat{\mathbf{A}} \leftarrow \operatorname{clip}\left( \hat{\mathbf{A}} + \frac{1}{K} \cdot \dfrac{\mu - \hat{\mathbf{A}}}{1 - i/K + \varepsilon}, \ 0, \ 1 \right)$
17:         $\hat{\mathbf{A}} \leftarrow \xi \odot \mathbf{A}^{\mathcal{O}} + (1 - \xi) \odot \hat{\mathbf{A}}$
18:     **end if**
19: **end for**
20: Return $\hat{\mathbf{A}}$

---

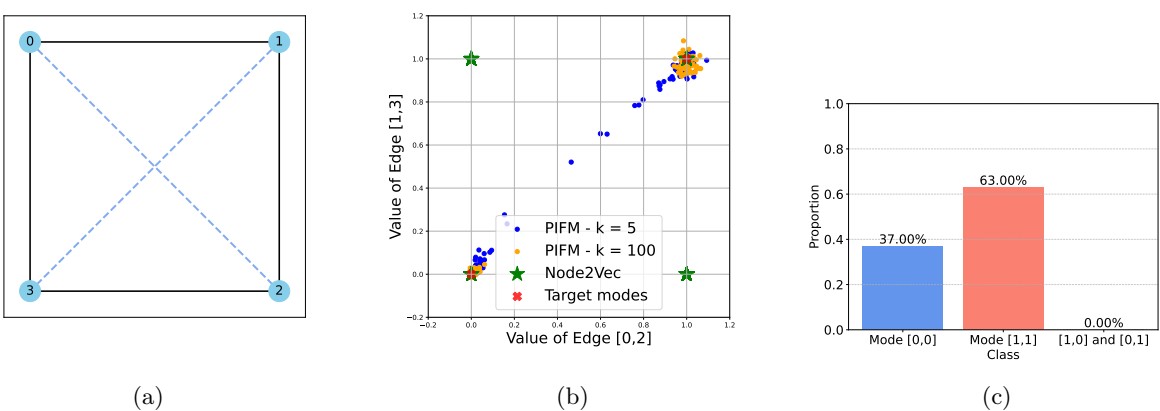

(a)                     (b)                     (c)

Figure 2: Toy experiment showcasing the advantage of PIFM (in this case, for link prediction). a) Graph $\mathcal{G}$ with four nodes, where the hidden edges are $e_{02}$ and $e_{13}$. b) Generated samples by using node2vec and PIFM (our proposed method): clearly, our method learns a probabilistic coupling, rendering a model that generates only the two valid modes. c) Proportions of samples generated with PIFM from each mode; remarkably, the method also learns a good approximation of the probability of each mode.

We then train a flow model using this node2vec prior to construct the initial state $\mathbf{A}_0$ as in (7). After training (see Appendix D for details), we generate 200 samples, illustrated in Fig. 2(b); the proportion of each mode is shown in Fig. 2(c). The results clearly demonstrate that the flow model (*i*) successfully leverages global information, learning a *probabilistic coupling* between the edges, to generate samples only from the two valid states, and (*ii*) learns the probability of each mode.

**Computational cost.** We remark that the flow model training time of PIFM is the **same** as flow model training time of PIFM with Gaussian prior, meaning that there is no additional cost when training the flow

model; the only added expense is training the prior when one is not already available. Inference is roughly 0.04–0.05s per graph at $K = 1$ on an NVIDIA A100 (vs. 0.0002–0.002s for the priors alone) and scales linearly in $K$ (e.g., $\sim$2.8s at $K{=}100$); full timings are in Appendix D.7 and D.8.

**Source distribution and prior model.** PIFM is agnostic to the prior architecture: any model that produces edge-wise marginal probabilities on the masked region can serve as the source distribution. In practice, we recommend validating candidate priors independently before flow training and selecting the strongest available edge predictor for the target domain.

## 5 Experiments

### 5.1 Setup

We evaluate our method on IMDB-B, PROTEINS, and ENZYMES (Morris et al., 2020), three inductive graph datasets (training and testing on disjoint graphs), and on CORA (McCallum et al., 2000; Yang et al., 2016), which is transductive and large-scale, adding additional challenges for training the flow-matching module. Thus, we focus on families of graphs that are diverse to show that our model learns a general predictor. For the inductive datasets, each one is split into 85% train, 10% validation, and 5% test graphs. We evaluate reconstruction quality under two masking levels. Each unordered node pair in the upper triangle is independently masked with probability 10% or 50%, regardless of whether the pair is an edge or a non-edge. The implementation details are provided in Appendix D.

**Evaluation metrics.** Performance is evaluated exclusively on masked unordered node pairs. We report both threshold-dependent and threshold-independent metrics. Threshold-dependent results use a fixed threshold of 0.5 in the main benchmark tables and include the false positive rate (FPR) and false negative rate (FNR). Threshold-independent performance is measured using ROC-AUC and average precision (AP).

We also evaluate a validation-selected operating point by choosing the threshold that maximizes the Matthews correlation coefficient (MCC) on the validation set and applying it unchanged to the test set. At this operating point, we report precision, recall, F1, MCC, balanced accuracy, normalized Hamming error, edge-count error, and graph-level recovery.

To assess distributional similarity between reconstructed and ground-truth graphs, we additionally report maximum mean discrepancy (MMD) (O'Bray et al., 2022), which serves as a proxy for perceptual quality. The appendix provides metric definitions and additional analyses, including validation-thresholded recovery, graph-statistic errors, connectivity, component counts, and Cora MMD (Appendices D.6, E.1, and E.2).

**Baselines.** We compare PIFM against several baselines, including diffusion-based. Recall that PIFM is composed of a one-shot prediction used as prior followed by a flow model. We compare PIFM to the accuracy of the one-shot prediction (without the flow) and with a flow with a random starting point:

- **Node2Vec Prior (Grover & Leskovec, 2016)/GraphSAGE Prior (Hamilton et al., 2017)/NCNC Prior (Wang et al., 2024)**: one-shot predictions using the structural prior directly.
- **LPFormer Prior** (Shomer et al., 2024): a task-specific graph Transformer.
- **Flow with Gaussian prior**: flow model initialized from Gaussian $\mathcal{N}(0.5, 1)$ noise on masked entries.
- **Direct GNN refiner**: a non-flow control matched to PIFM in inputs and parameter count but predicting the corrected adjacency in one forward pass at $t = 0$, with no interpolation and no ODE integration (results reported in Appendix E.7).
- **DiGress + RePaint (Vignac et al., 2023)**: unconditional DiGress combined with RePaint-style resampling (Lugmayr et al., 2022).
- **GDSS + RePaint (Jo et al., 2022)**: unconditional GDSS combined with RePaint-style resampling (Lugmayr et al., 2022).
- **VGAE** (Kipf & Welling, 2016): a graph autoencoder that learns latent node embeddings from the observed graph and reconstructs edges using a probabilistic decoder.

Algorithmic details of the baselines are provided in Appendix A.

**Scope and robustness checks.** Appendix E.12 reports categorical molecular reconstruction, four RePaint-adapted generators, Cora memory profiling, CE-weighting variants, and source-noise schedules. The molecular, generator, CE-weighting, and noise studies use one training seed and are therefore diagnostic extensions rather than statistically definitive model rankings. Together, they broaden the evaluated scope while leaving general heterogeneous graphs, optimized large-graph sampling, and cross-dataset adaptation open for future work.

## 5.2 Link Prediction

Table 2: Graph reconstruction with **50% of unordered node pairs masked**. Values are five-seed means $\pm$ sample standard deviations in percent; RePaint values are single runs. Stage-1 rows are frozen edge predictors evaluated directly, and each PIFM row uses the named predictor unchanged as its source. **Best** and second best are highlighted for the threshold-free metrics (AP, AUC); FNR and FPR are reported at the fixed 0.5 threshold, where they trade off against each other, and are therefore not ranked.

| Method | ENZYMES | | | | PROTEINS | | | | IMDB-B | | | |
|---|---|---|---|---|---|---|---|---|---|---|---|---|
| | AP↑ | AUC↑ | FNR↓ | FPR↓ | AP↑ | AUC↑ | FNR↓ | FPR↓ | AP↑ | AUC↑ | FNR↓ | FPR↓ |
| *Stage-1 priors (no flow)* | | | | | | | | | | | | |
| Node2Vec | $18.39_{\pm1.01}$ | $53.47_{\pm0.32}$ | $49.65_{\pm2.18}$ | $44.57_{\pm2.34}$ | $26.62_{\pm2.64}$ | $54.94_{\pm1.19}$ | $50.28_{\pm2.10}$ | $41.96_{\pm1.84}$ | $59.55_{\pm2.84}$ | $54.04_{\pm1.61}$ | $45.21_{\pm2.50}$ | $46.69_{\pm2.69}$ |
| GraphSAGE | $24.90_{\pm2.93}$ | $57.55_{\pm1.54}$ | $37.59_{\pm0.98}$ | $56.13_{\pm2.47}$ | $31.95_{\pm2.49}$ | $57.23_{\pm1.56}$ | $36.12_{\pm2.02}$ | $59.75_{\pm2.92}$ | $75.88_{\pm2.19}$ | $73.51_{\pm2.67}$ | $23.96_{\pm3.23}$ | $43.63_{\pm2.71}$ |
| VGAE | $27.04_{\pm2.29}$ | $64.57_{\pm2.00}$ | $36.29_{\pm3.50}$ | $42.25_{\pm0.34}$ | $34.14_{\pm2.99}$ | $64.36_{\pm1.64}$ | $39.73_{\pm2.69}$ | $40.80_{\pm0.77}$ | $71.76_{\pm2.96}$ | $69.75_{\pm2.17}$ | $40.80_{\pm1.88}$ | $31.85_{\pm3.04}$ |
| NCNC | $33.17_{\pm2.36}$ | $64.46_{\pm1.10}$ | $92.32_{\pm1.15}$ | $1.20_{\pm0.52}$ | $39.93_{\pm2.45}$ | $64.97_{\pm1.23}$ | $90.67_{\pm1.51}$ | $1.46_{\pm0.52}$ | $84.19_{\pm0.90}$ | $78.55_{\pm2.38}$ | $71.50_{\pm3.93}$ | $0.20_{\pm0.15}$ |
| LPFormer | $32.32_{\pm2.91}$ | $65.62_{\pm1.20}$ | $65.79_{\pm3.84}$ | $12.60_{\pm1.38}$ | $41.15_{\pm3.94}$ | $67.67_{\pm1.67}$ | $59.82_{\pm3.66}$ | $14.40_{\pm1.84}$ | $91.25_{\pm1.41}$ | $89.79_{\pm0.52}$ | $22.03_{\pm2.27}$ | $7.48_{\pm0.84}$ |
| *Generative/flow baselines* | | | | | | | | | | | | |
| DiGress + RePaint | 17.34 | 55.22 | 77.95 | 11.62 | 23.65 | 55.45 | 71.46 | 17.65 | 56.47 | 58.89 | 73.00 | 10.27 |
| GDSS + RePaint | 16.43 | 49.65 | 69.45 | 30.46 | 22.33 | 51.42 | 66.44 | 32.23 | 53.39 | 51.20 | 69.35 | 29.22 |
| Flow w/ Gaussian prior | $19.81_{\pm2.89}$ | $54.41_{\pm1.41}$ | $96.68_{\pm2.67}$ | $2.15_{\pm1.42}$ | $27.10_{\pm3.26}$ | $55.36_{\pm1.39}$ | $89.41_{\pm3.74}$ | $6.37_{\pm3.60}$ | $80.66_{\pm1.72}$ | $80.37_{\pm1.58}$ | $28.26_{\pm5.63}$ | $22.96_{\pm4.61}$ |
| *PIFM (MSE loss)* | | | | | | | | | | | | |
| PIFM (Node2Vec) | $25.29_{\pm2.65}$ | $62.48_{\pm1.25}$ | $96.82_{\pm2.07}$ | $1.50_{\pm1.12}$ | $34.88_{\pm2.79}$ | $64.96_{\pm1.33}$ | $86.16_{\pm2.49}$ | $5.90_{\pm2.42}$ | $85.77_{\pm1.48}$ | $84.75_{\pm1.20}$ | $24.27_{\pm2.67}$ | $18.18_{\pm2.61}$ |
| PIFM (GraphSAGE) | $27.14_{\pm2.61}$ | $62.55_{\pm0.88}$ | $96.80_{\pm1.80}$ | $1.16_{\pm0.91}$ | $35.61_{\pm3.30}$ | $64.91_{\pm2.14}$ | $85.15_{\pm5.13}$ | $5.11_{\pm2.49}$ | $92.95_{\pm1.15}$ | $93.46_{\pm1.11}$ | $15.81_{\pm0.71}$ | $9.02_{\pm2.67}$ |
| PIFM (VGAE) | $30.90_{\pm3.15}$ | $68.57_{\pm2.26}$ | $94.78_{\pm1.55}$ | $1.91_{\pm0.93}$ | $38.09_{\pm2.65}$ | $69.22_{\pm1.13}$ | $83.50_{\pm4.62}$ | $7.65_{\pm1.77}$ | $92.52_{\pm2.02}$ | $92.46_{\pm1.37}$ | $15.32_{\pm2.17}$ | $11.21_{\pm5.97}$ |
| PIFM (NCNC) | $34.38_{\pm4.43}$ | $67.07_{\pm3.01}$ | $87.87_{\pm6.11}$ | $2.63_{\pm1.71}$ | $41.52_{\pm4.41}$ | $68.00_{\pm3.51}$ | $63.64_{\pm5.26}$ | $14.60_{\pm5.99}$ | $92.92_{\pm0.52}$ | $91.07_{\pm2.00}$ | $18.53_{\pm3.59}$ | $10.10_{\pm6.17}$ |
| PIFM (LPFormer) | $35.99_{\pm3.01}$ | $73.16_{\pm1.39}$ | $92.35_{\pm2.07}$ | $1.79_{\pm0.59}$ | $43.40_{\pm3.05}$ | $74.12_{\pm0.98}$ | $80.20_{\pm6.00}$ | $6.74_{\pm1.86}$ | $95.28_{\pm1.11}$ | $94.99_{\pm0.48}$ | $9.79_{\pm1.67}$ | $7.59_{\pm2.01}$ |
| *PIFM (CE loss)* | | | | | | | | | | | | |
| PIFM (Node2Vec) | $25.13_{\pm2.65}$ | $62.17_{\pm1.24}$ | $28.48_{\pm10.30}$ | $60.52_{\pm10.36}$ | $33.17_{\pm3.33}$ | $63.79_{\pm1.44}$ | $9.03_{\pm5.23}$ | $86.45_{\pm6.76}$ | $85.74_{\pm1.31}$ | $84.59_{\pm1.46}$ | $19.16_{\pm5.91}$ | $25.81_{\pm6.06}$ |
| PIFM (GraphSAGE) | $27.35_{\pm3.57}$ | $61.97_{\pm2.14}$ | $33.00_{\pm7.47}$ | $58.51_{\pm7.87}$ | $34.24_{\pm3.04}$ | $63.40_{\pm1.15}$ | $9.45_{\pm4.63}$ | $88.24_{\pm5.23}$ | $91.96_{\pm1.28}$ | $92.14_{\pm1.11}$ | $14.92_{\pm3.11}$ | $13.91_{\pm3.05}$ |
| PIFM (VGAE) | $30.19_{\pm2.36}$ | $68.21_{\pm1.55}$ | $24.71_{\pm8.44}$ | $55.02_{\pm7.94}$ | $37.75_{\pm2.87}$ | $68.54_{\pm1.18}$ | $10.57_{\pm3.84}$ | $81.21_{\pm16.14}$ | $91.92_{\pm1.43}$ | $91.41_{\pm0.70}$ | $16.01_{\pm1.61}$ | $13.33_{\pm3.50}$ |
| PIFM (NCNC) | $38.45_{\pm3.97}$ | $75.35_{\pm1.77}$ | $26.42_{\pm4.29}$ | $39.40_{\pm5.75}$ | $45.05_{\pm1.99}$ | $75.01_{\pm1.71}$ | $14.03_{\pm5.30}$ | $62.70_{\pm8.45}$ | $94.14_{\pm0.83}$ | $93.07_{\pm1.79}$ | $12.13_{\pm2.02}$ | $13.59_{\pm4.93}$ |
| PIFM (LPFormer) | $34.21_{\pm3.17}$ | $72.06_{\pm2.02}$ | $23.91_{\pm6.46}$ | $49.68_{\pm8.25}$ | $42.03_{\pm3.66}$ | $72.95_{\pm2.59}$ | $12.97_{\pm2.88}$ | $69.07_{\pm6.98}$ | $95.08_{\pm1.42}$ | $94.75_{\pm0.71}$ | $9.28_{\pm2.51}$ | $10.37_{\pm3.09}$ |

Table 2 reports results for 50% masking, averaged over five random seeds. Results for 10% masking are provided in Appendix E.3. Across all three datasets, PIFM trained with the MSE objective improves the AUC of every corresponding Stage-1 prior. In particular, LPFormer is the strongest Stage-1 prior considered in this study, yet PIFM further increases its AUC by 7.54, 6.45, and 5.20 percentage points on ENZYMES, PROTEINS, and IMDB-B, respectively. These consistent gains suggest that the flow-based refinement captures global and joint information that the prior missed. Appendix E.2 further analyzes the reconstructed graphs beyond edge-wise ranking.

We notice that both distortions improve ranking over their Stage-1 sources in most metrics. At the fixed 0.5 cutoff, CE generally lowers FNR on sparse graphs while increasing FPR; MSE emphasizes the continuous-score distortion used by the flow objective. We therefore report both ranking and thresholded behavior without treating the losses as interchangeable; additional calibration results are in Appendix E.6. The prior supplies task-specific edge scores, while the flow learns a correction over the reconstructed adjacency matrix, and their benefits remain complementary across the tested prior families.

**Large-scale graph reconstruction.** We evaluate PIFM on Cora over five seeds as a large-scale feasibility check, not as a leaderboard comparison. The MSE model uses one-hop, single-edge subgraphs, whereas the CE model uses two-hop, multi-edge subgraphs. Appendix E.11 includes a diagnostic that changes only the neighborhood radius. Because it uses one seed and evaluates on the test split, we do not use it to claim that either neighborhood depth improves or worsens performance.

Table 3: Cora results over five seeds. AUC measures held-out-edge ranking; the remaining columns are subgraph-pool MMD$^2$ values. **Bold blue** values indicate the best result(s) in each metric: higher is better for AUC, while lower is better for MMD$^2$. The prior is repeated because each configuration induces a different subgraph pool.

| Configuration | Method | AUC $\uparrow$ | Clustering MMD $\downarrow$ | Degree MMD $\downarrow$ | NSPDK MMD $\downarrow$ |
|---|---|---|---|---|---|
| MSE, single-edge, 1-hop | NCN prior | $93.49 \pm 0.55$ | $.122 \pm .062$ | $.070 \pm .030$ | $.061 \pm .026$ |
| | PIFM | $93.36 \pm 0.66$ | $.100 \pm .080$ | $.072 \pm .063$ | $.060 \pm .044$ |
| CE, multi-edge, 2-hop | NCN prior | $93.49 \pm 0.55$ | $.172 \pm .086$ | $.085 \pm .041$ | $.030 \pm .009$ |
| | PIFM | $87.17 \pm 11.11$ | $.053 \pm .044$ | $.040 \pm .025$ | $.020 \pm .007$ |

For MSE, PIFM is on par with the NCN prior on AUC, and the small paired difference does not reach significance at five seeds. On the other side, PIFM slightly improves the clustering and NSPDK means, while degree MMD is nearly unchanged. Regarding the CE multi-edge configuration, it presents higher AUC variability but improves all three distributional metrics for all seeds. Additionlly for MSE, PIFM matches the NCN prior on AUC with slight improvements in clustering and NSPDK MMD. The CE multi-edge version shows higher AUC variability but improves all three distributional metrics on every seed: even where PIFM does not improve ranking, it brings the reconstructed graphs closer to the clean distribution. Since the two configurations differ in more than the loss, additional results and ablations are in Appendix E.11.

## 5.3 Blind graph reconstruction

We focus on two blind versions of link prediction, namely expansion and denoising. In the expansion case, we only get to observe a subset of the edges (but no non-edges), and we need to determine which other entries correspond to existing edges. Conversely, for denoising, we observe a set of edges that includes some spurious ones (but no confirmed non-edges), and we need to determine which of the observed edges are spurious and should be removed. These cases are more challenging than link prediction since transductive priors like node2vec cannot be trained on the masked graphs (since we do not have positive and negative edges). We report expansion here; denoising is in Appendix E.5.

**Expansion.** The goal in expansion is to predict a set of hidden edges $\mathcal{E}_M$ given $\mathbf{A}^{\mathcal{O}}$, such that the edge set of the ground truth is $\mathcal{E} = \mathcal{E}_M \cup \mathcal{E}_O$. Therefore, defining $\mathbf{A}_1 = \mathbf{A}$, the initialization becomes $\mathbf{A}_0 = \mathbf{A}^{\mathcal{O}} + (1 - \mathbf{A}^{\mathcal{O}}) \odot (f_{prior}(\mathbf{A}^{\mathcal{O}}) + \boldsymbol{\epsilon}_s)$. Throughout the blind-reconstruction experiments (expansion here and denoising in Appendix E.5), we use PIFM with the MSE loss. The results for a drop rate of 50% are shown in Table 4.

Across five seeds, PIFM (GraphSAGE) attains the top AP and AUC on all three datasets, improving over the diffusion baselines. Furthermore, it raises its own Stage-1 prior AUC by 10.3, 10.2, and 13.6 percentage points on ENZYMES, PROTEINS, and IMDB-B, respectively, with the gaps exceeding the seed-to-seed standard deviations. The uninformed flow with a Gaussian source stays near random guess with $p = 0.5$ on the sparse datasets (50.60 on ENZYMES and 51.73 on PROTEINS), so the gain for the flow module requires an informative prior estimate. On the denser IMDB-B, the uninformed flow reaches 63.72 AUC, illustrating that denser graphs have a richer structure that can be exploited even without a prior. As in link prediction, FNR at the fixed 0.5 threshold remains high on the sparse datasets under the MSE loss; Appendix E.1 reports validation-selected thresholds. Lastly, we remark that varying the Euler step count yields empirical operating points whose distortion and distributional behavior are examined in Appendix E.9; the step count is not treated as a calibrated trade-off parameter.

## 5.4 Distortion-perception trade-off.

We further analyze the distortion–perception trade-off using MMD as a proxy for perceptual quality, following GraphRNN (You et al., 2018), GDSS (Jo et al., 2022), and DeFoG (Qin et al., 2025).

Table 4: Performance for the **expansion** task with **50% of edges masked (0.5 Drop)** (see Table 17 for definitions). Values are five-seed means ± sample standard deviations in percent; [†] marks the RePaint baselines, which are single runs. Metrics are computed on the region the flow updates, $1 - \mathbf{A}^{\mathcal{O}}$. FNR and FPR use the fixed 0.5 threshold; see Appendix E.1 for validation-selected thresholds.

| | Mask Rate: 50% (0.5 Drop) | | | | | | | | | | | |
|---|---|---|---|---|---|---|---|---|---|---|---|---|
| | ENZYMES | | | | PROTEINS | | | | IMDB-B | | | |
| Method | AP ↑ | AUC ↑ | FNR ↓ | FPR ↓ | AP ↑ | AUC ↑ | FNR ↓ | FPR ↓ | AP ↑ | AUC ↑ | FNR ↓ | FPR ↓ |
| *Baselines* | | | | | | | | | | | | |
| GraphSAGE | 16.02±2.22 | 57.54±1.69 | 37.79±1.16 | 56.02±2.52 | 22.73±2.82 | 57.01±1.48 | **36.22**±2.19 | 60.83±2.85 | 67.30±2.72 | 73.26±2.91 | **24.09**±3.36 | 43.81±3.02 |
| DiGress + RePaint[†] | 8.09 | 50.69 | 87.21 | 11.40 | 13.50 | 52.46 | 80.02 | 15.07 | 39.70 | 53.07 | 81.10 | 13.00 |
| GDSS + RePaint[†] | 9.21 | 49.63 | 69.45 | 30.80 | 14.66 | 51.03 | 66.44 | 32.06 | 39.43 | 50.68 | 69.35 | 30.00 |
| Flow w/ Gaussian prior | 9.94±1.07 | 50.60±1.48 | 100.00±0.00 | **0.00**±0.00 | 16.81±2.67 | 51.73±0.50 | 94.71±2.83 | **4.25**±2.62 | 56.61±3.40 | 63.72±2.47 | 64.63±7.49 | 12.26±5.64 |
| *Ours* | | | | | | | | | | | | |
| PIFM (GraphSAGE) | **19.59**±2.00 | **67.86**±1.17 | 99.05±0.88 | 0.27±0.20 | **26.34**±2.05 | **67.22**±1.94 | 92.44±3.25 | 4.43±1.79 | **82.52**±0.90 | **86.89**±2.21 | 28.55±3.77 | **10.47**±4.29 |

Figure 3 illustrates this trade-off on ENZYMES. Increasing the number of Euler steps decreases $\mathrm{MMD}^2$ and moves degree, triangle, and clustering statistics closer to the reference distribution, whereas fewer steps produce reconstructions that favor edge-wise distortion metrics. Similar behavior appears in the five-seed LPFormer/MSE/50%-mask study: increasing the number of steps from $K = 1$ to $K = 100$ reduces NSPDK $\mathrm{MMD}^2$ by approximately $3.1\times$ on ENZYMES and $3.5\times$ on PROTEINS, although at the cost of lower AUC. On IMDB-B, however, increasing $K$ slightly increases MMD, indicating that the effect is dataset dependent.

Similar empirical trends with increasing integration steps have been reported for PMRF (Ohayon et al., 2025) and InDI (Delbracio & Milanfar, 2024). Nevertheless, these observations should not be interpreted as establishing a theoretical or solver-independent distortion–perception control. Understanding estimator-level interpolation, fixed-accuracy solver comparisons, and explicit terminal-time control remains future work. Additional analyses are provided in Appendices E.9, E.7, and F.1.

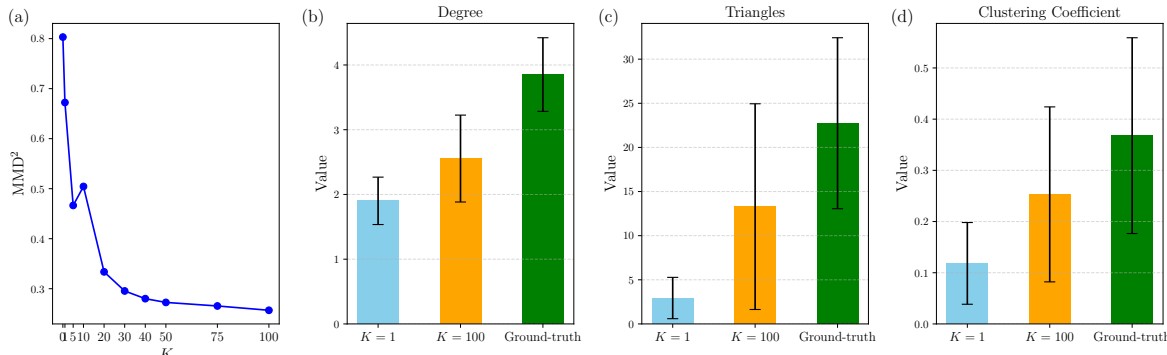

Figure 3: Empirical distortion–perception behavior on the ENZYMES expansion experiment. (a) MMD measures distance to the reference graph distribution and is lower after additional integration in this setting. (b–d) At $K = 100$, degree, triangle, and clustering-coefficient statistics more closely match the ground truth than at $K = 1$. This dataset-specific comparison is not a claim that MMD decreases monotonically with $K$ in every setting. Error bars show the standard deviation over 300 samples (10 per test graph).

## 6    Conclusions

In this paper, we introduced **Prior-Informed Flow Matching (PIFM)**, a flow-based estimator that learns global structural information by refining a local edge-prediction prior. PIFM is grounded in the distortion–perception framework: it transports a local edge-wise prior estimate (obtained from GraphSAGE, node2vec, VGAE, NCNC, or LPFormer) toward the clean graph distribution, under either a mean squared error or cross-entropy distortion. Experiments on diverse benchmarks show that PIFM consistently outperforms both classical embedding methods and recent diffusion/flow-based baselines. The current formulation handles mainly homogeneous graphs, though we showed the feasibility of extending PIFM to categorical data via the ZINC dataset experiment. Therefore, extending it to heterogeneous data via simplex-based (Eijkelboom et al., 2024) or discrete (Qin et al., 2025) flow formulations is another promising direction for future research.

Each limitation suggests a clear direction. The frozen, separately trained Stage-1 prior keeps PIFM modular but cannot adapt to the flow, and end-to-end training would require handling the resulting non-stationary source distribution, for example through a posterior-mean anchor and separate update timescales. The continuous relaxation could instead use categorical-simplex or discrete Markov dynamics, with rounding error reported alongside ranking metrics. Finally, scaling beyond the dense $B \times N \times N$ edge state will require sparse or block-wise subgraph training, evaluated for accuracy, memory, and latency over multiple seeds.

## Impact statement

Graph reconstruction can infer relationships that are absent from an observed network. In social, communication, biological, or organizational data, a missing edge may represent sensitive information that participants withheld or never consented to expose. PIFM should therefore be used only for authorized reconstruction and within a clearly limited purpose; predictions should be accompanied by uncertainty and should not be presented as established relationships or used alone to make decisions about individuals. Relevant safeguards include data minimization, consent and purpose limitation, access controls, privacy-preserving preprocessing where appropriate, differential-harm evaluation, and human or institutional review. Our public-benchmark experiments do not establish suitability for surveillance, deanonymization, or disclosure of private ties.

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

## Appendix

## A    Algorithm

In this section, we describe the algorithms that we use as baselines. Each method serves distinct purposes: the priors alone test whether the flow model provides meaningful improvement beyond the one-shot estimates given by the priors, uniform + flow evaluates whether the prior methods used (i.e. Graph-SAGE/Node2Vec/VGAE/NCNC/LPFormer) are good structural priors for effective denoising, and DiGress + RePaint compares our model to standard modified unconditional generation models.

### A.1    Uniform + flow Baseline

This baseline ablates the structural prior by initializing the flow from a state where unknown entries are filled with uniform noise. The model then learns to denoise from this less-informed starting point.

---

**Algorithm 2** Uniform + flow Training and Sampling

---

1: **Training**
2: Sample $\mathbf{A}_1$, a mask $\xi$, and time $t \sim U[0, 1]$.
3: Define initial state with a uniform fill plus a small Gaussian perturbation on the masked region: $\mathbf{A}_0 = \mathbf{A}^{\mathcal{O}} + (1 - \xi) \odot \mathcal{U}(0, 1)^{N \times N} + (1 - \xi) \odot \boldsymbol{\epsilon}_{\text{train}}$, where $\boldsymbol{\epsilon}_{\text{train}} \sim \mathcal{N}(0, \sigma_{\text{train}}^2)$.
4: Define interpolant $\mathbf{A}_t = (1 - t)\mathbf{A}_0 + t\mathbf{A}_1$.
5: Solve $\theta^* = \operatorname{argmin}_\theta \mathbb{E}_{\mathbf{A}_1, \xi, t} \| v_\theta(\mathbf{A}_t, t) - (\mathbf{A}_1 - \mathbf{A}_0) \|_F^2$.
6: **Sampling (Reconstruction)**
7: Given observed graph $\mathbf{A}^{\mathcal{O}}$, define the initial state with masked noise: $\hat{\mathbf{A}} \leftarrow \xi \odot \mathbf{A}^{\mathcal{O}} + (1 - \xi) \odot \mathcal{U}(0, 1)^{N \times N} + (1 - \xi) \odot \boldsymbol{\epsilon}_{\text{samp}}$, where $\boldsymbol{\epsilon}_{\text{samp}} \sim \mathcal{N}(0, \sigma_{\text{samp}}^2)$.
8: **for** $i \leftarrow 0$ to $K - 1$ **do**
9:     $\hat{\mathbf{A}} \leftarrow \hat{\mathbf{A}} + \frac{1}{K} v_{\theta^*}\left(\hat{\mathbf{A}}, \frac{i}{K}\right)$
10:     $\hat{\mathbf{A}} \leftarrow \xi \odot \mathbf{A}^{\mathcal{O}} + (1 - \xi) \odot \hat{\mathbf{A}}$          *// re-impose observed entries*
11: **end for**
12: **Return** $\hat{\mathbf{A}}$

---

### A.2    DiGress + RePaint Baseline

**Training (Unconditional)**    The model $p_\theta$ is trained unconditionally on complete graphs $\mathbf{A}_0 \sim p_{\text{data}}$ to reverse a discrete forward noising process $q$. The forward process is a fixed Markov chain $q(\mathbf{A}_t | \mathbf{A}_{t-1})$ that corrupts the graph over $T$ steps. The training objective is to learn the denoising distribution $p_\theta(\mathbf{A}_0 | \mathbf{A}_t)$, modeled as a categorical prediction task for each node and edge.

---

**Algorithm 3** DiGress Unconditional Training

---

1: Sample $t \sim \mathcal{U}\{1, \ldots, T\}$ and $\mathbf{A}_t \sim q(\mathbf{A}_t \mid \mathbf{A}_0)$.          */\* Forward process \*/*
2: Predict $\mathbf{A}_0$ from $(\mathbf{A}_t, t)$ with $p_\theta(\cdot, t)$.          */\* Denoising objective \*/*
3: Minimize expected cross-entropy:

$$\theta^* = \arg\min_\theta \mathbb{E}_{\mathbf{A}_0 \sim p_{\text{data}}, \, t \sim \mathcal{U}\{1, \ldots, T\}} \left[ \mathcal{L}_{\text{CE}}(\mathbf{A}_0, \, p_\theta(\mathbf{A}_t, t)) \right].$$

---

**Sampling (Conditional Reconstruction via RePaint)**    At inference, given an observed graph $\mathbf{A}^{\mathcal{O}} = \xi \odot \mathbf{A}_0$, the unconditionally trained model $p_{\theta^*}$ generates the missing entries. This is achieved by iteratively re-imposing the known (unmasked) information during the reverse diffusion process (Lugmayr et al., 2022).

---

**Algorithm 4** DiGress + RePaint Sampling

---

1: **Input:** Observed graph $\mathbf{A}^{\mathcal{O}}$, mask $\xi$, trained model $p_{\theta^*}$, steps $T$.
2: **Output:** Reconstructed graph $\hat{\mathbf{A}}_0$.
3: Initialize $\hat{\mathbf{A}}_T \sim p_{\text{prior}}(\cdot)$, where $p_{\text{prior}}$ is a random graph distribution.
4: **for** $t = T, T-1, \ldots, 1$ **do**
5:    *// Predict clean graph from current state*
6:    $\tilde{\mathbf{A}}_0 = p_{\theta^*}(\hat{\mathbf{A}}_t, t)$.
7:
8:    *// Impose known data by noising it to the next step*
9:    $\mathbf{A}_{t-1}^{\text{known}} \sim q(\mathbf{A}_{t-1}|\mathbf{A}^{\mathcal{O}})$.
10:
11:    *// Sample the unknown region by noising the prediction to the next step*
12:    $\mathbf{A}_{t-1}^{\text{unknown}} \sim q(\mathbf{A}_{t-1}|\tilde{\mathbf{A}}_0)$.
13:
14:    *// Combine known and unknown parts for the next state*
15:    $\hat{\mathbf{A}}_{t-1} = \xi \odot \mathbf{A}_{t-1}^{\text{known}} + (1 - \xi) \odot \mathbf{A}_{t-1}^{\text{unknown}}$.
16: **end for**
17: Return $\hat{\mathbf{A}}_0$

---

### A.3 Node2Vec Prior (Per-Graph Classifier)

This baseline learns a *per-graph* edge-probability model from the observed subgraph. We (i) fit node2vec embeddings on the observed topology and (ii) train a logistic classifier on Hadamard edge features to produce probabilities on the masked pairs.

---

**Algorithm 5** Node2Vec Prior: Training and Inference

---

1: **Inputs:** Full adjacency $\mathbf{A}_1$, mask $\xi$ ($\xi_{ij} = 1$ if observed), Node2Vec hyperparams (dim $d$, walk length $L$, walks/node $R$, window $w$, $p, q$), negatives/positive ratio $k$.
2: **Outputs:** Probabilities $\hat{P}$ on masked entries, i.e., $f_{\text{prior}}(\mathbf{A}^{\mathcal{O}})$.
3: **Training (per graph)**
4: Construct observed graph $\mathbf{A}^{\mathcal{O}} \leftarrow \xi \odot \mathbf{A}_1$.
5: Train Node2Vec on $\mathbf{A}^{\mathcal{O}}$ to obtain embeddings $\{\mathbf{z}_i\}_{i=1}^N \in \mathbb{R}^d$.
6: Build labeled edge set on observed pairs ($i < j$):
7: $\mathcal{P}^+ = \{(i, j) : \xi_{ij} = 1, A_{ij} = 1\}, \quad \mathcal{P}^- \sim k\text{-to-1 balanced samples from } \{(i, j) : \xi_{ij} = 1, A_{ij} = 0\}$.
8: Features: $\mathbf{x}_{ij} \leftarrow \mathbf{z}_i \odot \mathbf{z}_j$ (Hadamard);   Labels: $y_{ij} \in \{0, 1\}$.
9: Fit logistic classifier $g_\phi(\mathbf{x}) = \sigma(\mathbf{w}^\top \mathbf{x} + b)$ (L2; class-balanced).
10: **Inference (per graph)**
11: For each masked pair $(i, j)$ with $\xi_{ij} = 0$, compute $\mathbf{x}_{ij} \leftarrow \mathbf{z}_i \odot \mathbf{z}_j$.
12: Predict $\hat{P}_{ij} \leftarrow g_\phi(\mathbf{x}_{ij})$ and set $\hat{P}_{ji} \leftarrow \hat{P}_{ij}$.
13: Return $\hat{P}$ as $f_{\text{prior}}(\mathbf{A}^{\mathcal{O}})$ (used in Eq. (7)).

---

*Notes.* (i) We train embeddings *only* on $\mathbf{A}^{\mathcal{O}}$ to avoid leakage. (ii) The Hadamard feature works well and is symmetric; concatenation can be used but breaks symmetry unless sorted. (iii) Thresholding at 0.5 yields hard reconstructions; we use scores $\hat{P}$ directly in PIFM.

## B Extended background

### B.1 Graphons and graphon estimation

A *graphon*, defined as a symmetric function $\mathcal{W} : [0, 1]^2 \to [0, 1]$, serves as a generative model for a family of graphs:

$$z_i \sim \text{Uniform}[0, 1], \quad i = 1, \ldots, n, \tag{11}$$
$$A_{ij} \sim \text{Bernoulli}(\mathcal{W}(z_i, z_j)), \quad 1 \le i < j \le n.$$

Graphons provide a functional representation of exchangeable random graphs where the conditional edge probability is $[\mathbb{E}[\mathbf{A} \mid \mathbf{z}]]_{ij} = \mathcal{W}(z_i, z_j)$. This offers a natural, permutation-equivariant framework for estimating the posterior mean, though it requires access to the inverse mapping $z_i = [\mathbf{z}^{-1}(\mathbf{A}^{\mathcal{O}})]_i$. Since $\mathcal{W}$ is unknown, we estimate it using Scalable Implicit Graphon Learning (SIGL) (Azizpour et al., 2025), which combines a graph neural network (GNN) encoder with an implicit neural representation (INR). SIGL operates in three steps: (1) a GNN-based sorting step to estimate latent node positions $\mathbf{z}$; (2) a histogram approximation of the sorted adjacency matrices; and (3) learning a graphon parameterization $f_\phi$ by minimizing its error against the histograms. A key feature of SIGL is its ability to recover the inverse mapping $\mathbf{z}^{-1}$, making it uniquely suitable for our model (Xia et al., 2023).

The generative process in (11) can also be viewed in reverse: given a collection of graphs (represented by their adjacency matrices) $\mathcal{D} = \{\mathbf{A}_t\}_{t=1}^M$ that are sampled from an *unknown* graphon $\mathcal{W}$, estimate $\mathcal{W}$. Several methods have been proposed for this task (Chan & Airoldi, 2014; Airoldi et al., 2013; Xu et al., 2021; Xia et al., 2023; Azizpour et al., 2025). We focus on SIGL (Azizpour et al., 2025), a resolution-free method that, in addition to estimating the graphon, also *infers the latent variables* $\boldsymbol{\eta}$. This method parameterizes the graphon using an implicit neural representation (INR) (Sitzmann et al., 2020), a neural architecture defined as $f_\phi(x, y) : [0, 1]^2 \to [0, 1]$ where the inputs are coordinates from $[0, 1]^2$ and the output approximates the graphon value $\mathcal{W}$ at a particular position. In a nutshell, SIGL works in three steps: (1) a sorting step using a GNN $g_{\phi'}(\mathbf{A})$ that estimates the latent node positions or representations $\boldsymbol{\eta}$; (2) a histogram approximation of the sorted adjacency matrices; and (3) learning the parameters $\phi$ by minimizing the mean squared error between $f_\phi(x, y)$ and the histograms (obtained in step 2).

## B.2 Node2vec

*node2vec* (Grover & Leskovec, 2016) is a scalable model for learning continuous node representations in graphs. This method is *transductive*, meaning that it generates an embedding for each graph. It extends the Skip-gram model from natural language processing to networks by sampling sequences of nodes through biased random walks. Node2vec introduces two hyperparameters $(p, q)$ that interpolate between breadth-first and depth-first exploration. This flexibility allows embeddings to capture both *homophily* (nodes in the same community) and *structural equivalence* (nodes with similar roles, e.g., hubs), which frequently coexist in real-world graphs.

The embeddings are learned via stochastic gradient descent with negative sampling to maximize the likelihood of preserving sampled neighborhoods. Once learned, node embeddings can be combined through simple binary operators (e.g., Hadamard product) to form edge features, enabling applications such as link prediction. Empirically, node2vec has been shown to outperform prior unsupervised embedding methods across tasks like classification and link recovery, while remaining computationally efficient and scalable to large graphs (Grover & Leskovec, 2016).

## B.3 GraphSAGE

GraphSAGE (Hamilton et al., 2017) is an *inductive* technique for link prediction based on graph neural networks (GNN) framework designed to generate embeddings for nodes in large, evolving graphs. It consists of two steps: for a target node, it first samples a fixed-size neighborhood of adjacent nodes, and then it aggregates feature information from these sampled neighbors. By learning aggregation functions (such as a mean, pool, or LSTM aggregator) rather than embeddings for every single node, GraphSAGE can efficiently generate predictions for nodes that were not part of the training set, making it highly scalable and effective for real-world applications like social networks and recommendation systems.

## B.4 NCN and NCNC

*Neural Common Neighbor* (NCN) (Wang et al., 2024) is a link predictor that combines a message-passing encoder with a structural feature: node features $\{\mathbf{h}_i\}_{i=1}^N$ are produced by a single MPNN pass on the graph, and the score for an edge $(i, j)$ is a learned function of the Hadamard term $\mathbf{h}_i \odot \mathbf{h}_j$ and a pooled sum over common-neighbor representations $\sum_{u \in N(i) \cap N(j)} \mathbf{h}_u$. Because the MPNN runs only once per graph rather

than once per target link, NCN is significantly more scalable than methods that apply MPNN to a subgraph for each candidate edge, while remaining expressive enough to distinguish links by the identity of their common neighbors.

*NCN with Completion* (NCNC) extends NCN to address graph incompleteness: when the observed graph is partial, true common neighbors may be unobserved, distorting the structural feature on which NCN relies. NCNC softly completes the common-neighbor set by extending the pooled sum to all candidate intermediaries $u \in N(i) \cup N(j)$ and weighting each by a learned link-existence probability (1 if $u$ is already an observed common neighbor; the predicted probability of the unobserved endpoint-to-$u$ edge if exactly one endpoint connects to $u$; and 0 otherwise). In our work we use the depth-1 NCNC predictor (a single round of soft completion before re-applying NCN), corresponding to the `IncompleteCN1Predictor` module of the official implementation.

### B.5 Graph Diffusion Models

Diffusion models are generative frameworks composed of two processes: a **forward process** that systematically adds noise to data until it becomes pure noise, and a **reverse process** that learns to reverse this, generating new data by starting from noise and progressively denoising it. While these models exist for both discrete (Vignac et al., 2023) and continuous domains (Jo et al., 2022), we describe the continuous case which is the most related to our method. Here, a graph $\mathbf{G}_0$ is defined by its node features $\mathbf{X}_0 \in \mathbb{R}^{N \times F}$ and its weighted adjacency matrix $\mathbf{A}_0 \in \mathbb{R}^{N \times N}$. Following the GDSS framework, the forward process is described by a stochastic differential equation (SDE) that gradually perturbs the graph data over a time interval $t \in [0, T]$:

$$d\mathbf{G}_t = -\frac{1}{2}\beta(t)\mathbf{G}_t \, dt + \sqrt{\beta(t)} \, d\mathbf{W}_t$$

In this equation, $\mathbf{W}_t$ represents standard Brownian motion (i.e., noise), and $\beta(t)$ is a noise schedule that typically increases over time. This process is designed so that by the final time $T$, the noised graph $\mathbf{G}_T$ is indistinguishable from a standard Gaussian.

The generative reverse process is defined by another SDE that traces the path from noise back to data. This process relies on the **score function**, $\nabla_{\mathbf{G}_t} \log p(\mathbf{G}_t)$, which is the gradient of the log-density of the noisy data at time $t$. Since the true score function is unknown, it must be approximated. This is done using a neural network, or **score network**, which is trained to predict the score. For graphs, separate networks are often used for the adjacency matrix and node features: $\boldsymbol{\epsilon}_{\boldsymbol{\theta}_A}(\mathbf{A}_t, t)$ and $\boldsymbol{\epsilon}_{\boldsymbol{\theta}_X}(\mathbf{X}_t, t)$. These networks are trained by minimizing the denoising score-matching loss.

Once trained, these score networks can be plugged into the reverse SDE. New graphs are then generated by solving this SDE numerically using standard samplers like DDPM or DDIM.

#### B.5.1 Diffusion-based inverse problems solver for graphs

We now expand on diffusion-based solvers for graph inverse problems. Given a condition $\mathcal{C}$ and a reward function $r(\mathbf{G}_0)$ that quantifies how close the sample $\mathbf{G}_0$ is to meeting $\mathcal{C}$, the objective is to generate graphs $G_0$ that maximize the reward function. From a Bayesian perspective, this problem boils down to sampling from the posterior $p(\mathbf{G}_0|\mathcal{C}) \propto p(\mathcal{C}|\mathbf{G}_0)p(\mathbf{G}_0)$ where $p(\mathcal{C}|\mathbf{G}_0) \propto \exp(r(\mathbf{G}_0))$ is a likelihood term and $p(\mathbf{G}_0)$ is a prior given by the pre-trained diffusion model. We now describe previous works for both differentiable and non-differentiable reward functions.

**Guidance with Differentiable Reward Functions.** Several approaches have been developed to guide generative models when the objective can be expressed as a **differentiable reward function**, particularly for inverse problems in imaging. These methods typically leverage the differentiability of the reward – often a likelihood tied to a noisy measurement – to calculate a *conditional score* using Bayes' rule:

$$\nabla_{\mathbf{G}_t} \log p(\mathbf{G}_t|\mathcal{C}) = \nabla_{\mathbf{G}_t} \log p(\mathcal{C}|\mathbf{G}_t) + \nabla_{\mathbf{G}_t} \log p(\mathbf{G}_t)$$

In this formulation, the diffusion model naturally serves as the prior $(p(\mathbf{G}_t))$, while the likelihood term $(p(\mathcal{C}|\mathbf{G}_t))$ provides the guidance. However, directly computing the score of the likelihood term is intractable because it requires integrating over all possible clean data: $p(\mathcal{C}|\mathbf{G}_t) = \int p(\mathcal{C}|\mathbf{G}_0)p(\mathbf{G}_0|\mathbf{G}_t)d\mathbf{G}_0$.

To overcome this, a common technique is to approximate the posterior distribution $p(\mathbf{G}_0|\mathbf{G}_t)$ with a Gaussian centered at the MMSE denoiser. This denoised estimate can be calculated efficiently using **Tweedie's formula**:

$$\mathbb{E}[\mathbf{G}_0|\mathbf{G}_t] = \frac{1}{\alpha_t}\left(\mathbf{G}_t + \sigma_t^2 \nabla_{\mathbf{G}_t} \log p(\mathbf{G}_t, t)\right)$$

While this framework is established for images, its application to graph-based inverse problems is less explored. This is primarily because most interesting properties and constraints in graphs are **not differentiable**. Some graph-specific methods, like DiGress (Vignac et al., 2023), implement guidance by training an auxiliary model, similar to classifier-free guidance, which introduces additional complexity.

**Guidance with Non-Differentiable Reward Functions.** For the more common scenario of non-differentiable constraints in graph generation, alternative strategies have emerged. The **PRODIGY** method, for instance, operates by repeatedly applying a two-step process at each denoising step: generation followed by projection.

First, it uses the unconditional diffusion model to produce a candidate sample $\hat{\mathbf{G}}_{t-1}$. Second, it projects this candidate onto the set of valid solutions using a projection operator: $\Pi_{\mathcal{C}}(\hat{\mathbf{G}}_{t-1}) = \arg\min_{\mathbf{Z}\in\mathcal{C}} \|\mathbf{Z} - \hat{\mathbf{G}}_{t-1}\|_2^2$. Since applying the full projection at every step can destabilize the generation process, PRODIGY uses a partial update to balance constraint satisfaction with the learned diffusion trajectory:

$$\mathbf{G}_{t-1} \leftarrow (1 - \gamma_t)\hat{\mathbf{G}}_{t-1} + \gamma_t \Pi_{\mathcal{C}}(\hat{\mathbf{G}}_{t-1})$$

This approach has two main limitations. First, it is only practical for simple constraints where the projection operator $\Pi_{\mathcal{C}}(\cdot)$ has an efficient, closed-form solution. Second, it applies the projection directly to the noisy intermediate sample $\mathbf{G}_{t-1}$, whereas the constraint $\mathcal{C}$ is defined on the clean data $\mathbf{G}_0$, creating a domain mismatch. Recently, in Tenorio et al. (2025), the authors use zeroth-order optimization to build a guidance term, improving over PRODIGY in challenging tasks.

### B.5.2 Flow-based inverse solvers

More recently, two flow-based generative models for graphs have been proposed. Catflow, introduced in Eijkelboom et al. (2024), formulates flow matching as a variational inference problem, allowing to build a model for categorical data. The key difference between Catflow and traditional flow matching is that in the former, the objective is to approximate the posterior probability path, which is a distribution over possible end points of a trajectory. Compared to discrete diffusion, this formulation defines a path in the probability simplex, building a continuous path. This formulation boils down to a cross-entropy loss. Another recent work is DeFoG, introduced in Qin et al. (2025). This method is inspired by discrete flow matching (Campbell et al., 2024), where a discrete probability path is used. Similarly, the loss is a cross-entropy.

## C Proofs

### C.1 Proof of Theorem 1 and Proposition 1

**Proposition 1** (Permutation invariance of the reconstruction density). *In addition to (a)–(b), assume:*

*(b') $v_\theta$ is continuously differentiable in its first argument (a sharpening of (b) needed for the change-of-variables Jacobian);*

*(c) $f_{prior}$ and $v_\theta(\cdot, t)$ map symmetric, zero-diagonal matrices to symmetric, zero-diagonal matrices, so the dynamics stay in the space of undirected graphs;*

*(d) the source noise is non-degenerate, $\sigma_s > 0$.*

*The mask $\xi \in \{0, 1\}^{N \times N}$ is symmetric with zero diagonal (it selects unordered edges), which is what lets $w_\theta = (1 - \xi) \odot v_\theta$ preserve the symmetric, zero-diagonal subspace. We identify an undirected graph with its edge vector over the unordered pairs $\{i, j\}$, $i < j$; the masked edge coordinates are the pairs with $\xi_{ij} = 0$, of which there are $m$. Define the* projected *velocity field*

$$w_\theta(\mathbf{A}, t; \xi) = (1 - \xi) \odot v_\theta(\mathbf{A}, t),$$

which vanishes on observed coordinates by construction and is the idealized object the trained MSE field approximates. Define

$$\mathbf{A}_0 = \mathbf{A}^{\mathcal{O}} + (1 - \xi) \odot \big(f_{prior}(\mathbf{A}^{\mathcal{O}}) + \boldsymbol{\epsilon}_s\big), \quad \boldsymbol{\epsilon}_s \sim \mathcal{N}(0, \sigma_s^2 I_m) \text{ on masked edges,}$$

$$\mathbf{A}_t \text{ solves the projected flow } \dot{\mathbf{A}}_t = w_\theta(\mathbf{A}_t, t; \xi), \ \mathbf{A}_t|_{t=0} = \mathbf{A}_0 \text{ (assumed to exist on } [0, 1]), \qquad \tilde{\mathbf{A}}_1 = \mathbf{A}_{t=1}.$$

Then the conditional density of $\tilde{\mathbf{A}}_1$ on the masked edge coordinates, given by the CNF change-of-variables formula

$$\log p(\tilde{\mathbf{A}}_1 \mid \mathbf{A}^{\mathcal{O}}, \xi) = \log p(\mathbf{A}_0 \mid \mathbf{A}^{\mathcal{O}}, \xi) - \int_0^1 \mathrm{tr}_m\left(\frac{\partial w_\theta(\mathbf{A}_t, t; \xi)}{\partial \mathbf{A}_t}\right) dt, \tag{12}$$

with $\mathrm{tr}_m$ the trace restricted to masked edge coordinates, is invariant under simultaneous relabeling: for every permutation matrix $\mathbf{P}$,

$$\log p(\mathbf{P}^\top \tilde{\mathbf{A}}_1 \mathbf{P} \mid \mathbf{P}^\top \mathbf{A}^{\mathcal{O}} \mathbf{P}, \mathbf{P}^\top \xi \mathbf{P}) = \log p(\tilde{\mathbf{A}}_1 \mid \mathbf{A}^{\mathcal{O}}, \xi). \tag{13}$$

*Proof.* Throughout, write $\mathbf{A}'^{\mathcal{O}} = \mathbf{P}^\top \mathbf{A}^{\mathcal{O}} \mathbf{P}$, $\xi' = \mathbf{P}^\top \xi \mathbf{P}$, and let

$$\mu(\mathbf{X}, \eta) \ = \ \mathbf{X} + (1 - \eta) \odot f_{\mathrm{prior}}(\mathbf{X})$$

denote the deterministic center of the source state. For the density statement we work in undirected-edge coordinates: a symmetric, zero-diagonal matrix $\mathbf{X}$ is identified with its edge vector $\mathbf{e}(\mathbf{X}) \in \mathbb{R}^{N(N-1)/2}$ over unordered pairs $\{i, j\}$, simultaneous relabeling $\mathbf{X} \mapsto \mathbf{P}^\top \mathbf{X} \mathbf{P}$ acts on $\mathbf{e}(\mathbf{X})$ by an orthogonal permutation matrix $R_\mathbf{P}$ that maps masked edges to masked edges, $S_\xi \in \{0, 1\}^{m \times N(N-1)/2}$ extracts the $m$ masked edge coordinates, and $\mathbf{M}_\xi = S_\xi^\top S_\xi$ is the corresponding diagonal projector. Assumption (c) makes $w_\theta$ tangent to this subspace, so the projected flow remains in edge coordinates. We prove both results together: Steps (i) and (iii), together with the equivariance of the discrete sampler, establish the equivariance of the reconstruction (Theorem 1); Steps (ii) and (iv), combined with the change-of-variables conclusion, establish the density invariance (Proposition 1). The four steps are (i) equivariance of $\mu$; (ii) invariance of the conditional source density under simultaneous relabeling; (iii) equivariance of the ODE flow map; (iv) pointwise invariance of the masked-trace integrand.

**Step (i): equivariance of the deterministic center $\mu$.** Using equivariance of $f_{\mathrm{prior}}$ (assumption (a)) together with the identity $(\mathbf{P}^\top X \mathbf{P}) \odot (\mathbf{P}^\top Y \mathbf{P}) = \mathbf{P}^\top (X \odot Y) \mathbf{P}$:

$$
\begin{aligned}
\mu(\mathbf{A}'^{\mathcal{O}}, \xi') &= \mathbf{A}'^{\mathcal{O}} + (1 - \xi') \odot f_{\mathrm{prior}}(\mathbf{A}'^{\mathcal{O}}) \\
&= \mathbf{P}^\top \mathbf{A}^{\mathcal{O}} \mathbf{P} + (1 - \mathbf{P}^\top \xi \mathbf{P}) \odot f_{\mathrm{prior}}(\mathbf{P}^\top \mathbf{A}^{\mathcal{O}} \mathbf{P}) \\
&= \mathbf{P}^\top \mathbf{A}^{\mathcal{O}} \mathbf{P} + \mathbf{P}^\top (1 - \xi) \mathbf{P} \odot \mathbf{P}^\top f_{\mathrm{prior}}(\mathbf{A}^{\mathcal{O}}) \mathbf{P} \\
&= \mathbf{P}^\top \big[\mathbf{A}^{\mathcal{O}} + (1 - \xi) \odot f_{\mathrm{prior}}(\mathbf{A}^{\mathcal{O}})\big] \mathbf{P} \\
&= \mathbf{P}^\top \mu(\mathbf{A}^{\mathcal{O}}, \xi) \mathbf{P}.
\end{aligned}
$$

**Step (ii): invariance of the conditional source density.** In the masked edge coordinates the source is $\mathcal{N}(\mathbf{e}_\xi(\mu(\mathbf{A}^{\mathcal{O}}, \xi)), \sigma_s^2 I_m)$ on $\mathbb{R}^m$, where $\mathbf{e}_\xi(\mathbf{X}) = S_\xi \mathbf{e}(\mathbf{X}) \in \mathbb{R}^m$ extracts the masked edges (each edge counted once), so that

$$p(\mathbf{A}_0 \mid \mathbf{A}^{\mathcal{O}}, \xi) \ \propto \ \exp\left(-\tfrac{1}{2\sigma_s^2} \big\|\mathbf{e}_\xi\big(\mathbf{A}_0 - \mu(\mathbf{A}^{\mathcal{O}}, \xi)\big)\big\|_2^2\right).$$

Evaluating at $\mathbf{P}^\top \mathbf{A}_0 \mathbf{P}$ under the permuted inputs $(\mathbf{A}'^{\mathcal{O}}, \xi')$ and using Step (i): since relabeling maps masked edges to masked edges, $\mathbf{e}_{\xi'}(\mathbf{P}^\top \mathbf{X} \mathbf{P})$ is a permutation of the $m$ entries of $\mathbf{e}_\xi(\mathbf{X})$, so its $\ell_2$ norm is unchanged,

$$\big\|\mathbf{e}_{\xi'}\big(\mathbf{P}^\top \mathbf{A}_0 \mathbf{P} - \mu(\mathbf{A}'^{\mathcal{O}}, \xi')\big)\big\|_2^2 = \big\|\mathbf{e}_\xi\big(\mathbf{A}_0 - \mu(\mathbf{A}^{\mathcal{O}}, \xi)\big)\big\|_2^2.$$

Hence

$$p(\mathbf{P}^\top \mathbf{A}_0 \mathbf{P} \mid \mathbf{A}'^{\mathcal{O}}, \xi') \ = \ p(\mathbf{A}_0 \mid \mathbf{A}^{\mathcal{O}}, \xi).$$

**Step (iii): equivariance of the ODE flow map.** Let $g_\xi$ be a velocity field that is locally Lipschitz in $\mathbf{A}$ and satisfies the equivariance identity $\mathbf{P}^\top g_\xi(\mathbf{A}, t)\mathbf{P} = g_{\xi'}(\mathbf{P}^\top \mathbf{A}\mathbf{P}, t)$. This covers $g_\xi = v_\theta$ (the implemented continuous sampler, which does not depend on $\xi$, so $g_{\xi'} = v_\theta$) and $g_\xi = w_\theta(\cdot, \cdot; \xi)$ (the projected flow of Proposition 1, for which the identity follows from equivariance of $v_\theta$ and $1 - \xi' = \mathbf{P}^\top(1 - \xi)\mathbf{P}$). Let $\Phi_t^\xi(\mathbf{A}_0)$ solve $\dot{\mathbf{A}}_t = g_\xi(\mathbf{A}_t, t)$ from $\mathbf{A}_0$, and set $\widetilde{\mathbf{A}}_t := \mathbf{P}^\top \Phi_t^\xi(\mathbf{A}_0)\mathbf{P}$. Differentiating in $t$ and using the equivariance identity,

$$\dot{\tilde{\mathbf{A}}}_t \;=\; \mathbf{P}^\top g_\xi(\Phi_t^\xi(\mathbf{A}_0), t)\mathbf{P} \;=\; g_{\xi'}(\mathbf{P}^\top \Phi_t^\xi(\mathbf{A}_0)\mathbf{P}, t) \;=\; g_{\xi'}(\tilde{\mathbf{A}}_t, t),$$

with initial condition $\tilde{\mathbf{A}}_0 = \mathbf{P}^\top \mathbf{A}_0 \mathbf{P}$. By Picard-Lindelöf (assumption (b)), the $\xi'$-trajectory starting at $\mathbf{P}^\top \mathbf{A}_0 \mathbf{P}$ is exactly $\tilde{\mathbf{A}}_t$, so

$$\Phi_t^{\xi'}(\mathbf{P}^\top \mathbf{A}_0 \mathbf{P}) \;=\; \mathbf{P}^\top \Phi_t^\xi(\mathbf{A}_0)\mathbf{P} \qquad \text{for all } t \in [0, 1].$$

In particular, the inverse flow satisfies $(\Phi_1^{\xi'})^{-1}(\mathbf{P}^\top \tilde{\mathbf{A}}_1 \mathbf{P}) = \mathbf{P}^\top \mathbf{A}_0 \mathbf{P}$.

**Equivariance of the reconstruction (Theorem 1).** Step (i) gives equivariance of the deterministic center $\mu$; since $\boldsymbol{\varepsilon}_s$ is isotropic Gaussian, $\mathbf{P}^\top \boldsymbol{\varepsilon}_s \mathbf{P} \overset{d}{=} \boldsymbol{\varepsilon}_s$, so the source state obeys $\mathbf{A}_0(\mathbf{A}^{\mathcal{O}}, \xi) \overset{d}{=} \mathbf{P}^\top \mathbf{A}_0(\mathbf{A}^{\mathcal{O}}, \xi)\mathbf{P}$ (pointwise under the coupled draw $\mathbf{P}^\top \boldsymbol{\varepsilon}_s \mathbf{P}$, or when $\sigma_s = 0$). Step (iii) gives equivariance of the continuous flow map. The remaining operations of the discrete sampler in Alg. 1 act entrywise with shared parameters (the Euler increment, the sigmoid $\sigma(\cdot)$, the clip to $[0, 1]$, and the re-imposition $\xi \odot \mathbf{A}^{\mathcal{O}} + (1 - \xi) \odot \widehat{\mathbf{A}}$ of the CE branch), so each commutes with the relabeling $\mathbf{X} \mapsto \mathbf{P}^\top \mathbf{X}\mathbf{P}$. Composing an equivariant source with the equivariant $K$-step sampler yields $\text{PIFM}(\mathbf{A}^{\mathcal{O}}, \xi) \overset{d}{=} \mathbf{P}^\top \text{PIFM}(\mathbf{A}^{\mathcal{O}}, \xi)\mathbf{P}$, proving Theorem 1. This uses only assumptions (a)–(b) and holds for both distortions and every $K$.

**Step (iv): pointwise invariance of the masked-trace integrand.** Work in the edge coordinates of the preamble, on which simultaneous relabeling acts by the orthogonal permutation $R_\mathbf{P}$ (so $\mathbf{e}(\mathbf{A}_t') = R_\mathbf{P}\,\mathbf{e}(\mathbf{A}_t)$). Let $J_\xi(\mathbf{A}_t)$ be the edge-coordinate Jacobian of the projected field $w_\theta(\cdot, t; \xi)$. Equivariance of $w_\theta$ and chain-rule differentiation give

$$J_{\xi'}(\mathbf{A}_t') \;=\; R_\mathbf{P}\, J_\xi(\mathbf{A}_t)\, R_\mathbf{P}^\top.$$

Under simultaneous relabeling the masked-edge set maps to itself, so $\mathbf{M}_{\xi'} = R_\mathbf{P}\, \mathbf{M}_\xi\, R_\mathbf{P}^\top$. By cyclic trace invariance and $R_\mathbf{P}^\top R_\mathbf{P} = \mathbf{I}$,

$$\text{tr}_m\left(\tfrac{\partial w_\theta(\mathbf{A}_t', t; \xi')}{\partial \mathbf{A}_t'}\right) = \text{tr}(\mathbf{M}_{\xi'} J_{\xi'}(\mathbf{A}_t')) = \text{tr}(R_\mathbf{P}\, \mathbf{M}_\xi\, J_\xi(\mathbf{A}_t)\, R_\mathbf{P}^\top) = \text{tr}(\mathbf{M}_\xi J_\xi(\mathbf{A}_t)) = \text{tr}_m\left(\tfrac{\partial w_\theta(\mathbf{A}_t, t; \xi)}{\partial \mathbf{A}_t}\right).$$

The integrand is pointwise invariant, hence so is its integral over $t$.

**Conclusion (density invariance, Proposition 1).** Applying the CNF formula at the permuted endpoint $\mathbf{P}^\top \tilde{\mathbf{A}}_1 \mathbf{P}$, and chaining Steps (ii)–(iv):

$$\begin{aligned}
\log p(\mathbf{P}^\top \tilde{\mathbf{A}}_1 \mathbf{P} \mid \mathbf{A}'^{\mathcal{O}}, \xi') &= \log p\big((\Phi_1^{\xi'})^{-1}(\mathbf{P}^\top \tilde{\mathbf{A}}_1 \mathbf{P}) \,\big|\, \mathbf{A}'^{\mathcal{O}}, \xi'\big) - \int_0^1 \text{tr}_m\left(\tfrac{\partial w_\theta(\mathbf{A}_t', t; \xi')}{\partial \mathbf{A}_t'}\right) dt \\
&\overset{\text{(iii)}}{=} \log p(\mathbf{P}^\top \mathbf{A}_0 \mathbf{P} \mid \mathbf{A}'^{\mathcal{O}}, \xi') - \int_0^1 \text{tr}_m\left(\tfrac{\partial w_\theta(\mathbf{A}_t', t; \xi')}{\partial \mathbf{A}_t'}\right) dt \\
&\overset{\text{(ii)}}{=} \log p(\mathbf{A}_0 \mid \mathbf{A}^{\mathcal{O}}, \xi) - \int_0^1 \text{tr}_m\left(\tfrac{\partial w_\theta(\mathbf{A}_t', t; \xi')}{\partial \mathbf{A}_t'}\right) dt \\
&\overset{\text{(iv)}}{=} \log p(\mathbf{A}_0 \mid \mathbf{A}^{\mathcal{O}}, \xi) - \int_0^1 \text{tr}_m\left(\tfrac{\partial w_\theta(\mathbf{A}_t, t; \xi)}{\partial \mathbf{A}_t}\right) dt \\
&= \log p(\tilde{\mathbf{A}}_1 \mid \mathbf{A}^{\mathcal{O}}, \xi).
\end{aligned}$$

$\square$

**Remark (which priors satisfy assumption (a)).** SIGL, GraphSAGE, VGAE, and NCNC are permutation-equivariant by construction: each is built from equivariant GNN layers, and the edge-prediction head reduces to a symmetric function of node embeddings (Hadamard product, inner product, or common-neighbor pooling). When inference is stochastic (VGAE latent sampling, GraphSAGE neighborhood sampling), this equivariance holds in distribution, under the coupled randomness obtained by relabeling the sampling seeds together with the nodes; it holds pointwise only when inference is fully deterministic (e.g., full-neighborhood aggregation with a deterministic readout and no neighborhood sampling), since a deterministic final readout alone does not remove randomness from the encoder. Node2vec is not strictly equivariant because, unlike a GNN prior, it is not a shared deterministic function of the input graph. Its node embeddings are learned through stochastic optimization with random initialization, sampled walks, and negative sampling. As a result, even two runs on the same graph can produce different embeddings. For Node2vec, Theorem 1 and Proposition 1 should therefore be read as approximate guarantees that hold when the canonicalization is itself approximately invariant; the strictly equivariant priors give the unconditional guarantee.

# D   Experimental details

## D.1   Details about the architecture

Our model adopts a modified version of the adjacency score network architecture introduced in GDSS (Jo et al., 2022). The network is a permutation-equivariant graph neural network designed to approximate the scores $\nabla_{\mathbf{A}_t} \log p_t(X_t, \mathbf{A}_t)$ and $\nabla_{\mathbf{x}_t} \log p_t(\mathbf{x}_t, \mathbf{A}_t)$ at each diffusion step; in this paper, we use only score w.r.t. $\mathbf{A}_t$. Concretely, the architecture consists of stacked message-passing layers followed by a multi-layer perceptron. Each layer propagates node and edge information through adjacency-based aggregation, ensuring equivariance under node relabeling. Time information $t$ is incorporated by scaling intermediate activations with the variance of the forward diffusion process, following the practice in continuous-time score models. Residual connections and normalization layers are used to stabilize training. The final output is an $N \times N$ tensor matching the adjacency dimension. This design provides the required permutation-equivariance and expressive power while remaining computationally tractable for mid-sized benchmark graphs.

The modification that incorporates is a module to build an embedding for the variable $t$ and a FiLM style modulation to incorporate noise conditioning. In particular, we incorporate the following modules:

- A positional encoding based on a sinusoidal embedding following Karras et al. (2022)

- An MLP layer with SiLU activation per attention layer

- A modulation at each attention layer, where we scale the hidden features by an adaptive RMS norm operation (Crowson et al., 2024)

## D.2   Details about the datasets

In Table 5 we report the statistics of the datasets used in the main text.

Table 5: Statistics of the datasets used for evaluation.

| Dataset | # Graphs | Avg. Nodes | Avg. Edges | # Classes | Domain |
|---|---|---|---|---|---|
| ENZYMES | 600 | 32.63 | 62.14 | 6 | Bioinformatics |
| PROTEINS | 1,113 | 39.06 | 72.82 | 2 | Bioinformatics |
| IMDB-B | 1,000 | 19.77 | 96.53 | 2 | Social Network |

## D.3   Masking protocol and mask composition

For ENZYMES, PROTEINS, and IMDB-B, link-prediction masks are sampled over all unordered node pairs in the upper triangle. Each pair is masked independently of its adjacency value, so both positive edges

and negative pairs occur in the observed and masked regions. Table 6 reports aggregate counts over the stored training, validation, and test graphs. The resulting protocol is missing-at-random adjacency-matrix completion rather than positives-only edge removal; expansion instead hides a fraction of observed positive edges.

Table 6: Composition of the stored TU all-pairs masks, aggregated over the training, validation, and test graphs. "Positive" and "negative" refer to the corresponding ground-truth adjacency entry.

| Dataset | Mask rate | Observed positive | Observed negative | Masked positive | Masked negative |
|---|---|---|---|---|---|
| ENZYMES | 10% | 33,471 | 301,642 | 3,671 | 33,547 |
| ENZYMES | 50% | 18,695 | 167,928 | 18,447 | 167,261 |
| IMDB-B | 10% | 86,946 | 125,615 | 9,585 | 14,008 |
| IMDB-B | 50% | 48,228 | 69,440 | 48,303 | 70,183 |
| PROTEINS | 10% | 73,052 | 1,719,848 | 7,992 | 191,525 |
| PROTEINS | 50% | 40,279 | 955,342 | 40,765 | 956,031 |

### D.4 Training-time refinements of the flow model

The reported link-prediction and expansion results place explicit training weight at $t = 0$. We use two ways of doing so. Both address the same mismatch, both act only at training time, and neither changes the sampler: inference is the Euler procedure of Alg. 1 in every case. Because the two are used for different tables, we state here what they are, which configuration produced which result, and how each compares against training with $t \sim U[0,1]$ alone.

**The $t = 0$ mismatch.** Training draws $t \sim U[0,1]$, whereas the sampler first evaluates the velocity field at $t = 0$; for $K = 1$ it evaluates the model *only* at $t = 0$. A continuous uniform law never draws that exact point, so uniform-in-$t$ training places no explicit weight on the single input the $K = 1$ sampler uses. The two refinements below each supply that weight, in different ways.

**Refinement A: endpoint anchor (used for the main MSE results).** Time sampling is left uniform and an extra loss term is added, supervising the velocity at $t = 0$ against the residual it must predict:

$$\mathcal{L}_{\text{total}} = \mathcal{L}_{\text{FM}} + \lambda_0 \, \mathbb{E}[\text{MSE}_{\mathcal{U}}(v_\theta(\mathbf{A}_0, 0), \, \mathbf{A}_1 - \mathbf{A}_0)], \qquad \lambda_0 = 5 \tag{14}$$

where $\mathcal{L}_{\text{FM}}$ is the flow-matching objective of Eq. (2) and $\mathcal{U}$ is the supervised upper-triangular region (the masked node pairs for all-pairs completion; the unobserved region for expansion). The loss is averaged over selected pairs within each graph and then over the batch. Setting $\lambda_0 = 0$ recovers uniform-in-$t$ training.

**Refinement B: boundary-aware time sampling (used for the main CE results).** The loss is left unchanged and the way of sampling time is modified: with probability $p$ the training time is set to $t = 0$ exactly, and otherwise $t \sim U[0,1]$. We use $p = 0.25$. Equivalently, $t$ is drawn from the mixture $p \, \delta_0 + (1 - p) \, U[0,1]$; setting $p = 0$ likewise recovers uniform-in-$t$ training.

Table 7: Configuration used for each set of reported results. "Uniform-$t$" denotes training with $t \sim U[0,1]$ and no endpoint anchor, i.e. $\lambda_0 = 0$ and $p = 0$; A and B are the two ways of placing weight at $t = 0$ defined above. All three are training-time settings only; the sampler is identical throughout.

| Results | Task | Parameterization | Time law | Endpoint anchor | Configuration |
|---|---|---|---|---|---|
| Table 2, PIFM (MSE) | Link prediction | Velocity | $U[0,1]$ | $\lambda_0 = 5$ | A |
| Table 2, PIFM (CE) | Link prediction | Endpoint (BCE) | $p\delta_0 + (1-p)U[0,1], \, p = 0.25$ | — | B |
| Table 4 | Expansion | Velocity | $U[0,1]$ | $\lambda_0 = 5$ | A |
| Table 3 | Cora subgraph LP | Velocity / Endpoint | $U[0,1]$ | — | Uniform-$t$ |
| Appendix E.12 | Scope extensions | varies | $U[0,1]$ | — | Uniform-$t$ |

Two consequences of Table 7 deserve to be stated plainly. First, the MSE and CE blocks of Table 2 differ in the distortion *and* in the parameterization *and* in the time law; they are two instantiations of PIFM, not a controlled ablation of the loss alone, and we do not interpret their difference as a pure loss effect. Second, the endpoint anchor of Eq. (14) is an endpoint-MSE term and is not defined for the BCE parameterization, so Refinement A cannot be applied to the CE results and the ablation below is reported for the MSE instantiation only.

Table 8: Effect of the two refinements on link prediction with 50% of node pairs masked, $K = 1$. "Uniform-$t$" trains with $t \sim U[0, 1]$ and no endpoint anchor ($\lambda_0 = 0$, $p = 0$); A adds the endpoint anchor ($\lambda_0 = 5$); B replaces the time law with $p\,\delta_0 + (1 - p)\,U[0, 1]$, $p = 0.25$. The three arms share the same splits, masks, Stage-1 prior, architecture, optimizer, epochs, and source noise, and use an identical sampler; they differ only in where training weight is placed on $t$. Five-seed means $\pm$ sample standard deviations in percent. Column A reproduces the PIFM (MSE) block of Table 2.

| | AUC ↑ | | | AP ↑ | | |
|---|---|---|---|---|---|---|
| Stage-1 prior | Uniform-$t$ | A (anchor) | B (time law) | Uniform-$t$ | A (anchor) | B (time law) |
| *ENZYMES* | | | | | | |
| Node2Vec | $55.57_{\pm0.61}$ | $\mathbf{62.48}_{\pm1.25}$ | $59.46_{\pm2.49}$ | $19.77_{\pm0.99}$ | $\mathbf{25.29}_{\pm2.65}$ | $23.25_{\pm2.71}$ |
| GraphSAGE | $58.38_{\pm1.36}$ | $\mathbf{62.55}_{\pm0.88}$ | $60.96_{\pm1.73}$ | $23.74_{\pm2.83}$ | $\mathbf{27.14}_{\pm2.61}$ | $25.92_{\pm3.35}$ |
| VGAE | $65.96_{\pm0.55}$ | $\mathbf{68.57}_{\pm2.26}$ | $68.18_{\pm1.72}$ | $28.50_{\pm1.90}$ | $\mathbf{30.90}_{\pm3.15}$ | $30.61_{\pm2.95}$ |
| NCNC | $65.85_{\pm2.72}$ | $\mathbf{67.07}_{\pm3.01}$ | $65.78_{\pm5.48}$ | $31.21_{\pm2.90}$ | $\mathbf{34.38}_{\pm4.43}$ | $32.14_{\pm3.78}$ |
| LPFormer | $66.78_{\pm1.48}$ | $\mathbf{73.16}_{\pm1.39}$ | $72.30_{\pm1.00}$ | $29.22_{\pm1.06}$ | $\mathbf{35.99}_{\pm3.01}$ | $35.04_{\pm2.49}$ |
| *PROTEINS* | | | | | | |
| Node2Vec | $59.80_{\pm2.17}$ | $\mathbf{64.96}_{\pm1.33}$ | $62.32_{\pm2.11}$ | $31.11_{\pm2.65}$ | $\mathbf{34.88}_{\pm2.79}$ | $33.33_{\pm2.98}$ |
| GraphSAGE | $62.63_{\pm1.53}$ | $64.91_{\pm2.14}$ | $\mathbf{65.19}_{\pm2.00}$ | $34.11_{\pm3.19}$ | $35.61_{\pm3.30}$ | $\mathbf{35.85}_{\pm3.24}$ |
| VGAE | $66.74_{\pm1.26}$ | $\mathbf{69.22}_{\pm1.13}$ | $68.19_{\pm1.36}$ | $37.06_{\pm3.11}$ | $\mathbf{38.09}_{\pm2.65}$ | $38.00_{\pm2.57}$ |
| NCNC | $67.33_{\pm2.94}$ | $68.00_{\pm3.51}$ | $\mathbf{70.33}_{\pm3.55}$ | $39.43_{\pm1.72}$ | $41.52_{\pm4.41}$ | $\mathbf{42.26}_{\pm2.19}$ |
| LPFormer | $67.20_{\pm2.49}$ | $\mathbf{74.12}_{\pm0.98}$ | $72.99_{\pm1.06}$ | $37.05_{\pm4.22}$ | $\mathbf{43.40}_{\pm3.05}$ | $42.68_{\pm2.43}$ |
| *IMDB-B* | | | | | | |
| Node2Vec | $82.14_{\pm1.75}$ | $84.75_{\pm1.20}$ | $\mathbf{84.98}_{\pm1.67}$ | $82.79_{\pm1.26}$ | $85.77_{\pm1.48}$ | $\mathbf{85.81}_{\pm1.36}$ |
| GraphSAGE | $90.69_{\pm2.01}$ | $\mathbf{93.46}_{\pm1.11}$ | $91.88_{\pm1.51}$ | $90.47_{\pm1.37}$ | $\mathbf{92.95}_{\pm1.15}$ | $91.62_{\pm0.52}$ |
| VGAE | $90.74_{\pm1.10}$ | $\mathbf{92.46}_{\pm1.37}$ | $92.12_{\pm1.18}$ | $91.00_{\pm2.08}$ | $\mathbf{92.52}_{\pm2.02}$ | $92.25_{\pm1.96}$ |
| NCNC | $86.11_{\pm3.50}$ | $\mathbf{91.07}_{\pm2.00}$ | $90.18_{\pm2.28}$ | $87.70_{\pm2.26}$ | $\mathbf{92.92}_{\pm0.52}$ | $91.98_{\pm0.76}$ |
| LPFormer | $93.73_{\pm0.80}$ | $\mathbf{94.99}_{\pm0.48}$ | $94.98_{\pm0.49}$ | $93.79_{\pm1.65}$ | $\mathbf{95.28}_{\pm1.11}$ | $95.06_{\pm1.31}$ |

**Conclusion for this ablation.** Placing explicit weight on $t = 0$ helps, and the effect is consistent: the anchor improves mean AUC over uniform-in-$t$ training in all 15 dataset–prior cells, and boundary-aware time sampling does so in 14 of 15. The largest gains occur on the sparse datasets with the Node2Vec and LPFormer priors; the smallest are the NCNC cells (PROTEINS $67.33 \rightarrow 68.00$, ENZYMES $65.85 \rightarrow 67.07$), where the five-seed spread exceeds the effect, and IMDB-B with LPFormer ($93.73 \rightarrow 94.99$), where the flow is already near ceiling.

We do *not* claim that the anchor is better than the other refinement. With only five seeds and 15 cells, we do not draw significance conclusions for individual cells, especially without correcting for multiple comparisons. The more reliable pattern is the consistent improvement of both weighting schemes over uniform-in-$t$ training across nearly all cells.

### D.5 Hyperparameters

### D.5.1 Flow-based baselines

We report the hyperparameters of our model, both the architecture and training. All three baselines use the same rectified-flow architecture and optimizer family; the only substantive differences are the prior settings.

| PIFM - Link Prediction, 10% masked | |
|---|---|
| batch_size | 64 (IMDB-B & ENZYMES), 32 (PROTEINS) |
| optimizer | Adam |
| learning_rate | 2e-4 |
| dropout | 0.2 |
| hidden_dim | 32 |
| num_layers | 5 |
| num_linears | 2 |
| channels | {c_init: 2, c_hid: 8, c_final: 4} |
| train/val/test_noise_std | 0.05 (IMDB-B & PROTEINS), 0.1 (ENZYMES) |
| ode_steps (Euler, $K$) | 1 to 100 |
| prior | GraphSAGE (default hyperparameters) |

| PIFM - Link Prediction, 50% masked | |
|---|---|
| batch_size | 64 (IMDB-B & ENZYMES), 32 (PROTEINS) |
| optimizer | Adam |
| learning_rate | 2e-4 |
| dropout | 0.2 |
| hidden_dim | 32 |
| num_layers | 5 |
| num_linears | 2 |
| channels | {c_init: 2, c_hid: 8, c_final: 4} |
| train/val/test_noise_std | 0.05 (IMDB-B & PROTEINS), 0.1 (ENZYMES) |
| ode_steps (Euler, $K$) | 1 to 100 |
| prior | GraphSAGE (default hyperparameters) |

| PIFM (VGAE) - Link Prediction, 50% masked | |
|---|---|
| batch_size | 64 (IMDB-B & ENZYMES), 32 (PROTEINS) |
| optimizer | Adam |
| learning_rate | 2e-4 |
| dropout | 0.2 |
| hidden_dim | 32 |
| num_layers | 5 |
| num_linears | 2 |
| channels | {c_init: 2, c_hid: 8, c_final: 4} |
| train/val/test_noise_std | 0.01 (IMDB-B), 0.00 (PROTEINS, ENZYMES) |
| ode_steps (Euler, $K$) | 1 to 100 |
| prior | VGAE (epochs: 200, lr: 0.01, hidden: 32/16) |

**PIFM (NCNC) - Link Prediction, 50% masked**

| | |
|---|---|
| `batch_size` | 64 (IMDB-B & ENZYMES), 32 (PROTEINS) |
| `optimizer` | Adam |
| `learning_rate` | 2e-4 |
| `dropout` | 0.2 |
| `hidden_dim` | 32 |
| `num_layers` | 5 |
| `num_linears` | 2 |
| `channels` | {c_init: 2, c_hid: 8, c_final: 4} |
| `train/val/test_noise_std` | 0.05 (IMDB-B), 0.01 (ENZYMES, PROTEINS) |
| `ode_steps` (Euler, $K$) | 1 to 100 |
| `prior` | NCNC (default hyperparameters) |

**PIFM (CE) - Link Prediction, 50% masked**

| | |
|---|---|
| `epochs` | 300 |
| `batch_size` | 64 (IMDB-B & ENZYMES), 32 (PROTEINS) |
| `optimizer` | Adam |
| `learning_rate` | 2e-4 |
| architecture | as above (endpoint head; logits, no sigmoid at train time) |
| `train/val/sample_noise_std` ($\sigma_s$) | 0.00 (all datasets) |
| positive-class weight $w$ | $\min(n_-/n_+, 50)$, recomputed per batch |
| `ce_eps` ($\varepsilon$ in Eq. (9)) | $10^{-5}$ |
| time law | $p\,\delta_0 + (1-p)\,U[0,1]$, $p = 0.25$ (Appendix D.4) |
| `ode_steps` (Euler, $K$) | 1 to 100 |

| Flow w/ Gaussian Prior | Value |
|---|---|
| `epochs` | 1000 |
| `batch_size` | 64 (default), 32 (PROTEINS) |
| `optimizer` | Adam |
| `learning_rate` | 2e-4 |
| `dropout` | 0.2 |
| `hidden_dim` | 32 |
| `num_layers` | 5 |
| `num_linears` | 2 |
| `channels` | {c_init: 2, c_hid: 8, c_final: 4} |
| `train_noise_std` (masked $t=0$) | 0.00 |
| `val_noise_std` (masked $t=0$) | 0.00 |
| `ode_steps` (Euler, $K$) | 1 to 100 |
| `prior` | None (masked entries initialized from $\mathcal{N}(0.5, 1)$) |

### D.6   Metrics Calculation

We evaluate performance only on masked unordered node pairs in the upper triangle of the adjacency matrix. For each test graph, all metrics are computed on these entries and then aggregated as specified below.

The primary reconstruction tables use a fixed threshold of 0.5. In the additional link-prediction operating-point analysis, each seed selects

$$\tau^{\star}_{\mathrm{val}} = \arg\max_{\tau} \mathrm{MCC}(\widehat{\mathbf{y}}_{\mathrm{val}}(\tau), \mathbf{y}_{\mathrm{val}}) \tag{15}$$

using validation predictions and labels only, and freezes this threshold before test evaluation. Precision, recall, F1, MCC, balanced accuracy, and normalized Hamming error are pooled over masked upper-triangular pairs. Exact recovery requires every masked pair in a test graph to be correct, whereas near-exact recovery permits

at most 5% masked-pair errors. Absolute normalized edge-count error is $|\#\widehat{E} - \#E|/\#E$ on the pooled masked entries; AUC and AP use raw scores and are independent of the selected threshold.

**Metrics Used in Tables**   The primary reconstruction tables report AUC, AP, FPR, and FNR; distributional experiments additionally report MMD.

- **Area Under the ROC Curve (AUC).** Computed on the raw predicted scores (when available). AUC measures the probability that a randomly chosen true edge receives a higher predicted score than a randomly chosen non-edge. Larger AUC indicates stronger ranking performance.

- **Average Precision (AP).** Computed from the precision–recall curve induced by ranking the predictions. AP summarizes how well the model recovers true edges across all possible thresholds, with higher values indicating better precision–recall trade-offs.

- **False Positive Rate (FPR).** After thresholding predictions at 0.5, the FPR is defined as

$$\text{FPR} = \frac{\text{FP}}{\text{FP} + \text{TN}},$$

- **False Negative Rate (FNR).** After thresholding predictions at 0.5, the FNR is defined as

$$\text{FNR} = \frac{\text{FN}}{\text{FN} + \text{TP}},$$

- **MMD.** A kernel-based method that measures the difference between two probability distributions by embedding them in a feature space and finding the maximum difference between their mean embeddings.

AUC and AP are threshold-independent metrics (computed directly on the provided scores), while FPR and FNR are threshold-dependent error rates (obtained after binarizing at 0.5).

The primary AUC/AP/FPR/FNR benchmark tables express values in percent. The validation-selected tables in Appendix E.1 use proportions unless a percentage sign is shown, and MMD is reported on its native scale.

## D.7 Training cost

All models are trained on NVIDIA A100-SXM4-80GB GPUs. Once the prior is fixed, the flow model used in PIFM has essentially the same training loop and computational cost as the flow model initialized with a Gaussian prior. Thus, the main additional cost of PIFM comes from fitting the prior itself, not from the subsequent flow-matching stage. This observation is consistent with our empirical training runs, where we did not observe noticeable differences in optimization behavior between Gaussian-prior and prior-informed flow variants across edge-drop settings. Table 9 reports training time of the flow model for 100 epochs on each dataset, with PROTEINS taking the most time while the other 2 datasets remain similar training costs.

Table 9: Training time of the flow model for 100 epochs on each dataset under 50% mask rate.

| Dataset | Training Time (s) |
|---------|-------------------|
| ENZYMES | 206.4 |
| PROTEINS | 646.7 |
| IMDB-B | 243.7 |

## D.8 Inference cost

Since PIFM reconstructs graphs by integrating a learned velocity field over $K$ Euler steps, its runtime grows approximately linearly with the number of Euler steps integrated. We observe a cost of about 0.03 seconds

per step per graph. Increasing $K$ therefore raises latency proportionally; its effect on reconstruction quality is empirical and dataset-dependent.

Table 10 reports the per-graph inference time of PIFM as a function of the number of Euler steps. The results show near-linear scaling, from approximately 0.03 seconds in the lowest-step setting to about 2.8 seconds when using 100 steps.

Table 10: Per-graph inference time of PIFM as a function of the number of Euler steps.

| Setting | Inference Time per Graph (s) |
|---|---|
| $K = 1$ | 0.03 |
| $K = 10$ | 0.28 |
| $K = 50$ | 1.40 |
| $K = 100$ | 2.80 |

To contextualize these costs, Table 11 compares PIFM against the standalone prior predictors on each dataset. As expected, the one-shot priors are faster because they do not require iterative refinement.

PIFM with $K = 1$ remains below 0.06 seconds per graph on all three benchmarks and is the distortion-oriented setting used in the primary link-prediction results. Larger $K$ should be chosen only when a validated distributional benefit justifies the added latency.

Table 11: Per-graph inference time comparison between standalone priors and PIFM.

| Method | PROTEINS (s) | ENZYMES (s) | IMDB-B (s) |
|---|---|---|---|
| Node2Vec Prior | 0.0016 | 0.0020 | 0.0010 |
| GraphSAGE Prior | 0.0003 | 0.0003 | 0.0002 |
| PIFM ($K = 1$) | 0.0425 | 0.0532 | 0.0440 |
| PIFM ($K = 100$) | 2.8207 | 2.8419 | 2.8660 |

# E  Additional results

## E.1  Validation-selected binary reconstruction

We evaluate endpoint-anchored PIFM with LPFormer, MSE, $K = 1$, and 50% all-pairs masking. For each of five seeds, the threshold is selected by validation MCC and then frozen for the test set. The analysis is limited to all-pairs link prediction and does not include expansion or denoising. The LPFormer rows evaluate the clean Stage-1 predictions before the source-noise perturbation used to initialize the flow.

Table 12: Test performance at the validation-MCC threshold. Values are means ± sample standard deviations over five seeds. Pairwise metrics pool masked upper-triangular entries; exact recovery is measured per graph.

| Dataset | Method | $\tau^{\star}_{\text{val}}$ | Precision | Recall | F1 | MCC | Exact recovery (%) |
|---|---|---|---|---|---|---|---|
| ENZYMES | LPFormer prior | $0.568 \pm 0.217$ | $0.273 \pm 0.060$ | $0.313 \pm 0.030$ | $0.289 \pm 0.033$ | $0.208 \pm 0.031$ | $0.0 \pm 0.0$ |
| ENZYMES | Endpoint-anchored PIFM | $0.184 \pm 0.044$ | $\mathbf{0.285} \pm 0.060$ | $\mathbf{0.454} \pm 0.079$ | $\mathbf{0.341} \pm 0.026$ | $\mathbf{0.266} \pm 0.016$ | $0.0 \pm 0.0$ |
| PROTEINS | LPFormer prior | $0.458 \pm 0.246$ | $0.209 \pm 0.041$ | $0.358 \pm 0.055$ | $0.261 \pm 0.041$ | $0.229 \pm 0.035$ | $1.4 \pm 1.5$ |
| PROTEINS | Endpoint-anchored PIFM | $0.164 \pm 0.037$ | $\mathbf{0.261} \pm 0.077$ | $\mathbf{0.450} \pm 0.084$ | $\mathbf{0.317} \pm 0.063$ | $\mathbf{0.296} \pm 0.042$ | $\mathbf{2.9} \pm 1.0$ |
| IMDB-B | LPFormer prior | $0.681 \pm 0.133$ | $0.947 \pm 0.023$ | $0.809 \pm 0.037$ | $0.872 \pm 0.025$ | $0.784 \pm 0.028$ | $20.8 \pm 3.9$ |
| IMDB-B | Endpoint-anchored PIFM | $0.513 \pm 0.064$ | $\mathbf{0.950} \pm 0.019$ | $\mathbf{0.909} \pm 0.020$ | $\mathbf{0.929} \pm 0.017$ | $\mathbf{0.870} \pm 0.025$ | $\mathbf{38.0} \pm 5.1$ |

At $\tau^{\star}_{\text{val}}$, endpoint-anchored PIFM improves recall, F1, and MCC over LPFormer on every dataset. Exact recovery remains zero on ENZYMES, showing that improved edge-level reconstruction does not imply perfect graph recovery.

Table 13: Additional test metrics at the same frozen validation-selected thresholds. Near-exact recovery permits at most 5% masked-pair errors.

| Dataset | Method | Balanced accuracy | Normalized Hamming | Absolute normalized edge-count error | Near-exact recovery (%) |
|---|---|---|---|---|---|
| ENZYMES | LPFormer prior | $0.610 \pm 0.008$ | $\mathbf{0.151} \pm 0.029$ | $\mathbf{0.263} \pm 0.123$ | $0.6 \pm 1.4$ |
| ENZYMES | Endpoint-anchored PIFM | $\mathbf{0.664} \pm 0.025$ | $0.168 \pm 0.026$ | $0.691 \pm 0.596$ | $\mathbf{1.9} \pm 2.9$ |
| PROTEINS | LPFormer prior | $0.649 \pm 0.030$ | $0.085 \pm 0.036$ | $\mathbf{0.780} \pm 0.528$ | $4.3 \pm 3.2$ |
| PROTEINS | Endpoint-anchored PIFM | $\mathbf{0.695} \pm 0.036$ | $\mathbf{0.081} \pm 0.037$ | $1.001 \pm 1.029$ | $\mathbf{7.1} \pm 2.2$ |
| IMDB-B | LPFormer prior | $0.885 \pm 0.018$ | $0.110 \pm 0.015$ | $0.146 \pm 0.041$ | $29.6 \pm 6.5$ |
| IMDB-B | Endpoint-anchored PIFM | $\mathbf{0.934} \pm 0.013$ | $\mathbf{0.064} \pm 0.012$ | $\mathbf{0.044} \pm 0.018$ | $\mathbf{58.0} \pm 6.3$ |

The MCC-selected point does not improve every calibration metric. On sparse datasets, lowering the threshold trades false negatives for false positives, so Hamming or edge-count error can worsen even when recall, F1, MCC, and balanced accuracy improve.

Table 14: Threshold-free context for the same LPFormer/MSE/50%-mask, five-seed study. AUC and AP use raw test scores and are macro graph-wise means over graphs whose masked region contains both classes.

| Dataset | Method | AUC | AP |
|---|---|---|---|
| ENZYMES | LPFormer prior | $0.6562 \pm 0.0120$ | $0.323 \pm 0.029$ |
| ENZYMES | Endpoint-anchored PIFM | $\mathbf{0.7316} \pm 0.0139$ | $\mathbf{0.360} \pm 0.030$ |
| PROTEINS | LPFormer prior | $0.6767 \pm 0.0167$ | $0.396 \pm 0.041$ |
| PROTEINS | Endpoint-anchored PIFM | $\mathbf{0.7412} \pm 0.0098$ | $\mathbf{0.419} \pm 0.030$ |
| IMDB-B | LPFormer prior | $0.8979 \pm 0.0052$ | $0.8960 \pm 0.0144$ |
| IMDB-B | Endpoint-anchored PIFM | $\mathbf{0.9499} \pm 0.0048$ | $\mathbf{0.944} \pm 0.013$ |

Without a threshold, endpoint-anchored PIFM improves on the LPFormer prior in mean AUC and mean AP on all three datasets, so the gains in Table 12 are not an artifact of the validation-selected operating point. The AUC gains are large relative to the seed spread ($0.6562 \rightarrow 0.7316$ on ENZYMES, $0.6767 \rightarrow 0.7412$ on PROTEINS, $0.8979 \rightarrow 0.9499$ on IMDB-B); the AP gains on the sparse datasets are not, since the one-standard-deviation bands overlap and absolute AP stays at or below 0.419 for either method.

### E.2 Graph-level reconstruction analysis

ROC-AUC and AP measure held-out-edge ranking independently of a threshold. We also evaluate thresholded recovery and structural properties of the completed graphs. The analysis uses LPFormer and endpoint-anchored MSE PIFM (Model 1) with $K = 1$, 50% all-pairs masking, and five seeds; thresholds maximize validation MCC and are then fixed for test evaluation. Tables 12 and 13 report exact recovery, near-exact recovery, normalized Hamming error, and edge-count error, while Table 15 adds paired triangle, degree, and clustering errors on the completed test graphs.

Table 15: Paired graph-statistic errors at validation-selected thresholds. We first average over test graphs, then report mean $\pm$ sample standard deviation over five seeds. Lower is better. The paired $p$-value is a supporting diagnostic; "wins" counts seeds with lower PIFM error.

| Dataset | Statistic | LPFormer error | Model 1 error | Model 1 wins | Paired $p$ |
|---------|-----------|----------------|----------------|---------------|------------|
| ENZYMES | Triangles | $42.671 \pm 16.702$ | $61.335 \pm 39.130$ | 1/5 | .29 |
| ENZYMES | Degree MAE | $1.869 \pm .253$ | $2.409 \pm .802$ | 1/5 | .17 |
| ENZYMES | Clustering | $.144 \pm .026$ | $.146 \pm .018$ | 2/5 | .82 |
| PROTEINS | Triangles | $113.264 \pm 87.935$ | $105.246 \pm 113.805$ | 3/5 | .63 |
| PROTEINS | Degree MAE | $1.944 \pm .227$ | $2.390 \pm .756$ | 1/5 | .14 |
| PROTEINS | Clustering | $.153 \pm .024$ | $.135 \pm .010$ | 4/5 | .13 |
| IMDB-B | Triangles | $30.280 \pm 8.638$ | $17.016 \pm 7.062$ | 5/5 | .001 |
| IMDB-B | Degree MAE | $.963 \pm .092$ | $.496 \pm .066$ | 5/5 | $< .001$ |
| IMDB-B | Clustering | $.095 \pm .015$ | $.044 \pm .005$ | 5/5 | .001 |

On IMDB-B, Model 1 improves all three errors in every seed (5/5), consistent with a clear benefit. On ENZYMES and PROTEINS, however, the paired differences are not significant with five seeds. Given the $N = 5$ minimum detectable effect of Cohen's $d_z \approx 1.68$, these nonsignificant results are underpowered rather than evidence that the methods are equivalent.

The direction of the per-statistic differences is also dataset dependent. On PROTEINS, Model 1 improves clustering error in 4/5 seeds and triangle error in 3/5, but improves degree error in only 1/5. On ENZYMES, all three statistics instead favor the prior. The signed errors explain this pattern: the flow generally increases predicted density. This corrects the under-dense Stage-1 reconstructions on IMDB-B, but can increase error when the Stage-1 reconstruction is already too dense.

We evaluate connectivity and component-count error under a validation-prevalence policy. Validation labels determine the positive-rate budget, which sets the cutoff among the unlabeled test scores; test labels are not used to select the cutoff. This operating point is distinct from the validation-MCC thresholds used in Table 15.

Table 16: Connectivity and paired absolute component-count error, aggregated over five seeds under the validation-prevalence threshold. Connectivity is the fraction of completed test graphs with one component; its ground-truth rate is shown for context.

| Dataset | GT connected | LPFormer connected | Model 1 connected | LPFormer $|\Delta C|$ | Model 1 $|\Delta C|$ |
|---------|--------------|--------------------|--------------------|-----------------------|----------------------|
| ENZYMES | .981 | .026 | **.342** | 5.890 | **3.690** |
| PROTEINS | .950 | .129 | **.486** | 6.911 | **4.600** |
| IMDB-B | 1.000 | .736 | **.916** | .416 | **.136** |

Model 1 increases the connected-reconstruction rate and lowers component-count error on all three datasets. Table 3 further reports clustering, degree, and NSPDK MMD over Cora subgraph pools.

### E.3 Link prediction with 10% of unordered node pairs masked

Table 17: Graph reconstruction performance with **10% of unordered node pairs masked (0.1 Drop)** (see Table 2 for definitions).

| Method | ENZYMES | | | | PROTEINS | | | | IMDB-B | | | |
|---|---|---|---|---|---|---|---|---|---|---|---|---|
| | AP ↑ | AUC ↑ | FNR ↓ | FPR ↓ | AP ↑ | AUC ↑ | FNR ↓ | FPR ↓ | AP ↑ | AUC ↑ | FNR ↓ | FPR ↓ |
| *Baselines* | | | | | | | | | | | | |
| Node2Vec | 24.62 | 59.60 | 51.39 | 37.96 | 33.24 | 64.40 | 48.56 | 35.37 | 65.00 | 56.36 | 50.27 | 41.68 |
| GraphSAGE | 41.27 | 71.34 | **15.49** | 69.77 | 46.36 | 74.58 | **11.00** | 63.50 | 83.55 | 83.26 | 16.42 | 36.89 |
| DiGress + RePaint | 33.39 | 67.86 | 58.92 | 5.19 | 40.34 | 72.39 | 47.82 | 6.00 | 59.25 | 58.63 | 76.44 | 7.68 |
| GDSS + RePaint | 18.35 | 47.04 | 74.31 | 32.19 | 26.96 | 51.39 | 63.07 | 32.09 | 57.89 | 46.11 | 69.75 | 36.17 |
| Flow w/ Gaussian prior | 40.09 | 72.44 | 71.03 | 5.87 | 57.86 | 80.83 | 65.09 | 3.07 | 98.89 | 98.37 | 2.26 | **2.54** |
| *Ours* | | | | | | | | | | | | |
| PIFM (Node2Vec) | 41.67 | 76.86 | 72.09 | 5.11 | **58.25** | **81.74** | 59.37 | 6.34 | 97.60 | 97.28 | **1.37** | 3.77 |
| PIFM (GraphSAGE) | **47.21** | **80.25** | 72.85 | **2.40** | 54.79 | 81.02 | 55.73 | 5.40 | **99.37** | **98.79** | 1.81 | 3.37 |

Mask Rate: 10% (0.1 Drop)

At the easier 10% rate PIFM attains the best AP and AUC on all three datasets, and each PIFM row improves on the Stage-1 predictor it starts from. The advantage over the uninformed flow is smaller and not systematic: the Gaussian-prior flow exceeds PIFM (GraphSAGE) in AP on PROTEINS (57.86 versus 54.79) and PIFM (Node2Vec) on IMDB-B in both metrics. The informed source therefore contributes less when fewer pairs are masked. These entries are single runs, so sub-point differences should not be over-interpreted.

### E.4 PIFM with a graphon-based prior (SIGL)

The priors used in the main text (GraphSAGE, node2vec, VGAE, NCNC) are all embedding-based link predictors. To test whether PIFM is agnostic to the *family* of structural prior, we additionally evaluate a graphon-based prior: Scalable Implicit Graphon Learning (SIGL) (Azizpour et al., 2025), which fits a graphon to the training graphs and predicts masked node pairs from the learned latent node coordinates (background in Appendix B.1). Unlike the main-text priors, SIGL is a generative model of an exchangeable graph family rather than a discriminative edge predictor, making it a useful test on PIFM's prior-agnosticism.

Table 18: PIFM with a graphon-based prior (SIGL) on link prediction, at 10% and 50% unordered-pair masking. AP, AUC, FNR, FPR in % (see Table 2 for definitions). **Bold** marks the better of the two rows per ranking metric (AP, AUC) within each block.

| Method | ENZYMES | | | | PROTEINS | | | | IMDB-B | | | |
|---|---|---|---|---|---|---|---|---|---|---|---|---|
| | AP ↑ | AUC ↑ | FNR ↓ | FPR ↓ | AP ↑ | AUC ↑ | FNR ↓ | FPR ↓ | AP ↑ | AUC ↑ | FNR ↓ | FPR ↓ |
| *Mask rate: 10% (0.1 Drop)* | | | | | | | | | | | | |
| SIGL prior | 18.17 | 48.04 | 69.33 | 25.43 | 26.77 | 48.91 | 100.00 | 0.00 | 58.91 | 50.61 | 88.44 | 16.33 |
| PIFM (SIGL) | **26.93** | **59.48** | 71.33 | 11.33 | **42.21** | **60.76** | 60.75 | 7.48 | **85.60** | **83.21** | 16.37 | 18.41 |
| *Mask rate: 50% (0.5 Drop)* | | | | | | | | | | | | |
| SIGL prior | 16.88 | **49.30** | 72.01 | 27.85 | 22.90 | 52.55 | 100.00 | 0.00 | 50.05 | 45.41 | 87.68 | 18.80 |
| PIFM (SIGL) | **17.08** | 49.15 | 86.06 | 12.28 | **28.38** | **59.58** | 61.20 | 20.38 | **59.83** | **58.11** | 38.90 | 36.76 |

The SIGL prior is itself close to chance on the ranking metrics (48.04 and 49.30 AUC on ENZYMES, 48.91 and 52.55 on PROTEINS), and on PROTEINS it predicts no edge at all at the fixed 0.5 threshold (100.00 FNR, 0.00 FPR). PIFM improves AP over this prior in all six dataset–rate blocks and AUC in five of six, with the largest gain on IMDB-B at 10% masking (50.61 → 83.21 AUC), and it restores a non-degenerate operating point on PROTEINS (FNR 100.00 → 60.75 and 61.20, at an FPR cost of 7.48 and 20.38). The exception is ENZYMES at 50% masking, where PIFM is *worse* than the prior on AUC (49.15 vs. 49.30): when the graphon fit carries essentially no signal, the flow has nothing to amplify and both rows remain at chance. We therefore read this table as evidence that PIFM is not tied to a particular *family* of prior, rather than as a claim that SIGL is a competitive Stage-1 choice.

## E.5 Denoising

Denoising is the complement of expansion, meaning that we seek to remove a set of spurious edges $\mathcal{E}_S$ from $A^{\mathcal{O}}$, such that the edge set of the ground truth is $\mathcal{E} = \mathcal{E}_O \setminus \mathcal{E}_S$. Hence, the initialization becomes $\mathbf{A}_0 = \mathbf{A}^{\mathcal{O}} \odot (f_{prior}(\mathbf{A}^{\mathcal{O}}) + \boldsymbol{\epsilon}_s)$. As in the expansion task, PIFM here is trained with the squared Frobenius distortion (MSE loss). We corrupt the observed graph by flipping 20% of the upper-triangle zero-entries into spurious edges (denoted 0.2 Flip); the results are shown in Table 19. Similarly to expansion, PIFM (GraphSAGE) attains the best AUC/AP on all datasets, again surpassing the GraphSAGE prior and remaining baselines. It *strongly* reduces false positives from the given prior initialization, while FNR is low on dense IMDB-B (2.67) and higher on sparser sets. Overall, PIFM removes spurious edges more reliably while also improving other metrics.

Table 19: Performance for the **denoising** task with **20% of upper-triangle 0-entries flipped (0.2 Flip)**. We report AUC, Average Precision (AP↑), False Positive Rate (FPR↓), and False Negative Rate (FNR↓), all in percent (%). The best result for each metric is in **bold blue** and the second best is green. The metrics are restricted on the upper-triangle 1-region of $A^{\mathcal{O}}$, and compared against $\mathbf{A}_1$ on that region.

| | Flip Rate: 20% (0.2 Flip) | | | | | | | | | | | |
|---|---|---|---|---|---|---|---|---|---|---|---|---|
| | **ENZYMES** | | | | **PROTEINS** | | | | **IMDB-B** | | | |
| Method | AP ↑ | AUC ↑ | FNR ↓ | FPR ↓ | AP ↑ | AUC ↑ | FNR ↓ | FPR ↓ | AP ↑ | AUC ↑ | FNR ↓ | FPR ↓ |
| *Baselines* | | | | | | | | | | | | |
| GraphSAGE | 68.19 | 73.89 | **16.14** | 61.72 | 73.79 | 76.70 | **12.68** | 60.47 | 92.54 | 77.29 | 16.77 | 53.37 |
| DiGress + RePaint | 41.98 | 49.38 | 87.91 | **13.33** | 49.36 | 51.20 | 78.44 | **18.19** | 80.54 | 51.59 | 73.41 | 23.49 |
| GDSS + RePaint | 44.59 | 50.86 | 69.18 | 29.70 | 49.72 | 49.43 | 68.39 | 32.64 | 82.36 | 53.23 | 69.05 | 26.00 |
| Flow w/ Gaussian prior | 49.90 | 54.56 | 52.30 | 38.93 | 57.18 | 58.70 | 62.84 | 24.32 | 96.75 | 94.63 | 3.80 | 12.66 |
| *Ours* | | | | | | | | | | | | |
| PIFM (GraphSAGE) | **69.40** | **77.17** | 45.66 | 18.14 | **77.43** | **81.78** | 32.41 | 20.91 | **98.46** | **96.52** | **2.67** | **12.10** |

## E.6 Loss-calibration analysis on sparse graphs

The MSE and CE instantiations of PIFM (Table 2) show a pronounced calibration difference on the sparser datasets, PROTEINS and ENZYMES. Here we analyze its cause.

Under the MSE distortion, PIFM is trained as a continuous regression toward the conditional mean. On a sparse graph, non-edges vastly outnumber edges, so the loss is dominated by the negative class and the fitted edge scores concentrate near zero (the conditional-mean target of the rare positive class is itself small). True edges are then treated as rare events and underweighted, and at the fixed decision threshold of 0.5 almost all of them are missed, inflating the false-negative rate (FNR). This is a thresholded-calibration effect rather than a ranking failure: the threshold-independent metrics (AP, AUC) remain informative even where FNR is high.

The cross-entropy (CE) instantiation addresses this directly. Inspired by CatFlow (Eijkelboom et al., 2024), which establishes cross-entropy as an effective distortion for flow matching on graphs, the CE distortion is an element-wise binary cross-entropy on the upper triangle of the masked region. In contrast to CatFlow's unweighted categorical objective, we additionally apply a per-batch positive-class weight $w = \min(n_{\text{neg}}/n_{\text{pos}}, 50)$ to counter the class imbalance specific to our conditional, sparse-graph setting. We emphasize that the positive weighting, not the switch from MSE to cross-entropy alone, is what restores calibration: an *unweighted* cross-entropy on these datasets would face the same negative-dominated gradient and leave the fitted probabilities below the 0.5 threshold, so the thresholded predictions again collapse to no edges, as with MSE. The weighting rebalances the per-edge contributions so that the rare positive class is no longer ignored, sharply reducing FNR (Table 2) at a substantial cost in FPR (which can rise from near zero to 45–89% on the sparsest datasets), while AP/AUC stay comparable. The clamp at 50 caps the weight on the very sparsest graphs, where an unbounded ratio would destabilize training.

We further compare the paper's batch-scalar weight with unweighted, per-graph, and focal objectives at 50% masking. The batch rule uses $w = \min(n_-/n_+, 50)$, the per-graph rule computes this ratio separately for

each graph, and focal CE uses $\alpha = .25$ with $\gamma \in \{1, 2\}$. This ablation uses the uniform-$t$ CE flow, a VGAE prior, $K = 100$, and one training seed.

Table 20: Sparse-aware CE weighting. AUC/AP, FNR/FPR, and the absolute FNR–FPR gap are percentages. Lower gap indicates a more balanced fixed-threshold operating point.

| | ENZYMES | | | PROTEINS | | |
|---|---|---|---|---|---|---|
| Weighting | AUC/AP ↑ | FNR/FPR | Gap ↓ | AUC/AP ↑ | FNR/FPR | Gap ↓ |
| Unweighted | 64.5/27.1 | 67.8/16.6 | 51.3 | 65.1/34.4 | 63.6/19.3 | 44.3 |
| Batch scalar (paper) | 65.1/25.9 | 38.3/40.4 | **2.0** | 64.2/32.0 | 23.3/57.8 | 34.6 |
| Per-graph | 64.0/26.4 | 40.6/38.0 | 2.6 | 64.7/33.1 | 44.7/34.8 | **10.0** |
| Focal, $\gamma = 1$ | 62.5/26.0 | 72.6/12.7 | 59.9 | 64.7/34.0 | 72.7/12.4 | 60.3 |
| Focal, $\gamma = 2$ | 62.0/25.1 | 71.0/14.5 | 56.5 | 65.8/34.3 | 62.9/19.2 | 43.7 |

Per-graph weighting reduces the PROTEINS gap from 34.6 to 10.0 points while preserving AUC/AP. On ENZYMES, the batch and per-graph variants are already balanced, and the batch rule is marginally better. Focal weighting does not improve the fixed-threshold balance, so the benefit of per-graph weighting is dataset-specific rather than uniform. Appendix E.12 separately tests density-adaptive source noise on the MSE branch; that ablation is negative overall, and fixed Gaussian perturbation remains the default.

### E.7 Direct-refiner control and endpoint-anchored PIFM

We compare PIFM with a Direct GNN refiner that has the same prior-informed input, observed-entry mask, clean training graphs, denoising backbone, and parameter count. Direct fixes the time input at $t = 0$, applies one residual correction, and performs no interpolation or ODE integration. All LPFormer rows in Table 21 evaluate the clean Stage-1 output before source-noise perturbation. To study endpoint coverage, Model 2 assigns 25% of training-time draws to $t = 0$, whereas Model 1 uses uniform-time flow matching together with the endpoint loss in Eq. (14).

Table 21: Matched Direct control under LPFormer, MSE, and 50% all-pairs masking. All four $MMD^2$ values use reconstructions binarized at 0.5, with each completed test graph as one distributional sample; degree, clustering, and orbit use the GDSS kernels. AUC is macro graph-wise over eligible graphs. Values are means $\pm$ sample standard deviations over five seeds; higher AUC and lower MMD are better.

| Dataset | Method | $K$ | AUC at $K$ | NSPDK $MMD^2$ | Degree $MMD^2$ | Clustering $MMD^2$ | Orbit $MMD^2$ |
|---|---|---|---|---|---|---|---|
| ENZYMES | LPFormer prior | 0 | $0.6562 \pm 0.0120$ | $0.1254 \pm 0.0102$ | $0.4354 \pm 0.0577$ | $0.1659 \pm 0.0429$ | $0.1117 \pm 0.0409$ |
| ENZYMES | Direct GNN refiner | one-shot | $\mathbf{0.7320} \pm 0.0145$ | $0.2453 \pm 0.0205$ | $1.0002 \pm 0.1093$ | $0.4258 \pm 0.0680$ | $0.1929 \pm 0.0376$ |
| ENZYMES | Model 2 | 1 | $0.7230 \pm 0.0100$ | $0.2414 \pm 0.0275$ | $0.9892 \pm 0.0876$ | $0.4308 \pm 0.0592$ | $0.1967 \pm 0.0271$ |
| ENZYMES | Model 2 | 100 | $0.6197 \pm 0.0167$ | $0.0842 \pm 0.0070$ | $0.2496 \pm 0.0288$ | $\mathbf{0.1763} \pm 0.0365$ | $\mathbf{0.0357} \pm 0.0084$ |
| ENZYMES | Model 1 | 1 | $0.7316 \pm 0.0139$ | $0.2472 \pm 0.0267$ | $1.0116 \pm 0.1379$ | $0.4297 \pm 0.0481$ | $0.2014 \pm 0.0388$ |
| ENZYMES | Model 1 | 100 | $0.6099 \pm 0.0171$ | $\mathbf{0.0796} \pm 0.0092$ | $\mathbf{0.2140} \pm 0.0300$ | $0.1905 \pm 0.0354$ | $0.0539 \pm 0.0139$ |
| PROTEINS | LPFormer prior | 0 | $0.6767 \pm 0.0167$ | $0.0886 \pm 0.0098$ | $0.3419 \pm 0.0511$ | $0.1446 \pm 0.0230$ | $0.0772 \pm 0.0289$ |
| PROTEINS | Direct GNN refiner | one-shot | $\mathbf{0.7427} \pm 0.0154$ | $0.1760 \pm 0.0319$ | $0.7210 \pm 0.1566$ | $0.3318 \pm 0.0958$ | $0.1277 \pm 0.0277$ |
| PROTEINS | Model 2 | 1 | $0.7299 \pm 0.0106$ | $0.1727 \pm 0.0319$ | $0.7131 \pm 0.1632$ | $0.3192 \pm 0.0916$ | $0.1293 \pm 0.0247$ |
| PROTEINS | Model 2 | 100 | $0.6541 \pm 0.0104$ | $0.0493 \pm 0.0035$ | $0.1534 \pm 0.0599$ | $\mathbf{0.1410} \pm 0.0651$ | $\mathbf{0.0327} \pm 0.0235$ |
| PROTEINS | Model 1 | 1 | $0.7412 \pm 0.0098$ | $0.1650 \pm 0.0336$ | $0.6840 \pm 0.1423$ | $0.3057 \pm 0.0923$ | $0.1251 \pm 0.0222$ |
| PROTEINS | Model 1 | 100 | $0.6384 \pm 0.0133$ | $\mathbf{0.0468} \pm 0.0052$ | $\mathbf{0.1474} \pm 0.0334$ | $0.1707 \pm 0.0540$ | $0.0401 \pm 0.0381$ |
| IMDB-B | LPFormer prior | 0 | $0.8979 \pm 0.0052$ | $0.0366 \pm 0.0026$ | $0.0540 \pm 0.0137$ | $0.1904 \pm 0.0505$ | $0.0877 \pm 0.0202$ |
| IMDB-B | Direct GNN refiner | one-shot | $\mathbf{0.9511} \pm 0.0034$ | $0.0200 \pm 0.0022$ | $0.0126 \pm 0.0034$ | $0.0560 \pm 0.0155$ | $0.0114 \pm 0.0042$ |
| IMDB-B | Model 2 | 1 | $0.9498 \pm 0.0049$ | $0.0200 \pm 0.0027$ | $0.0148 \pm 0.0051$ | $0.0574 \pm 0.0219$ | $0.0137 \pm 0.0036$ |
| IMDB-B | Model 2 | 100 | $0.9333 \pm 0.0030$ | $0.0251 \pm 0.0042$ | $0.0176 \pm 0.0126$ | $0.1480 \pm 0.0558$ | $0.0186 \pm 0.0091$ |
| IMDB-B | Model 1 | 1 | $0.9499 \pm 0.0048$ | $\mathbf{0.0192} \pm 0.0015$ | $\mathbf{0.0116} \pm 0.0044$ | $\mathbf{0.0499} \pm 0.0151$ | $\mathbf{0.0108} \pm 0.0020$ |
| IMDB-B | Model 1 | 100 | $0.9357 \pm 0.0052$ | $0.0219 \pm 0.0030$ | $0.0125 \pm 0.0050$ | $0.1262 \pm 0.0592$ | $0.0162 \pm 0.0079$ |

At $K = 1$, Model 1 matches Direct's AUC within observed run-to-run variation: the absolute Direct-minus-PIFM gaps are 0.0004, 0.0015, and 0.0012 on ENZYMES, PROTEINS, and IMDB-B, respectively, each smaller than the reported across-seed standard deviation. On ENZYMES and PROTEINS, moving Model 1 from $K = 1$ to $K = 100$ lowers NSPDK $\text{MMD}^2$ by 3.1–3.5× and lowers degree and orbit $\text{MMD}^2$ by a similar or larger factor, while reducing AUC by about 0.10–0.12; on IMDB-B, $K = 100$ does not improve any of the four $\text{MMD}^2$ values over $K = 1$, although all remain below the LPFormer prior. The one exception to the distributional gains is clustering $\text{MMD}^2$ on the sparse datasets, where the $K = 100$ reconstructions (0.1905 on ENZYMES, 0.1707 on PROTEINS) remain slightly above the LPFormer prior (0.1659 and 0.1446); we report this rather than restricting the comparison to the metrics that favor the flow. Direct therefore provides a strong distortion-oriented control, while the multi-step flow outputs define empirical operating points.

### E.8 Ablation

Our method has two main hyperparameters:

1. $\sigma_s$, which is used for computing $\boldsymbol{\epsilon}_s \sim \mathcal{N}(0, \sigma_s^2)$ in $\mathbf{A}_0 = \mathbf{A}^{\mathcal{O}} + (1 - \xi) \odot \left( f_{\text{prior}}(\mathbf{A}^{\mathcal{O}}) + \boldsymbol{\epsilon}_s \right)$

2. $K$, which is the total number of steps in the Euler approximation

We ablate these two hyperparameters.

**Performance as a function of $\sigma_s$.** We run an ablation of the performance of PIFM with GraphSAGE as a function of $\sigma_s$. We focus on ENZYMES and IMDB, and we evaluate the ROC for the best value of $K$ for each noise level.

The ablation is illustrated in Fig. 4. First, we observe that the gains of using PIFM are higher for a smaller drop rate, as expected; in particular, we observe that PIFM with $\sigma_s$ jumps from $\approx 0.73$ for $\sigma_s = 0$ to $\approx 0.81$ for $\sigma_s = 0.1$. Second, for both configurations, performance peaks not at zero noise, but at a small noise level of $\sigma_s = 0.1$. This suggests that a slight injection of noise benefits model generalization. Beyond this optimal point, increasing the noise level leads to a steady decline in performance, meaning that the effect of the prior decreases, as expected.

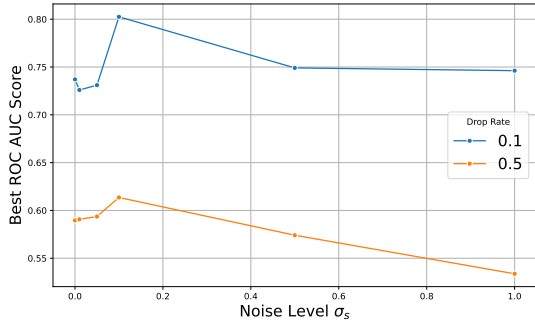

Figure 4: ROC as a function of the noise $\sigma_s$ in $p(\mathbf{A}_0)$. The impact of noise level $\sigma_s$ on model performance, measured by the best ROC AUC score. Results are shown for two different drop rates: 0.1 (blue) and 0.5 (orange). A small amount of noise improves performance for both configurations, after which increasing noise leads to performance degradation.

**Performance as a function of $K$.** We measure empirical AUC sensitivity while varying $K$ from 1 to 100. Figures 5 and 6 report 10% and 50% all-pairs masking, respectively, across five noise levels. The highest observed AUC usually occurs for $K < 10$; with moderate noise, the highest overall value ($\approx 0.80$) occurs at $K = 1$, and its advantage diminishes as the step count increases. The zero-noise model is more stable at larger

$K$, whereas $\sigma_s = 1$ degrades AUC throughout the sweep. These curves are descriptive test-set sensitivities rather than a validation-selected deployment rule or evidence that $K$ is a theoretical trade-off parameter.

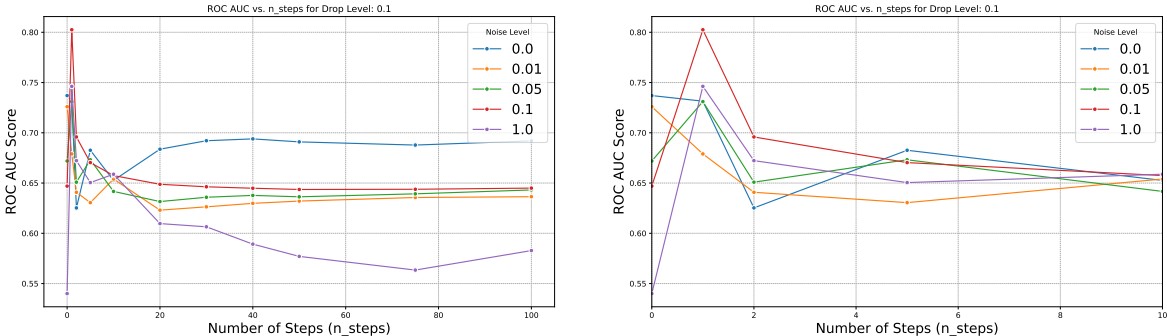

Figure 5: Empirical ROC-AUC sensitivity to the Euler step count $K$ under 10% all-pairs masking and five noise levels. The highest observed AUC in this sweep usually occurs below 10 steps; this is not a validation-selected operating point.

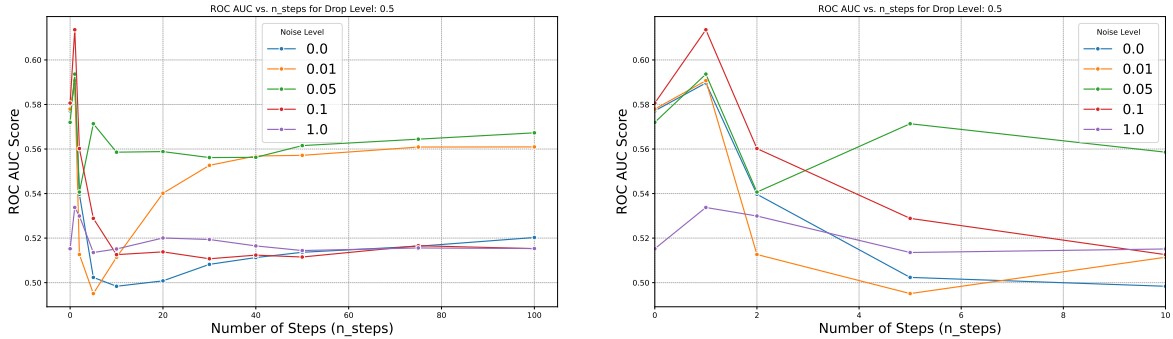

Figure 6: Empirical ROC-AUC sensitivity to the Euler step count $K$ under 50% all-pairs masking and five noise levels. The highest observed AUC in this sweep usually occurs below 10 steps; this is not a validation-selected operating point.

**Threshold levels.** We investigated the impact of alternative thresholding levels (0.3 and 0.7, beyond the standard 0.5), with results for ENZYMES and PROTEINS in Figs. 7 and 8, each reporting the ranking metrics and the binarized error rates as a function of the number of steps $K$. Since AP and AUC are computed from the raw scores, they are threshold-independent: the ROC-AUC-vs-$K$ curves are the same across the three thresholds (left panels), so thresholding does not change ranking quality. The threshold instead governs the binarized error rates (center and right panels): lowering it reduces the FNR at the cost of a higher FPR (e.g., on ENZYMES at $K{=}1$ the FNR drops from 95.4 to 82.0 as the FPR rises from 2.8 to 9.9 when lowering from $\tau = 0.7$ to $\tau = 0.3$). This effect is appreciable only for small $K$ (and for the prior at $K{=}0$), where the model's outputs lie nearer the decision boundary; for larger $K$ the outputs are pushed towards the binary extremes (0 or 1) and the three thresholds nearly coincide. Across steps, these three pre-specified thresholds mainly trade FNR against FPR; this sweep is a sensitivity diagnostic and is not used to select an operating point. The validation-only MCC selection and its test metrics are reported separately in Appendix E.1. Fig. 9 gives the complementary view, sweeping $\tau$ continuously at a fixed set of step counts.

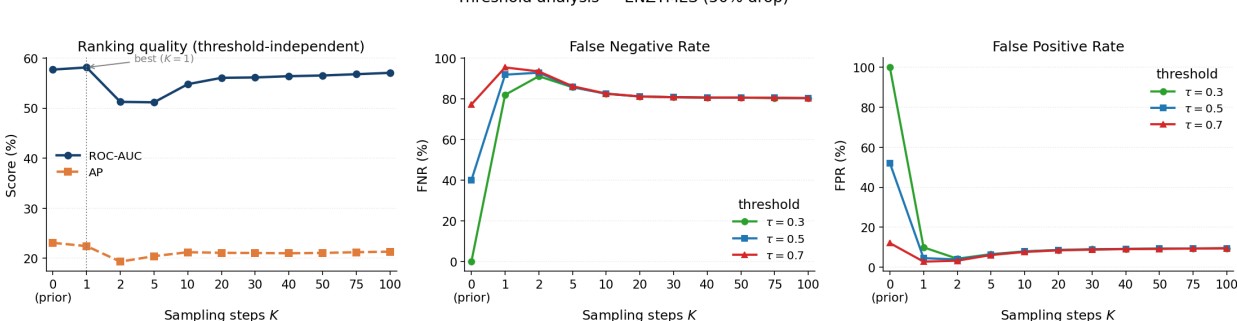

Figure 7: Threshold analysis on ENZYMES (link prediction, 50% drop) versus the number of sampling steps $K$. **Left:** ROC-AUC and AP are computed on the raw scores and are therefore *identical* for all binarization thresholds $\tau \in \{0.3, 0.5, 0.7\}$ (threshold-independent), peaking at $K=1$. **Center/Right:** the false-negative (FNR) and false-positive (FPR) rates at $\tau = 0.3, 0.5, 0.7$; the threshold shifts only these binarized error rates, and only appreciably at small $K$, where the outputs lie near the decision boundary. "$K=0$" is the prior initialization.

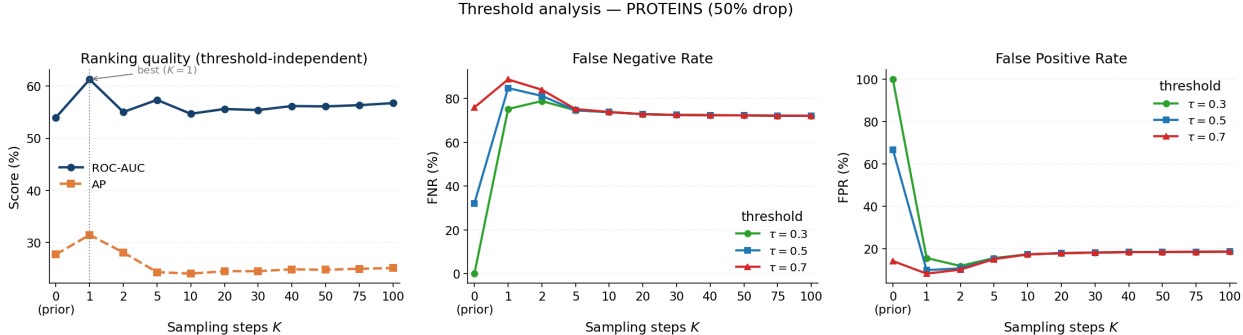

Figure 8: Threshold analysis on PROTEINS (link prediction, 50% drop) versus the number of sampling steps $K$, with panels as in Fig. 7. ROC-AUC and AP (left) coincide across thresholds, while lowering the threshold trades a lower FNR (center) for a higher FPR (right).

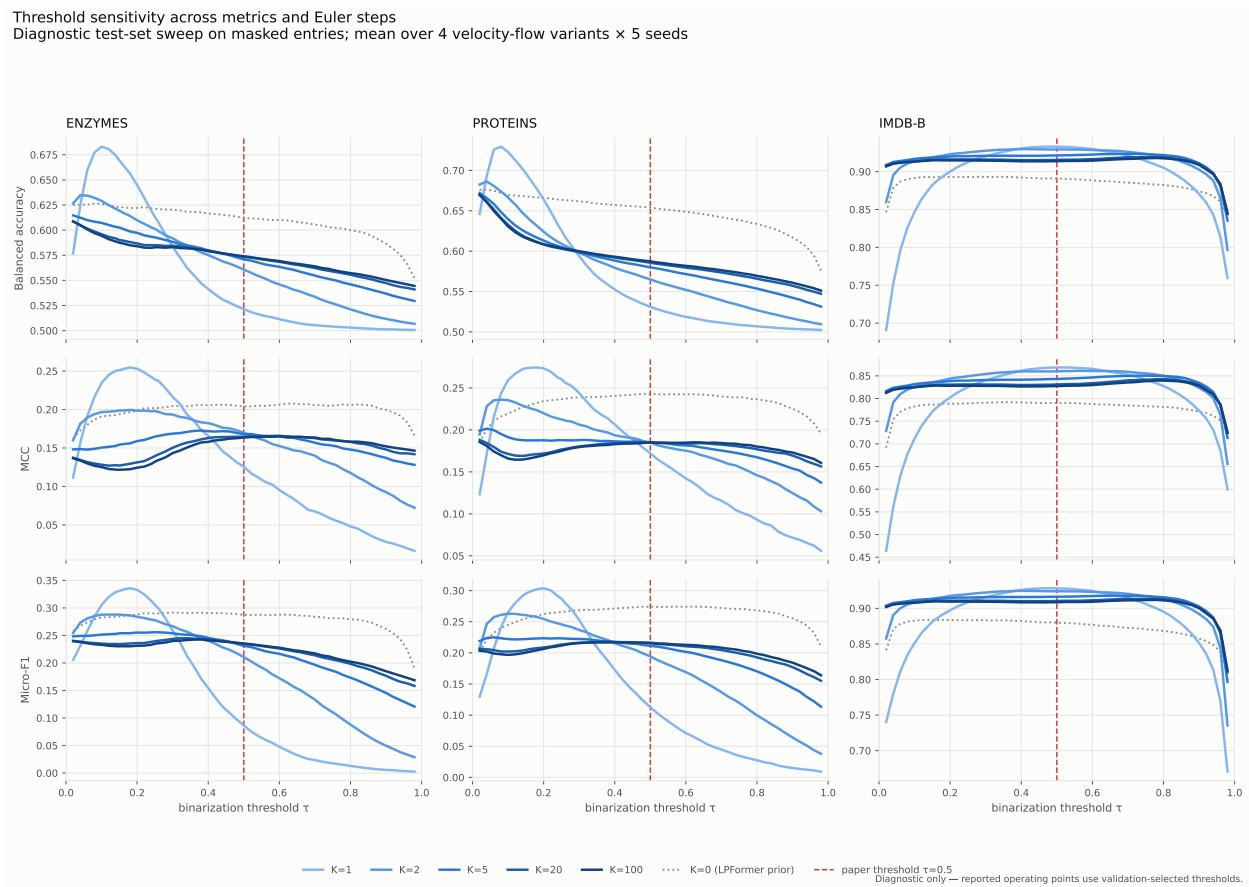

Figure 9: **Threshold sensitivity of binarized metrics.** Rows show balanced accuracy, MCC, and micro-F1; columns show ENZYMES, PROTEINS, and IMDB-B. Curves sweep the binarization threshold $\tau$: solid curves show $K \in \{1, 2, 5, 20, 100\}$, averaged over four velocity-flow variants and five seeds, while the dotted curve shows the LPFormer prior ($K=0$). The dashed line marks $\tau=0.5$. All results use LPFormer, MSE, and 50% all-pairs masking. On the sparse datasets, $K=1$ is most threshold-sensitive, whereas larger $K$ produces flatter curves. On IMDB-B, all curves are nearly flat for $\tau \approx 0.2$–$0.9$. At $\tau=0.5$, the prior exceeds the flow models in MCC and micro-F1 on ENZYMES and PROTEINS, despite weaker ranking. This diagnostic selects no thresholds; reported operating points use validation-selected values in Tables 12–13.

## E.9 Distortion-perception trade-off.

Figures 10 and 11 report $\text{MMD}^2$ over the Euler step count. MMD often drops sharply over the first few steps, especially for $0 < \sigma_s \leq 0.1$, but can plateau or rebound after an early minimum. These curves characterize sensitivity to the numerical operating point rather than a calibrated map from $K$ to perception, so noise and step count should be selected using validation criteria appropriate to the downstream task. The fixed-$K$ comparison in Table 21 illustrates the dataset dependence: $K = 100$ improves NSPDK MMD over $K = 1$ on ENZYMES and PROTEINS, but not on IMDB-B.

Table 22 extends this fixed-$K$ comparison to all five priors and both distortions, on the models reported in Table 2. Across the 30 cells, AUC is highest at $K = 1$ in every case, confirming that the $K = 1$ endpoint minimizes distortion; the perceptual gain from larger $K$ is real for the MSE flow on sparse graphs (from $K = 1$ to $K = 100$, NSPDK and degree $\text{MMD}^2$ each improve in 10/15 cells and clustering $\text{MMD}^2$ in 9/15, concentrated on ENZYMES and PROTEINS) but is far weaker for the CE flow (at most 7/15 cells on any metric: 7/15 NSPDK, 3/15 degree, 4/15 clustering, with weak ratios), and it is always accompanied by a drop in AUC.

Table 22: Distortion–perception $K$-sweep on the models reported in Table 2, across all five priors and both distortions, at 50% masking with fixed-0.5 binarization. Values are mean $\pm$ sample standard deviation over five seeds; AUC (higher is better) measures held-out-edge ranking, and NSPDK, degree, and clustering $\text{MMD}^2$ (lower is better) measure distributional fidelity (degree and clustering use the same GDSS kernels as Table 21). The MSE and CE panels are the Refinement A and Refinement B models defined in Appendix D.4. AUC is highest at $K=1$ in every cell. For the MSE flow, larger $K$ lowers NSPDK $\text{MMD}^2$ on the sparse datasets (all ENZYMES and 4/5 PROTEINS) and lowers degree $\text{MMD}^2$ on a similar majority (4/5 of each), but improves neither on dense IMDB-B; clustering $\text{MMD}^2$ follows the same but weaker trend.

| | | AUC ↑ | | NSPDK MMD² ↓ | | Degree MMD² ↓ | | Clustering MMD² ↓ | |
|---|---|---|---|---|---|---|---|---|---|
| Dataset | Prior | $K$=1 | $K$=100 | $K$=1 | $K$=100 | $K$=1 | $K$=100 | $K$=1 | $K$=100 |
| **MSE distortion** — endpoint-anchored flow (Refinement A, main MSE results) | | | | | | | | | |
| ENZYMES | Node2Vec | **0.625** ± 0.013 | 0.582 ± 0.014 | 0.273 ± 0.028 | **0.113** ± 0.020 | 1.150 ± 0.107 | **0.294** ± 0.124 | 0.515 ± 0.074 | **0.211** ± 0.066 |
| ENZYMES | GraphSAGE | **0.625** ± 0.009 | 0.557 ± 0.011 | 0.270 ± 0.034 | **0.090** ± 0.014 | 1.161 ± 0.147 | **0.196** ± 0.063 | 0.507 ± 0.093 | **0.338** ± 0.141 |
| ENZYMES | VGAE | **0.686** ± 0.023 | 0.629 ± 0.023 | 0.255 ± 0.032 | **0.085** ± 0.014 | 1.075 ± 0.143 | **0.131** ± 0.046 | 0.467 ± 0.077 | **0.251** ± 0.116 |
| ENZYMES | NCNC | **0.671** ± 0.030 | 0.632 ± 0.015 | 0.201 ± 0.053 | **0.186** ± 0.006 | **0.863** ± 0.214 | 0.895 ± 0.093 | **0.379** ± 0.113 | 0.400 ± 0.046 |
| ENZYMES | LPFormer | **0.732** ± 0.014 | 0.610 ± 0.017 | 0.247 ± 0.027 | **0.080** ± 0.009 | 1.012 ± 0.138 | **0.214** ± 0.030 | 0.430 ± 0.048 | **0.191** ± 0.035 |
| PROTEINS | Node2Vec | **0.650** ± 0.013 | 0.569 ± 0.013 | 0.204 ± 0.022 | **0.078** ± 0.018 | 0.853 ± 0.090 | **0.262** ± 0.154 | 0.391 ± 0.112 | **0.202** ± 0.078 |
| PROTEINS | GraphSAGE | **0.649** ± 0.021 | 0.577 ± 0.010 | 0.199 ± 0.027 | **0.058** ± 0.007 | 0.843 ± 0.139 | **0.243** ± 0.059 | 0.377 ± 0.097 | **0.237** ± 0.070 |
| PROTEINS | VGAE | **0.692** ± 0.011 | 0.631 ± 0.011 | 0.181 ± 0.025 | **0.058** ± 0.010 | 0.765 ± 0.108 | **0.107** ± 0.025 | 0.343 ± 0.108 | **0.207** ± 0.058 |
| PROTEINS | NCNC | **0.680** ± 0.035 | 0.624 ± 0.035 | **0.082** ± 0.015 | 0.113 ± 0.017 | **0.299** ± 0.067 | 0.547 ± 0.092 | **0.172** ± 0.029 | 0.269 ± 0.046 |
| PROTEINS | LPFormer | **0.741** ± 0.010 | 0.638 ± 0.013 | 0.165 ± 0.034 | **0.047** ± 0.005 | 0.684 ± 0.142 | **0.147** ± 0.033 | 0.306 ± 0.092 | **0.171** ± 0.054 |
| IMDB-B | Node2Vec | **0.848** ± 0.012 | 0.828 ± 0.012 | 0.042 ± 0.003 | **0.036** ± 0.003 | 0.041 ± 0.008 | **0.027** ± 0.008 | 0.427 ± 0.079 | **0.354** ± 0.072 |
| IMDB-B | GraphSAGE | **0.935** ± 0.011 | 0.900 ± 0.020 | **0.030** ± 0.003 | 0.032 ± 0.006 | 0.030 ± 0.008 | **0.029** ± 0.011 | **0.171** ± 0.045 | 0.228 ± 0.051 |
| IMDB-B | VGAE | **0.925** ± 0.014 | 0.894 ± 0.021 | **0.029** ± 0.003 | 0.036 ± 0.006 | 0.030 ± 0.008 | **0.028** ± 0.013 | **0.195** ± 0.048 | 0.321 ± 0.035 |
| IMDB-B | NCNC | **0.911** ± 0.020 | 0.890 ± 0.019 | **0.030** ± 0.004 | 0.040 ± 0.008 | **0.036** ± 0.012 | 0.048 ± 0.024 | **0.156** ± 0.029 | 0.306 ± 0.036 |
| IMDB-B | LPFormer | **0.950** ± 0.005 | 0.936 ± 0.005 | **0.019** ± 0.001 | 0.022 ± 0.003 | **0.012** ± 0.004 | 0.013 ± 0.005 | **0.050** ± 0.015 | 0.126 ± 0.059 |
| **CE distortion** — boundary-aware flow (Refinement B, main CE results) | | | | | | | | | |
| ENZYMES | Node2Vec | **0.622** ± 0.012 | 0.563 ± 0.011 | 0.291 ± 0.053 | **0.262** ± 0.018 | 1.054 ± 0.168 | **0.974** ± 0.071 | **0.411** ± 0.135 | 0.586 ± 0.072 |
| ENZYMES | GraphSAGE | **0.620** ± 0.021 | 0.593 ± 0.017 | 0.273 ± 0.024 | **0.236** ± 0.043 | **1.028** ± 0.023 | 1.191 ± 0.248 | **0.317** ± 0.042 | 0.367 ± 0.134 |
| ENZYMES | VGAE | **0.682** ± 0.015 | 0.647 ± 0.021 | 0.301 ± 0.027 | **0.294** ± 0.019 | 1.184 ± 0.063 | **1.161** ± 0.122 | **0.446** ± 0.082 | 0.516 ± 0.088 |
| ENZYMES | NCNC | **0.754** ± 0.018 | 0.492 ± 0.020 | **0.249** ± 0.026 | 0.297 ± 0.010 | **1.062** ± 0.090 | 1.095 ± 0.070 | **0.337** ± 0.080 | 0.635 ± 0.135 |
| ENZYMES | LPFormer | **0.721** ± 0.020 | 0.643 ± 0.017 | **0.268** ± 0.043 | 0.284 ± 0.013 | **1.065** ± 0.109 | 1.111 ± 0.070 | **0.333** ± 0.099 | 0.419 ± 0.102 |
| PROTEINS | Node2Vec | **0.638** ± 0.014 | 0.576 ± 0.011 | 0.208 ± 0.026 | **0.190** ± 0.030 | 0.792 ± 0.067 | **0.771** ± 0.099 | 0.459 ± 0.106 | **0.417** ± 0.080 |
| PROTEINS | GraphSAGE | **0.634** ± 0.012 | 0.574 ± 0.022 | **0.208** ± 0.025 | 0.210 ± 0.034 | **0.783** ± 0.067 | 0.932 ± 0.102 | 0.440 ± 0.088 | **0.375** ± 0.086 |
| PROTEINS | VGAE | **0.685** ± 0.012 | 0.634 ± 0.009 | 0.224 ± 0.026 | **0.202** ± 0.026 | 0.825 ± 0.072 | **0.818** ± 0.084 | 0.423 ± 0.092 | **0.402** ± 0.085 |
| PROTEINS | NCNC | **0.750** ± 0.017 | 0.515 ± 0.012 | 0.231 ± 0.037 | **0.217** ± 0.032 | **0.867** ± 0.094 | 0.868 ± 0.149 | **0.372** ± 0.081 | 0.523 ± 0.162 |
| PROTEINS | LPFormer | **0.729** ± 0.026 | 0.629 ± 0.026 | 0.221 ± 0.027 | **0.201** ± 0.030 | **0.845** ± 0.067 | 0.837 ± 0.112 | 0.353 ± 0.046 | **0.348** ± 0.083 |
| IMDB-B | Node2Vec | **0.846** ± 0.015 | 0.812 ± 0.040 | 0.046 ± 0.002 | **0.043** ± 0.006 | 0.052 ± 0.017 | **0.051** ± 0.033 | 0.498 ± 0.052 | **0.433** ± 0.043 |
| IMDB-B | GraphSAGE | **0.921** ± 0.011 | 0.903 ± 0.014 | **0.033** ± 0.005 | 0.033 ± 0.007 | 0.037 ± 0.012 | **0.033** ± 0.009 | 0.247 ± 0.090 | **0.238** ± 0.083 |
| IMDB-B | VGAE | **0.914** ± 0.007 | 0.897 ± 0.009 | 0.032 ± 0.002 | **0.031** ± 0.005 | 0.030 ± 0.008 | **0.026** ± 0.009 | **0.223** ± 0.036 | 0.251 ± 0.077 |
| IMDB-B | NCNC | **0.931** ± 0.018 | 0.809 ± 0.041 | **0.027** ± 0.004 | 0.071 ± 0.010 | **0.017** ± 0.005 | 0.145 ± 0.038 | **0.137** ± 0.034 | 0.475 ± 0.083 |
| IMDB-B | LPFormer | **0.948** ± 0.007 | 0.932 ± 0.015 | **0.020** ± 0.003 | 0.026 ± 0.006 | **0.013** ± 0.008 | 0.024 ± 0.022 | **0.065** ± 0.032 | 0.151 ± 0.067 |

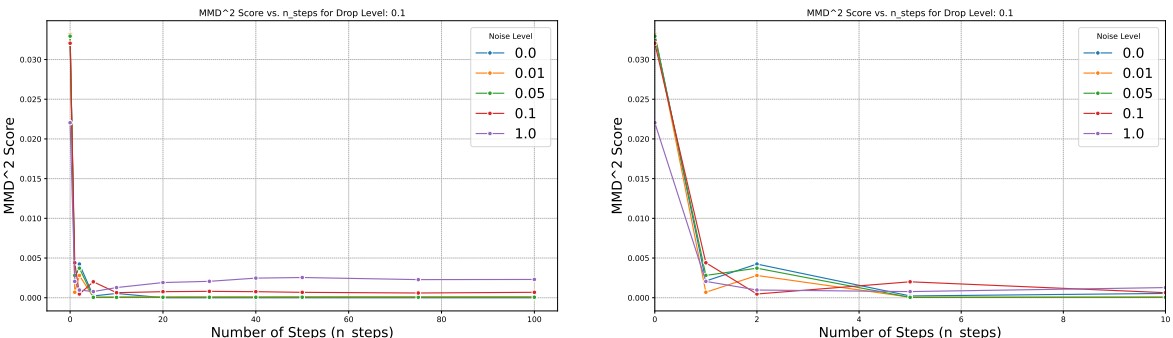

Figure 10: Empirical $\text{MMD}^2$ (lower is better) over the Euler step count $K$ under 10% all-pairs masking. Each line is a noise level $\sigma_s$; the curves need not be monotone.

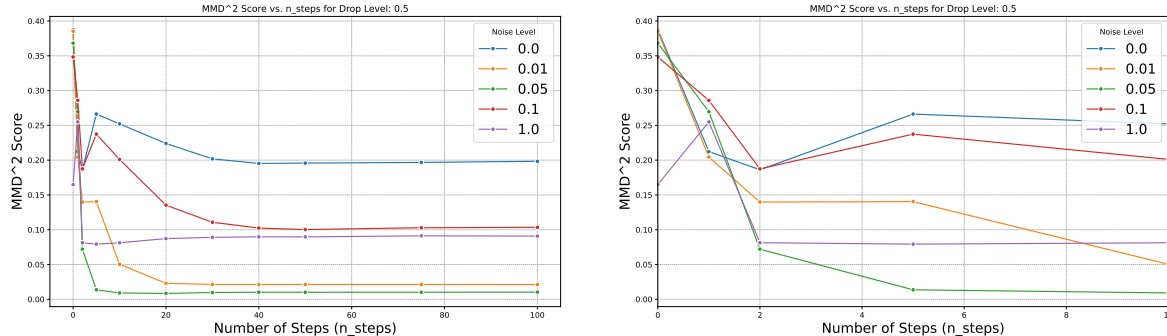

Figure 11: Empirical $\text{MMD}^2$ (lower is better) over the Euler step count $K$ under 50% all-pairs masking. Each line is a noise level $\sigma_s$; the curves need not be monotone.

## E.10 Evaluation using additional metrics

This section incorporates additional metrics to showcase the observed performance trade-off. The results for IMDB are in Figs. 12 and 13, for PROTEINS in Figs. 14 and 15, and for ENZYMES in Figs. 16a and 16b. In these runs, additional integration can move selected graph statistics toward the ground-truth distribution while AUC declines from the distortion-oriented initial step. The trend depends on the dataset and statistic and is not monotone in $K$: on dense IMDB-B, AUC decreases after the initial optimum at both drop rates, while PROTEINS and ENZYMES fluctuate across steps but remain below the $K = 1$ AUC.

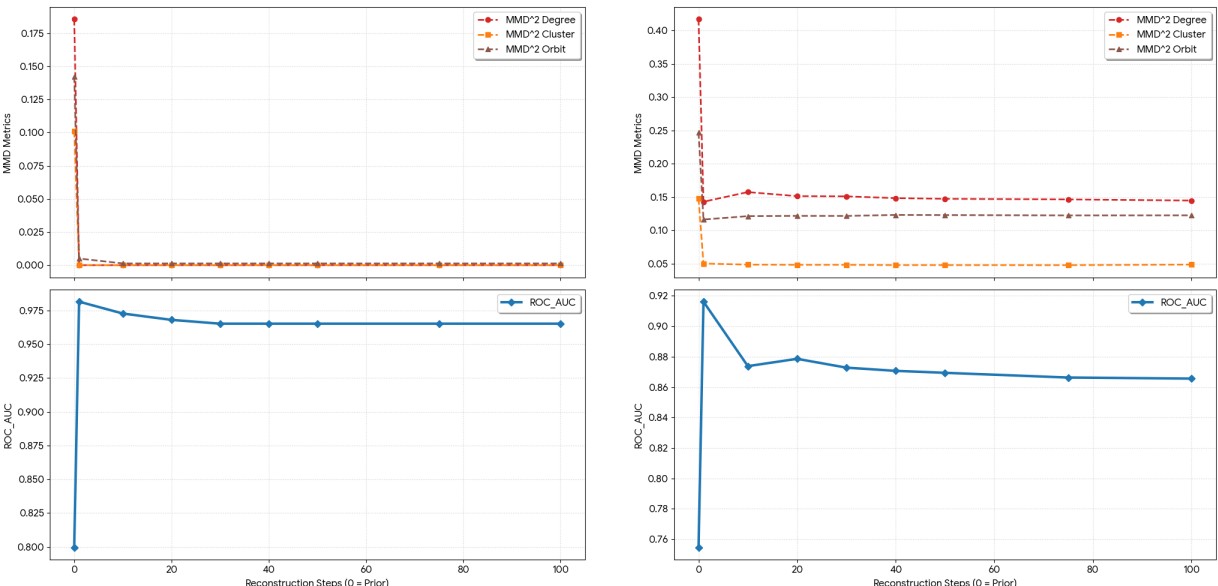

Figure 12: IMDB dataset, expansion task, 10% drop rate

Figure 13: IMDB dataset, expansion task, 50% drop rate

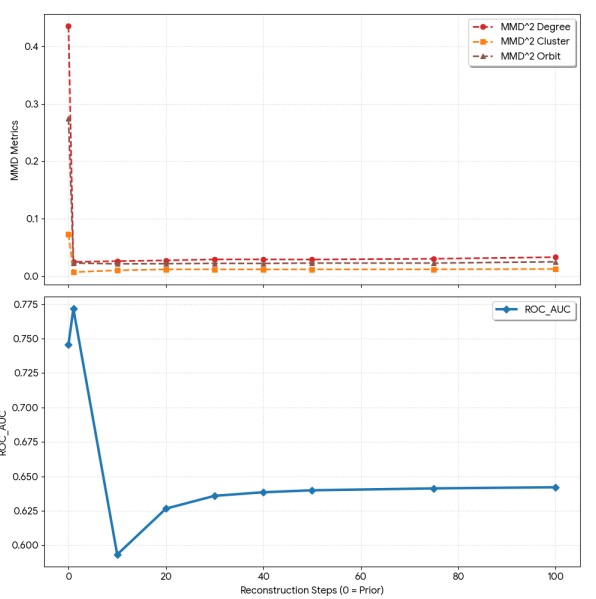

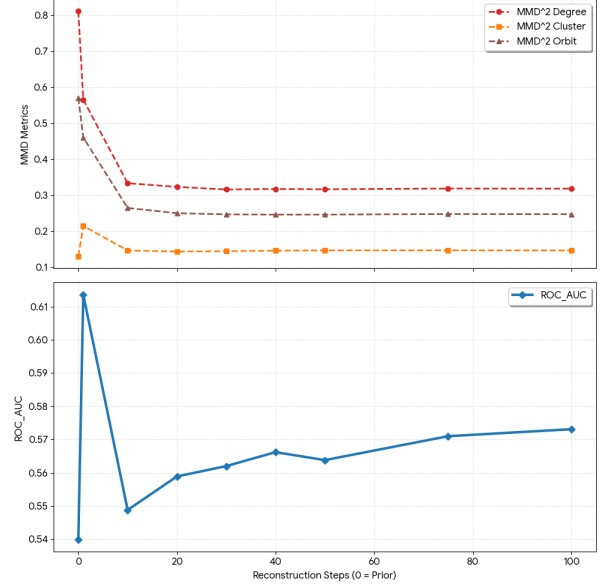

Figure 14: PROTEINS dataset, expansion, 10% drop rate

Figure 15: PROTEINS dataset, expansion, 50% drop rate

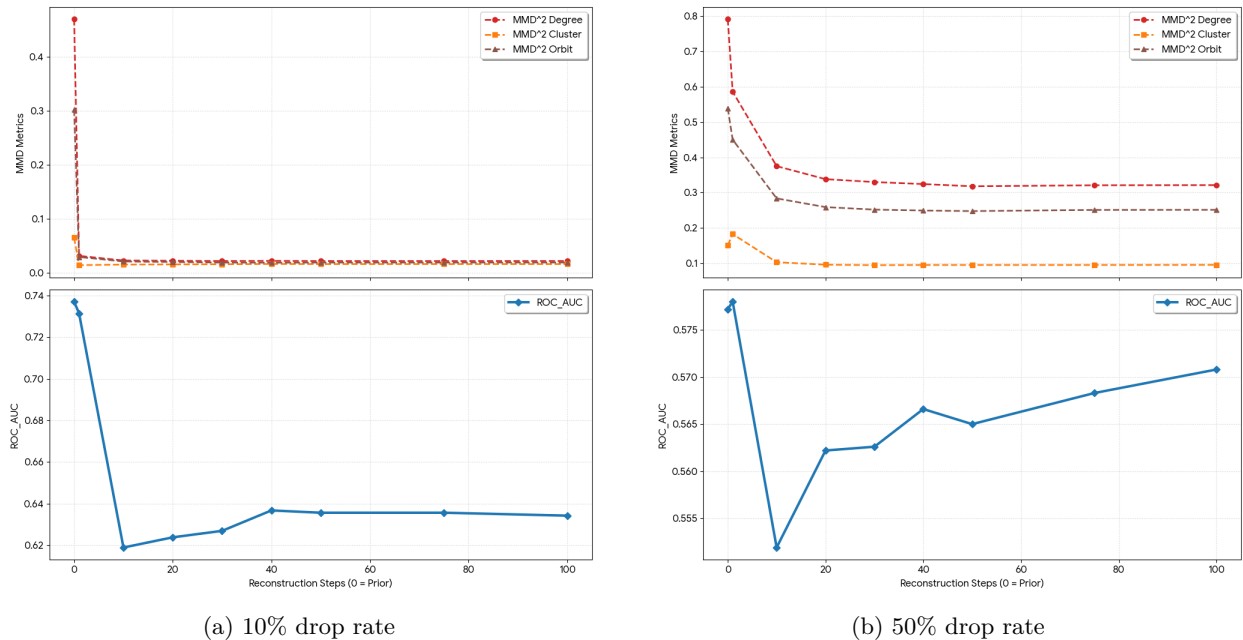

(a) 10% drop rate                    (b) 50% drop rate

Figure 16: ENZYMES dataset, expansion task.

### E.11 PIFM on large-scale graphs

In this section, we train PIFM on large-scale graphs. In particular, we focus on CORA Yang et al. (2016).

**Building the dataset.** To enable scalable diffusion training on Cora while maintaining full-graph link prediction capabilities, we introduce a subgraph-based variant of PIFM following Limnios et al. (2023). We instantiate this via an edge-centered sampling scheme, where each subgraph represents a k-hop ego-network (capped at a maximum node count) centered around a seed edge from the training split. To ensure reproducibility, we sample a fixed set of seed edges (both positive and negative) and corresponding subgraphs which remain constant throughout each run.

Following the 50/15/35 (train/validation/test) positive-edge split, we evaluate link prediction on held-out positive edges and sampled negative pairs. Furthermore, we utilize purely structural node features. By default, we concatenate: (i) Laplacian positional encodings derived from the smallest $k$ generalized eigenvectors of $Lv = \lambda Dv$ computed on the training adjacency; and (ii) a 2-dimensional local context vector comprising both raw and normalized node degrees.

This protocol differs from the TU all-pairs masks in Appendix D.3. Cora contains 2,639/792/1,847 positive edges in the train/validation/test partitions, and evaluation samples negatives one-to-one. In the edge-centered training configuration, each subgraph masks and supervises its seed pair, bypassing the generic loader's global random all-pairs masks.

**Training.** To initialize PIFM, we first pre-train the NCN structural prior on the full Cora graph using only the training edge split (50% of edges). These learned embeddings are subsequently used to initialize PIFM for inference on the test split (35% of edges).

For MSE, each training edge seeds an edge-centered subgraph. The remaining training edges form the observed context, while the centered pair is removed and used as the supervision target. For CE, we use multi-edge supervision: within each sampled subgraph, we select 50% of the remaining positive edges, that is, those other than the centered seed edge, as additional supervision targets, capped at 20, and add an equal number of negative pairs when available. All selected targets are removed from the observed context, and the endpoint CE loss is evaluated over the resulting supervision mask. We construct $\mathbf{A}_0$ from the observed

context and NCN predictions in the hidden region. MSE trains the velocity field on the centered pair, whereas CE trains the endpoint parameterization on the multi-edge mask. In seed-zero MSE multi-edge runs, one-hop and two-hop AUC were 92.33 and 90.60, versus 93.78 for NCN; we therefore use single-edge supervision for MSE and study multi-edge supervision with CE.

**Inference.** For inference, every held-out test edge seeds a k-hop subgraph. We define the observed context using the training edges, remove the centered test edge from the mask, and apply diffusion to the resulting hidden region. We evaluate stitched scores for held-out edges. To resolve potential overlaps, we aggregate predictions via logit averaging across all subgraphs where a specific edge is present, producing a single probability matrix over all node pairs. We then compute metrics on the held-out positive and negative edges. Since PIFM and the structural prior baseline are evaluated on the exact same set of pairs, the results isolate the specific benefits of the diffusion process.

**Results.** Table 23 reports the five individual runs summarized in Table 3.

Table 23: Per-seed Cora results using the shared 50/15/35 edge split. MSE uses one-hop single-edge supervision; CE uses two-hop multi-edge supervision. Metrics are percentages.

| Configuration | Seed | PIFM AUC | PIFM AP | PIFM FPR | PIFM FNR | NCN AUC | NCN AP |
|---|---|---|---|---|---|---|---|
| | 0 | 93.87 | 93.82 | 22.47 | 6.93 | 93.78 | 93.89 |
| | 1 | 92.99 | 92.85 | 30.91 | 6.23 | 93.15 | 93.66 |
| MSE, single-edge, 1-hop | 2 | 92.81 | 93.15 | 31.35 | 6.71 | 92.80 | 93.15 |
| | 3 | 92.87 | 91.86 | 20.74 | 9.20 | 93.52 | 94.08 |
| | 4 | 94.25 | 94.47 | 28.37 | 5.25 | 94.22 | 94.41 |
| | 0 | 93.83 | 93.10 | 0.54 | 86.63 | 93.78 | 93.89 |
| | 1 | 67.45 | 60.64 | 43.85 | 19.49 | 93.15 | 93.66 |
| CE, multi-edge, 2-hop | 2 | 92.20 | 90.40 | 2.22 | 55.22 | 92.80 | 93.15 |
| | 3 | 92.48 | 90.45 | 4.06 | 31.19 | 93.52 | 94.08 |
| | 4 | 89.87 | 87.23 | 92.04 | 2.92 | 94.22 | 94.41 |

Across the five seeds the MSE configuration tracks the NCN prior rather than improving on it: PIFM AUC is higher on three seeds and lower on two, and PIFM AP is below the prior on three. The CE configuration is weaker and unstable—its AUC is below the prior on four of five seeds, and seed 1 collapses to 67.45 AUC, which alone accounts for the ±11.11 spread reported in Table 3. On Cora, PIFM at best matches the NCN prior on edge ranking; its reported gain is confined to the distributional metrics.

Table 24: Matched seed-zero Cora neighborhood control with multi-edge supervision and the same network configuration.

| Loss | 1-hop AUC | 2-hop AUC | $\Delta(2\text{-hop} - 1\text{-hop})$ |
|---|---|---|---|
| MSE | 92.33 | 90.60 | $-1.73$ |
| CE | 93.57 | 93.83 | $+0.26$ |

Going from one to two hops changes test AUC by $-1.73$ points under MSE ($92.33 \rightarrow 90.60$) and by $+0.26$ under CE ($93.57 \rightarrow 93.83$). These are single-seed test differences rather than validation-selected comparisons, and the CE gap is far smaller than the CE seed spread in Table 23; we therefore do not read this control as evidence of a neighborhood-depth benefit.

### E.12 Scope extensions and robustness analyses

Except where a row is explicitly taken from Table 2, the PIFM variants in this subsection are trained with uniform time sampling, $t \sim U[0, 1]$, and without the endpoint anchor in Eq. (14). They are reported as scoped diagnostics rather than as the configuration used for the main results. Appendix D.4 states the configuration used for each set of reported results.

**Categorical molecular reconstruction.**   We extend PIFM to ZINC molecules (Dwivedi et al., 2023), using 28-way atom features and four bond states for each unordered pair: no bond, single, double, or triple. A learned prior predicts a four-way distribution on each masked pair, while observed one-hot bonds remain fixed. Hidden pairs follow a linear simplex path from the prior distribution to the clean bond tensor, and the endpoint network is trained with inverse-frequency categorical CE. Sampling updates only hidden pairs, projects them to the simplex, restores observed bonds, and takes the final class argmax. We evaluate categorical VGAE, GRIT, NCN, and NBFNet priors; Table 25 reports seed-zero Macro-F1 on hidden pairs and bond-aware NSPDK MMD on reconstructed graph sets.

Table 25: Categorical ZINC reconstruction. Each entry compares the prior output at $K = 0$ with the common refined endpoint at $K = 100$. Higher Macro-F1 and lower NSPDK MMD are better. All values use seed 0.

| Mask | Prior | Macro-F1 ↑ | | NSPDK MMD ↓ | |
|---|---|---|---|---|---|
| | | $K = 0$ | $K = 100$ | $K = 0$ | $K = 100$ |
| 10% | VGAE | .462 | .483 | .009 | .006 |
| | GRIT | .493 | **.524** | .008 | **.002** |
| | NCN | .468 | .491 | .012 | .007 |
| | NBFNet | .492 | .511 | .007 | .004 |
| 20% | VGAE | .430 | .488 | .070 | .033 |
| | GRIT | .484 | **.534** | .047 | **.018** |
| | NCN | .453 | .513 | .065 | .019 |
| | NBFNet | .476 | .517 | .038 | .023 |
| 30% | VGAE | .350 | .447 | .167 | .061 |
| | GRIT | .403 | **.479** | .123 | .054 |
| | NCN | .410 | .473 | .107 | **.042** |
| | NBFNet | .395 | .461 | .103 | .050 |
| 50% | VGAE | .279 | .354 | .209 | .151 |
| | GRIT | .296 | **.379** | .209 | .131 |
| | NCN | .299 | .370 | .189 | .092 |
| | NBFNet | .262 | .329 | .183 | **.081** |

Refinement improves both metrics over the corresponding source in all 16 prior–mask combinations, and the mean Macro-F1 gain rises from .024 at 10% masking to .074 at 50%. The best informed result also exceeds uniform and Dirichlet-simplex sources at every mask rate; at 50%, their Macro-F1/NSPDK values are .261/.100 and .272/.268, respectively. Validity alone is unreliable in this regime: the uniform source reaches .324 validity but only .024 Tanimoto similarity, while FCD is undefined when too few valid molecules remain. This single-seed experiment establishes categorical molecular reconstruction, not general heterogeneous-graph support, and the prior rankings should be read as descriptive.

**Unconditional generators adapted to reconstruction.**   We train DiGress, GDSS, DeFoG, and CatFlow on clean graphs, then apply RePaint-style clamping so every observed entry is restored after each reverse or flow step. This adaptation enforces observation consistency but does not provide an observation-dependent reconstruction prior. Table 26 reports raw hidden-pair scores at 50% masking.

Table 26: Unconditional generators adapted with RePaint. Entries are AUC/AP percentages on hidden pairs. DeFoG and CatFlow use $K = 100$ and seed 0; DiGress and GDSS are the single-run controls from Table 2. The final row reproduces PIFM with an NCNC prior and CE loss, the configuration attaining the best AUC and AP in Table 2 on ENZYMES and PROTEINS, as a five-seed mean $\pm$ standard deviation.

| Method | ENZYMES | PROTEINS | IMDB-B |
|---|---|---|---|
| DiGress + RePaint | 55.2/17.3 | 55.5/23.7 | 58.9/56.5 |
| GDSS + RePaint | 49.7/16.4 | 51.4/22.3 | 51.2/53.4 |
| DeFoG + RePaint | 52.2/15.4 | 49.9/18.9 | 49.8/50.0 |
| CatFlow + RePaint | 49.4/15.4 | 49.2/20.4 | 51.4/53.0 |
| PIFM (NCNC, CE) | $\mathbf{75.4_{\pm 1.8}/38.5_{\pm 4.0}}$ | $\mathbf{75.0_{\pm 1.7}/45.1_{\pm 2.0}}$ | $\mathbf{93.1_{\pm 1.8}/94.1_{\pm 0.8}}$ |

DeFoG and CatFlow lie in the narrow AUC range 49.2–52.2, similar to the existing RePaint controls and near chance under this conditional adaptation. These single-seed results support the use of an observation-dependent source in this setting, but do not establish that either generator is intrinsically unsuitable for conditional tasks. The gap to PIFM does not depend on which PIFM configuration is chosen: every PIFM row of Table 2 exceeds the strongest RePaint variant on both metrics and on all three datasets, and even the weakest such row does so by at least 6.7 AUC and 7.7 AP points.

**Cora memory profile.** We profile the two-hop Cora pipeline on an A100-80GB with batch size 128. The node cap controls dense padding and therefore the $B \times N \times N$ edge state.

Table 27: Cora memory profile. The maximum node count is the dense-padding cap, not the realized mean.

| Node cap | Mean realized nodes | Peak allocated VRAM (GB) |
|---|---|---|
| 32 | 25.9 | 1.22 |
| 64 | 39.6 | 4.23 |
| 96 | 48.1 | 9.07 |
| 128 | 54.1 | 15.66 |
| 192 | 60.9 | 34.29 |
| 256 | 62.8 | 60.13 |

Raising the cap from 128 to 256 increases peak allocation from 15.66 to 60.13 GB; the latter reserves 77.85 GB. Over three seeds, k-hop-union and ego-union sampling are statistically tied: their AUCs are $82.75 \pm 3.92$ versus $82.14 \pm 4.59$ at cap 128 and $80.25 \pm 1.29$ versus $80.45 \pm 2.44$ at 256. This controlled $K = 1$ Node2Vec ablation is not a reproduction of the main Cora configurations, so its scores support only the relative sampler comparison. It is a memory profile and sampler ablation, not an optimized implementation.

**Adaptive source noise.** We test density-dependent noise on the uniform-$t$ MSE flow with a GraphSAGE prior, 50% masking, $\sigma_s = 0.1$, and one training seed. For observed density $\rho_i$, the Gaussian scale is .1 (fixed), $.2\sqrt{\rho_i(1 - \rho_i)}$ (density-Bernoulli), $.1\rho_i/.5$ (density-linear), or $.05/\max(\rho_i, \varepsilon)$ (density-inverse). Degree MMD is computed on each reconstructed graph set and averaged over three inference rollouts.

Table 28: Adaptive source-noise ablation. Entries compare $K = 1$ with $K = 100$. Higher AUC and lower degree MMD are better.

| Dataset | Schedule | AUC, $K = 1 \rightarrow 100$ | Degree MMD, $K = 1 \rightarrow 100$ |
|---------|----------|------------------------------|--------------------------------------|
| ENZYMES | Fixed | .591 → .576 | 1.060 → .231 |
| | Density-Bernoulli | .582 → .562 | 1.001 → .313 |
| | Density-linear | .592 → .574 | .402 → .786 |
| | Density-inverse | .551 → .530 | 1.040 → .821 |
| IMDB-B | Fixed | .914 → .879 | .064 → .046 |
| | Density-Bernoulli | .886 → .842 | .059 → .035 |
| | Density-linear | .901 → .870 | .070 → .124 |
| | Density-inverse | .893 → .848 | .070 → .053 |
| PROTEINS | Fixed | .590 → .542 | .758 → .261 |
| | Density-Bernoulli | .579 → .527 | .692 → .236 |
| | Density-linear | .596 → .555 | .574 → .491 |
| | Density-inverse | .542 → .501 | 1.132 → .767 |

Fixed Gaussian noise is the strongest overall default. Density-Bernoulli lowers MMD on IMDB-B and PROTEINS, but lowers AUC on every dataset and worsens ENZYMES MMD; density-linear and density-inverse also produce poor high-$K$ MMD in several settings. Direct Bernoulli resampling lowers AUC on ENZYMES and PROTEINS relative to fixed Gaussian noise. This is a negative single-seed MSE ablation, not a test of adaptive noise on the CE branch, and the evidence does not justify added schedule complexity. The sparse-aware CE-weighting comparison appears in Table 20.

### E.13 Transferability

We evaluate whether a uniform-$t$ PIFM checkpoint (no endpoint anchor) trained on a source dataset transfers to a different target dataset at the same masking rate. Every run starts from a GraphSAGE prior computed on the target graphs; Table 30 therefore isolates the target-only prior, while Table 29 measures the correction induced by the transferred flow. The best-over-$K$ row is a descriptive test-set oracle rather than a deployable selection rule, and the final row reports the fixed $K = 100$ endpoint.

Table 29: **PIFM transfer.** A flow trained on the source dataset is applied to a different target dataset at the same masking rate, starting from the target-only prior in Table 30. We report the descriptive test-best AUC step ($K$ in parentheses) and the fixed $K = 100$ endpoint. Metrics are percentages.

| Source checkpoint | Target dataset | Step | AP↑ | AUC↑ | FNR↓ | FPR↓ |
|-------------------|----------------|------|-----|------|------|------|
| PROTEINS (10%) | ENZYMES (10%) | best ($K$=1) | 43.87 | 76.96 | 71.96 | 4.24 |
| | | final ($K$=100) | 37.83 | 59.43 | 64.45 | 4.87 |
| IMDB-B (10%) | PROTEINS (10%) | best ($K$=1) | 46.79 | 75.53 | 47.07 | 11.62 |
| | | final ($K$=100) | 43.12 | 60.24 | 43.65 | 51.66 |
| ENZYMES (10%) | PROTEINS (10%) | best ($K$=1) | 49.63 | 76.60 | 59.02 | 6.63 |
| | | final ($K$=100) | 42.36 | 70.64 | 53.12 | 8.20 |
| PROTEINS (50%) | ENZYMES (50%) | best ($K$=1) | 25.68 | 61.88 | 95.37 | 1.70 |
| | | final ($K$=100) | 21.05 | 58.68 | 80.82 | 8.67 |
| IMDB-B (50%) | PROTEINS (50%) | best ($K$=75) | 24.88 | 59.02 | 61.02 | 24.32 |
| | | final ($K$=100) | 24.42 | 57.22 | 60.80 | 24.37 |
| ENZYMES (50%) | PROTEINS (50%) | best ($K$=1) | 26.81 | 55.80 | 86.13 | 9.97 |
| | | final ($K$=100) | 24.54 | 54.26 | 76.68 | 16.23 |

Table 30: **Prior only (no flow).** The GraphSAGE prior computed on the target graphs — the starting point for every PIFM run in Table 29. It depends only on the target, not on the source, so it is the same for every source→target pair with the same target. Metrics are AP, AUC, FNR, and FPR, all in percent.

| Target dataset | AP↑ | AUC↑ | FNR↓ | FPR↓ |
|---|---|---|---|---|
| ENZYMES (10%) | 41.27 | 71.34 | 15.49 | 69.77 |
| PROTEINS (10%) | 46.36 | 74.58 | 11.00 | 63.50 |
| ENZYMES (50%) | 23.08 | 57.72 | 40.02 | 52.16 |
| PROTEINS (50%) | 27.71 | 53.99 | 32.16 | 66.86 |

The transferred flow is dataset-dependent and sensitive to integration depth. The descriptive best-over-$K$ AUC is above the target-only prior in all six transfers, but the fixed $K = 100$ endpoint often degrades substantially, especially at 10% masking. These single-seed results diagnose unstable transfer rather than validate a mitigation strategy. We do not evaluate target adaptation, flow fine-tuning, or shared multi-dataset training; these comparisons, together with validation-based selection of integration depth, remain future work.

# F  Visual results

## F.1  Examples of reconstructed graphs

We show here a few samples for the expansion case. We plot the samples from ENZYMES, using a subset of the dataset used in Section 5.3.

**Binary comparison.** In this case, we first compare the thresholded versions (with 0.5) of the mean matrices. We compute this for 3 graphs in the test set (24, 16 and 21). The plots are in Figs. 17, 18 and 19

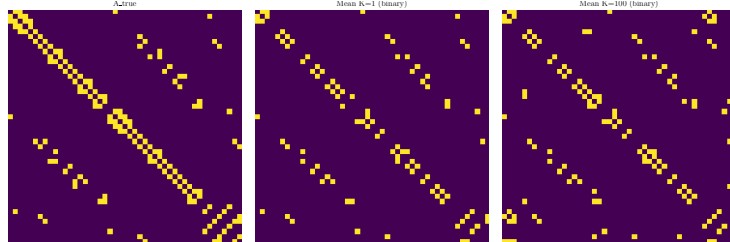

Figure 17: Graph reconstruction for sample 24, thresholded with 0.5

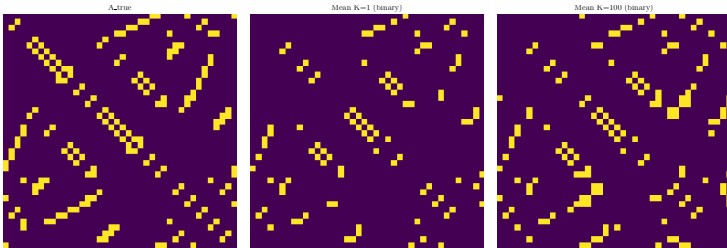

Figure 18: Graph reconstruction for sample 16, thresholded with 0.5

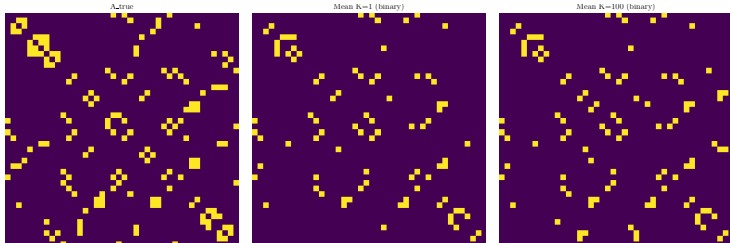

Figure 19: Graph reconstruction for sample 21, thresholded with 0.5

**Raw comparison - Mean.** In this case, we compare the raw versions of the mean matrices. We compute this for 3 graphs in the test set (24, 16 and 21). The plots are in Figs. 20, 21 and 22. Notice that the mean reconstructions for $K = 100$ have values that are between 0 and 1; this can be explained by looking at individual samples (see below), which are more diverse, and therefore, they have non-overlapping set of existing edges.

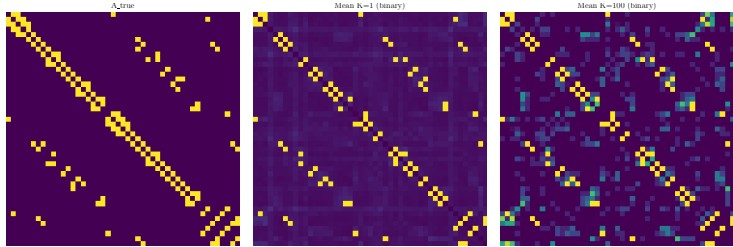

Figure 20: Graph reconstruction for sample 24, mean raw

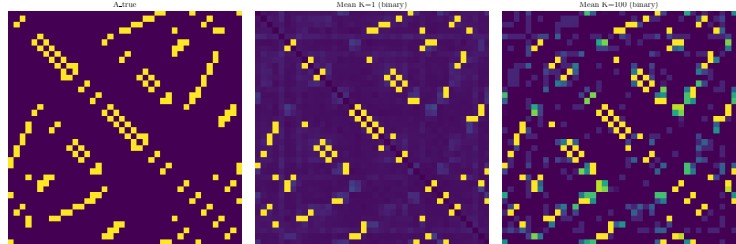

Figure 21: Graph reconstruction for sample 16, mean raw

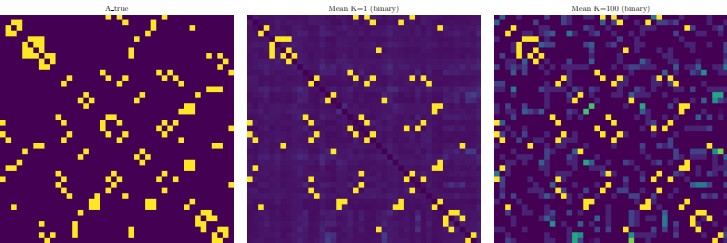

Figure 22: Graph reconstruction for sample 21, mean raw

**Raw comparison - Median.** In this case, we compare the raw versions of the median matrices. We compute this for 3 graphs in the test set (24, 16 and 21). The plots are in Figs. 23, 24 and 25.

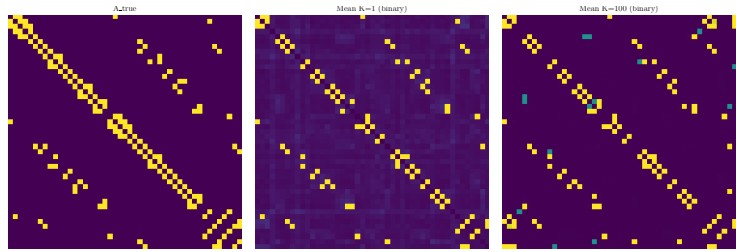

Figure 23: Graph reconstruction for sample 24, median raw

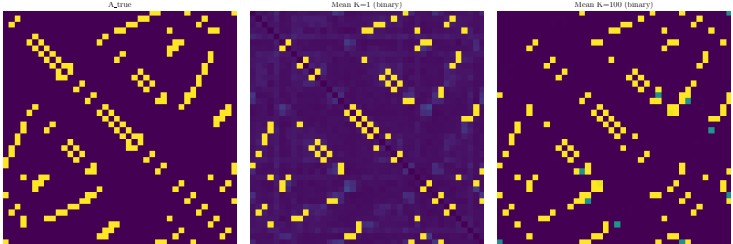

Figure 24: Graph reconstruction for sample 16, median raw

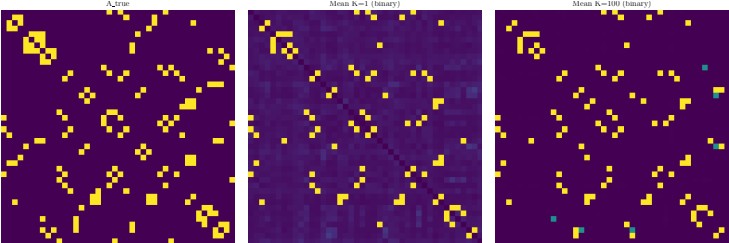

Figure 25: Graph reconstruction for sample 21, median raw

**Individual samples for each graph.** Lastly, we show the raw versions of different realizations (individual samples) for each graph. We compute this for 3 graphs in the test set (24, 16 and 21). Interestingly, the samples for $K = 100$ are more diverse (similar to the case of images in Ohayon et al. (2025)); this diversity explains why the raw mean in Figs. 20, 21 and 22 have values that are not exactly 0 or 1 (which indicates that the samples disagree on these entries rather than sharing a single structure).



Figure 26: Individual samples for $K = 1$ and for sample 24

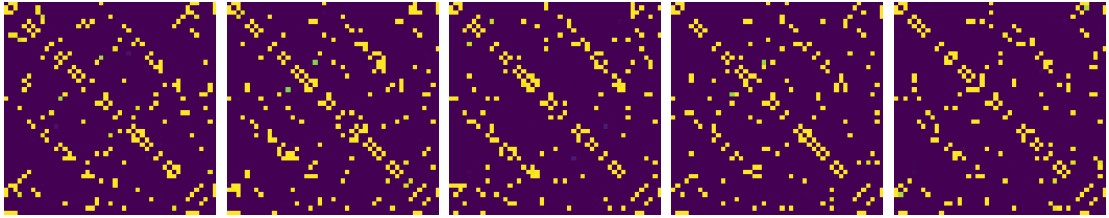

Figure 27: Individual samples for $K = 100$ and for sample 24

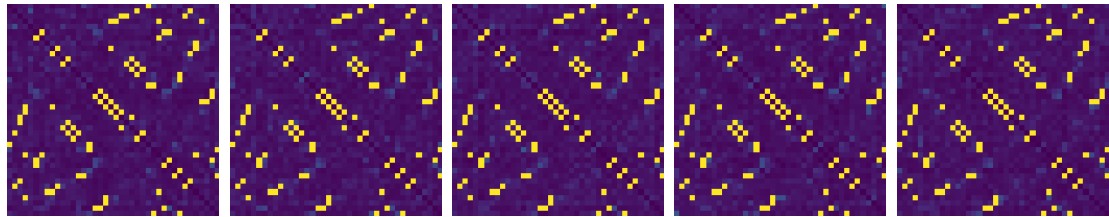

Figure 28: Individual samples for $K = 1$ and for sample 16

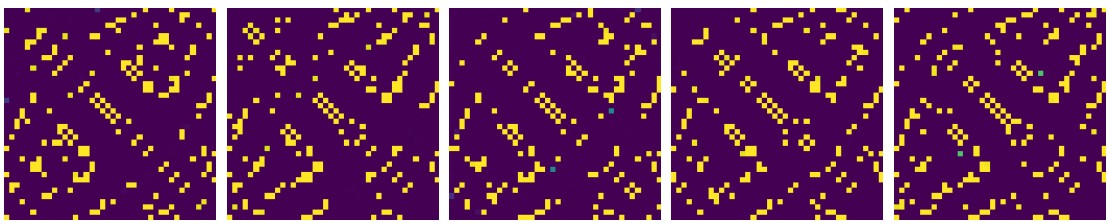

Figure 29: Individual samples for $K = 100$ and for sample 16



Figure 30: Individual samples for $K = 1$ and for sample 21

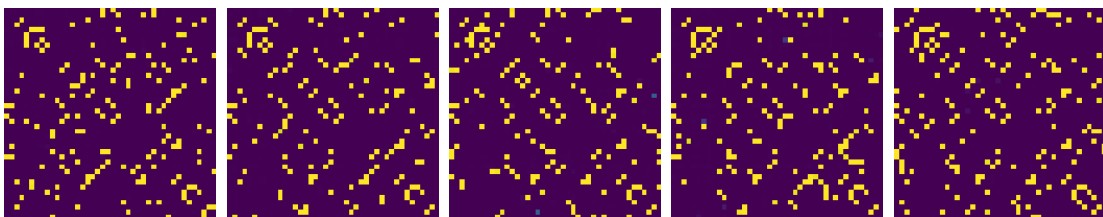

Figure 31: Individual samples for $K = 100$ and for sample 21

## F.2 Intermediate adjacency matrices

In Figs. 32- 34, we show visualizations of the diffusion trajectory of the link-prediction sampler by "snapshotting" the predicted adjacency matrix at steps 0 to 100 in 10-step-increments of a sample path with 100 total steps. Each panel shows the raw adjacency values, zeroed on the diagonal, rendered with the colormap with black associated to 0 and yellow to 1. The bottom-rightmost panel is the ground-truth adjacency for comparison.

To see the sampling process, progress from left-to-right and top-to-bottom shows how the sampler denoises toward the final reconstruction (steps=100). From these images we could see the smooth transitions along the full reconstruction trajectory of PIFM.

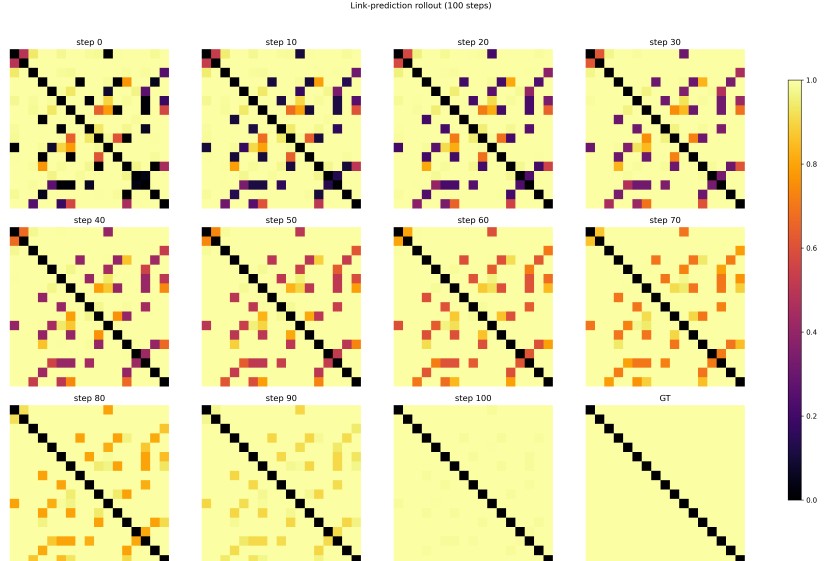

Figure 32: Visualization of IMDB 50% drop rate reconstruction. (Graph 1)

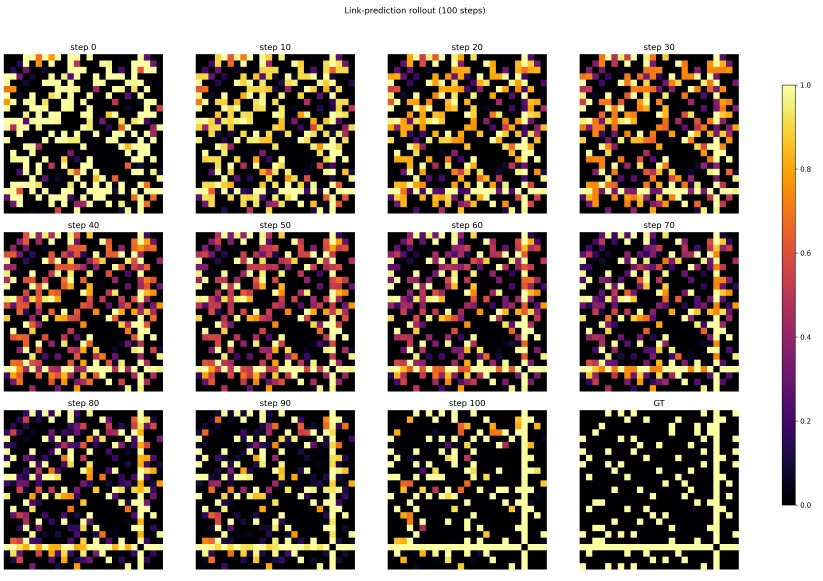

Figure 33: Visualization of IMDB 50% drop rate reconstruction. (Graph 2)

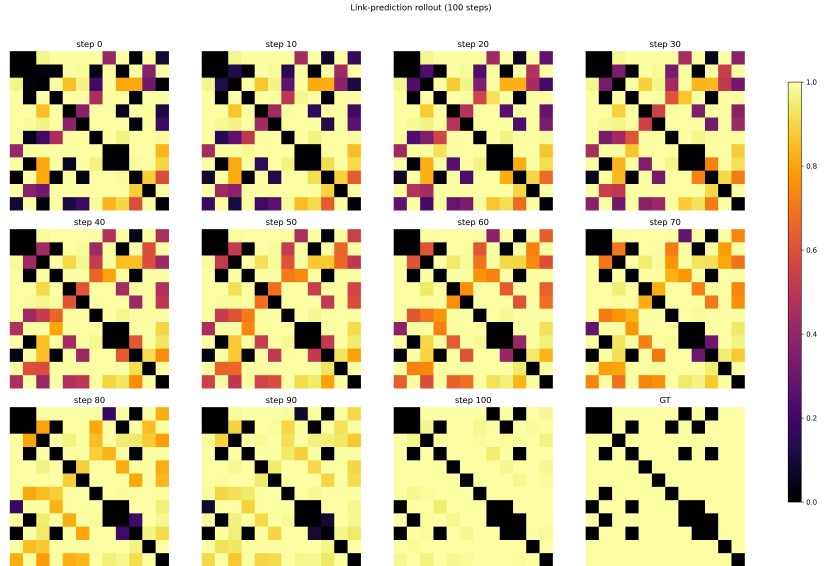

Figure 34: Visualization of IMDB 50% drop rate reconstruction. (Graph 3)

