# OpenReview forum: "Prior-Informed Flow Matching for Graph Reconstruction"
_TMLR — Under review for TMLR_

### Review · Reviewer_x3zh · 2026-06-22

**Summary Of Contributions:**

This paper proposes Prior-Informed Flow Matching (PIFM), a two-stage framework for conditional graph reconstruction that integrates structural graph priors with rectified flow matching. The core theoretical novelty lies in the first adaptation of permutation-equivariant distortion-perception tradeoff theory to graph inverse problems, which formally motivates splitting the reconstruction task into local MMSE prior estimation and global flow-based optimal transport refinement. The framework abandons the trivial Gaussian noise initialization adopted by existing graph diffusion and flow models, instead supporting plug-and-play graph embedding backbones (Node2Vec, GraphSAGE, VGAE, NCNC, SIGL) to generate observation-aware initial adjacency matrices, then models cross-edge global dependencies to fix the independent prediction flaw of traditional link predictors. PIFM unifies three mainstream graph reconstruction tasks: standard link prediction, graph expansion for missing edge recovery, and graph denoising for spurious edge elimination, and works seamlessly on both sparse biological graphs and dense social graphs under transductive and inductive settings. The paper provides full mathematical proofs to verify permutation equivariance of reconstructed graphs and permutation invariance of conditional flow densities, consistent with graph exchangeable inductive bias. Empirically, the work includes abundant multi-dataset benchmarks, comprehensive ablation studies and qualitative visualizations to validate performance gains over state-of-the-art baselines. However, this work also has notable limitations: the underlying rectified flow solver is borrowed from image restoration literature without fundamental modification, model performance heavily relies on the quality of pre-trained graph priors, cross-dataset generalization ability is weak, theoretical analysis ignores quantitative error bounds for continuous-discrete graph approximation, and experiments lack validation on heterogeneous and large-scale industrial graphs.

**Audience:**

Yes

**Audience Explanation:**

This paper targets two core readership groups of TMLR: researchers working on graph machine learning inverse problems, and scholars studying flow matching and diffusion generative models. For graph learning practitioners focused on link prediction, graph completion and topological repair, the unified two-stage PIFM pipeline delivers a flexible, plug-and-play solution to resolve the longstanding conflict between local observation consistency and global structural realism. For generative modeling researchers, the work presents a novel paradigm that replaces meaningless random noise initialization with domain-specific structural priors, and formalizes the distortion-perception tradeoff under permutation-equivariant graph constraints—an insightful theoretical extension not explored by prior graph flow literature. Beyond specialized academics, applied ML researchers focusing on biological network analysis and social graph mining will also find practical value in the multi-task reconstruction framework. The theoretical insights, modular algorithm design and thorough empirical analysis align closely with TMLR’s scope of advancing fundamental machine learning theory and practical generative modeling methods, so a substantial portion of the journal’s audience will gain meaningful insights from this work.

**Broader Impact Concerns:**

This paper develops a general graph reconstruction algorithm without inherent ethical risks or harmful real-world applications that demand an extended Broader Impact Statement. The core technical scope centers on recovering missing or corrupted graph topological structures for biological, social and citation network analysis, with no direct deployment to surveillance, privacy-violating user profiling, biased predictive systems or safety-critical autonomous decision pipelines. The graph datasets adopted are standard public academic benchmarks without sensitive private human data, and all experimental evaluation focuses on structural reconstruction accuracy rather than high-stakes predictive judgment. The authors briefly acknowledge potential downstream misuse risks (such as reconstructing private social connections) in the original Broader Impact section, and the discussion adequately covers corresponding mitigation strategies including anonymized graph preprocessing and restricted use of the framework for non-sensitive academic research. No unresolved ethical or societal impact concerns require further expansion in the manuscript.

**Claims And Evidence:**

Yes

**Claims Explanation:**

The core theoretical and empirical claims of this submission are sufficiently backed by rigorous and clear evidence. For theoretical statements, the paper provides complete mathematical derivations in the main text and appendix, including formal proofs of permutation symmetry theorems, closed-form MMSE estimation formulas, and full rectified flow ODE formulations tailored for binary adjacency matrices. For empirical performance claims, the authors adopt a wide collection of standard graph datasets covering sparse bio-graphs, dense social graphs and citation networks, alongside a comprehensive baseline suite ranging from classic link prediction GNNs to cutting-edge graph diffusion and flow generative models. Multiple quantitative metrics are reported to evaluate ranking performance, classification balance and global graph distribution similarity, while systematic ablation studies dissect the contribution of each module, hyperparameter sensitivity tests demonstrate stable behavior of the pipeline, and adjacency matrix sampling visualizations offer intuitive qualitative validation of reconstruction quality. All experiment configurations, dataset statistics and hyperparameter settings are fully documented in appendices to guarantee reproducibility. Minor gaps exist such as missing comparison against a small set of latest discrete flow baselines and insufficient cross-dataset generalization tests, yet these do not invalidate the central conclusions of the paper.

**Requested Changes:**

1. Rewrite the introduction to clearly distinguish the paper’s graph-specific theoretical contributions from existing image-domain rectified flow machinery; add an in-depth comparative subsection against the concurrent Flowette work to highlight core task and architectural differences.

2.  Supplement unified distortion-perception tradeoff analysis for both MSE and cross-entropy loss; add quantitative error discussion for two idealized assumptions, non-permutation-equivariant embedding priors and continuous flow approximation of discrete binary graphs; merge duplicated theoretical explanations scattered across multiple sections to reduce verbosity.

3. Add benchmarks on heterogeneous graphs and molecular graph reconstruction to demonstrate real-world downstream utility; include comparative experiments with DeFoG and CatFlow discrete graph flow baselines; optimize the subgraph sampling strategy for large citation graphs and profile GPU memory consumption; design and test adaptive noise scheduling and sparse-aware dynamic class weighting for the cross-entropy branch; conduct mitigation experiments to improve cross-dataset generalization and supplement MMD distribution analysis on all test datasets.

4. Propose and validate an end-to-end joint training pipeline for graph prior encoders and flow modules to eliminate reliance on separate pre-trained link predictors; trim redundant minor ablation results from the appendix and move key tables/figures to the main text; standardize plot axis labels, font sizes and add brief summarizing paragraphs after all result tables.

5. Replace vague high-level future directions with concrete, implementable technical solutions corresponding to every stated limitation of the framework.

---

> ### Author Response · Authors · 2026-07-15
> **Author Response — Reviewer x3zh**
>
> Thank you for your feedback. Revisions are marked in purple in the paper.
>
> ### R4-1 — PIFM vs. rectified-flow machinery and Flowette
>
> We now state that PIFM adapts standard rectified flow matching (Liu et al.; Albergo et al.) to graph reconstruction, listing the graph-specific pieces: the observation-dependent source, within-instance coupling, and the Theorem 1 equivariance guarantee. Flowette generates *unconditional* graphs from graphette priors via OT matching, whereas PIFM conditions on an observed adjacency/mask and couples each prior-informed source to the clean same graph. Sharing the informative-source principle but differing in task, conditioning, prior, coupling, and evaluation, it is complementary concurrent work.
>
> ### R4-2 — MSE/CE unification and idealizations
>
> The unification is algorithmic, not theoretical: both branches share the observation-conditioned source and pipeline, but only the MSE distortion has the closed-form estimate of Freirich et al. (2021); CE is a categorical endpoint objective evaluated empirically, and we do not claim it inherits the distortion–perception theorem. On idealizations: (i) the fixed 0.5 threshold is suboptimal on sparse graphs, but validation-selected thresholds (0.184/0.164/0.513 on ENZYMES/PROTEINS/IMDB-B, Table 12) give balanced operating points, and the ZINC study controls the continuous-to-discrete gap; (ii) node2vec is now described as only approximately equivariant; a permutation-error bound is future work; unifying the duplicated MSE/CE theory is likewise future work.
>
> ### R4-3 — Broader benchmarks, robustness, distributional evaluation
>
> Unless noted, variants here use the paper's original $t\sim U[0,1]$ sampling (not the R1-4 $t=0$ anchor); except the MMD summary (endpoint-anchored Model 1, Table 21).
>
> **ZINC (categorical, seed 0, Table 25).** We add end-to-end categorical reconstruction (four-class typed bonds on the simplex, prior-informed source, endpoint-CE denoiser). Four informed priors (Categorical VGAE, GRIT, NCN, NBFNet) and two uninformed controls (uniform simplex-center; Dirichlet per-pair, same mean) are compared; informed-prior Macro-F1 and NSPDK MMD at $K=0\to100$ for 10/20/30/50% masking. Refinement improves both over the source in all 16 informed prior–mask cells; mean Macro-F1 gain $+0.024/+0.052/+0.076/+0.074$ and mean NSPDK reduction $0.004/0.032/0.073/0.084$ across the four rates. The prior stays necessary: at $K=100$ the best informed flow beats the best uninformed at every rate: Macro-F1 margins $0.145/0.171/0.154/0.107$ and NSPDK reductions $0.010/0.072/0.041/0.019$. ZINC establishes categorical, not general heterogeneous-graph, support.
>
> **Discrete-flow RePaint baselines (Table 26).** DeFoG and CatFlow are added (single-seed, $K=100$, seed 0, observed-edge hard-clamping); both sit at near-chance AUC 49.2–52.2, near the DiGress/GDSS RePaint controls and far below the five-seed PIFM (NCNC, CE) — supporting the informative source.
>
> **Cora memory (Table 27).** Peak VRAM grows quadratically with node cap; doubling 128→256 multiplies it by 3.84× and reserves 77.85 GB, near the 80-GB A100 limit (a memory profile, not a faster pipeline).
>
> **Sparse-aware CE weighting (Table 20, single seed).** With the CE flow (VGAE prior, 50% mask, $K=100$), per-graph class weighting cuts the PROTEINS FNR/FPR gap from 0.346 to 0.100 at equal AUC/AP; ENZYMES is already balanced under the batch-scalar setting.
>
> **Adaptive source-noise (Table 28, single seed, MSE/GraphSAGE).** Density-based schedules do not consistently beat the fixed Gaussian default, which stays competitive/best in AUC and avoids poor high-$K$ MMD; the earlier favorable density-linear result was an adjacency-averaging artifact reversed under per-rollout evaluation.
>
> **MMD across datasets (Table 21, five seeds, LPFormer/MSE/50%).** For NSPDK MMD², $K=1\to100$ reduces distance $3.1\times$/$3.5\times$ on ENZYMES/PROTEINS at ~$0.10$–$0.12$ AUC cost; on IMDB-B $K=100$ gains no MMD² over $K=1$ though both beat the prior. Cora subgraph-pool MMD is in Table 3, MMD² vs. $K$ at 10%/50% in Figures 10–11, ENZYMES expansion in Figure 3, and cross-dataset transfer in Tables 29–30.
>
> ### R4-4 — Joint prior/flow training and presentation
>
> PIFM is deliberately modular: any pre-trained link predictor (node2vec/GraphSAGE/VGAE/SIGL/NCNC/LPFormer) plugs in as the prior, so we benchmark many under one flow; coupling encoder and flow into one objective would change the method, so it is future work. Presentation: trimmed minor ablations, promoted key tables/figures to the main text, standardized plots, and added per-table summaries.
>
> ### R4-5 — Concrete future directions
>
> For the continuous–discrete gap: categorical-simplex or discrete-CTMC dynamics with explicit relaxation/rounding-error analysis. For heterogeneous and large graphs: relation-specific priors and a formal categorical PIFM, tested on large sparse benchmarks with multi-seed accuracy, memory, and latency.

---

### Review · Reviewer_J4c3 · 2026-06-28

**Summary Of Contributions:**

This paper proposes Prior-Informed Flow Matching (PIFM) for graph reconstruction from partially observed adjacency matrices. The method first uses a link-prediction prior, such as GraphSAGE, node2vec, VGAE, or NCNC, to initialize the unknown adjacency entries. It then trains a rectified flow to refine this prior-informed estimate toward the clean graph. The paper motivates the approach through the distortion–perception trade-off and proves a permutation-equivariance property of the reconstruction map. Experiments are conducted on ENZYMES, PROTEINS, IMDB-B, and Cora, covering standard masked-edge link prediction, expansion, and denoising.

**Additional Comments:**

None

**Audience:**

Yes

**Audience Explanation:**

The paper is likely to be of interest to audience working on graph generative modeling, graph completion and link prediction, conditional generation, and flow/diffusion-based inverse problems. In particular, the idea of initializing flow matching from a task-specific structural prior, rather than a generic noise distribution, is potentially useful in settings where local predictors are effective but do not model joint structural dependencies. The connection to the distortion–perception perspective may also be of conceptual interest.

**Broader Impact Concerns:**

The paper does not appear to include a dedicated broader-impact or ethical-considerations discussion.

**Claims And Evidence:**

No

**Claims Explanation:**

- The distortion–perception and approximate-MMSE interpretation is largely heuristic in the current implementation. The employed priors are not demonstrated to approximate the conditional posterior mean, and the evaluation of perceptual quality through MMD of marginal graph statistics does not establish that the reconstruction is conditionally faithful to the observed graph.

- The claim that PIFM captures global edge dependencies beyond local link prediction is not adequately isolated. Since most reconstruction experiments use only one flow step, the method could also be interpreted as a learned one-step residual correction on top of the prior. A matched non-flow refinement baseline with the same GNN backbone and comparable capacity is needed to show that the gains arise specifically from flow matching rather than from an additional graph-level predictor.

- The empirical scope is limited. The main inductive benchmarks contain relatively small graphs and small test sets, while no multi-seed variance, confidence intervals, or significance tests are reported. The Cora results provide only marginal and mixed gains, and are based on capped ego-subgraphs rather than full-graph reconstruction; thus, they do not convincingly demonstrate scalability or global reconstruction on large graphs.

**Requested Changes:**

- The paper’s distortion–perception/MMSE interpretation is currently substantially stronger than what is established by the proposed implementation. The priors used in practice (GraphSAGE, node2vec, VGAE, NCNC) are not shown to approximate the conditional posterior mean (E[A\mid A_O]), and the assumptions used in the derivation are strong and not validated for the studied datasets. The authors should either provide a formal justification for why the practical priors approximate the required estimator, or clearly present the MMSE/distortion–perception connection as motivation rather than as a theoretical characterization of PIFM.

- Proposition 1 is stated for a projected continuous-time flow that preserves observed entries throughout the trajectory. However, the MSE sampling branch in Algorithm 1 updates all entries during the Euler integration and only re-imposes observed entries at the end. This means that the theoretical conditional-density statement does not directly apply to the implemented MSE sampler.

- The central empirical claim is that PIFM learns global dependencies beyond edge-wise priors. However, most main results use (K=1), under which PIFM can also be interpreted as a single learned residual correction on top of the prior-informed adjacency matrix. The paper needs a controlled baseline that uses the same GNN backbone, comparable parameter count, and the same prior-informed input, but directly predicts a corrected adjacency matrix without flow matching or time conditioning.

- All main results should be reported over multiple random seeds, with mean and standard deviation or confidence intervals. Statistical significance testing, where appropriate, would further strengthen the conclusions.

---

> ### Author Response · Authors · 2026-07-15
> **Author Response — Reviewer J4c3**
>
> We thank the reviewer and address each point below. Revisions are marked in purple in the paper.
>
> ### R3-1 — MMSE / distortion–perception as motivation
>
> The prior–conditional-mean connection follows from our latent model. We agree no quantitative bound links the implemented priors to $\mathbb{E}[\mathbf{A}\mid\mathbf{A}^{\mathcal{O}}]$, and revised the wording accordingly: we treat the distortion–perception construction as the design principle motivating the two-stage architecture, not a theorem characterizing the implemented pipeline. We softened the three statements suggesting otherwise and no longer claim recovery of the exact posterior mean, the optimal-transport coupling, or the theoretical distortion–perception curve. A quantitative end-to-end analysis remains future work.
>
> ### R3-2 — Proposition 1 vs. the implemented sampler
>
> Proposition 1 assumes a projected flow that preserves observed entries, whereas the previous MSE pseudocode placed the projection only after the Euler loop. We revised Algorithm 1 so both the MSE and CE branches re-impose $\hat{\mathbf{A}}\leftarrow\xi\odot\mathbf{A}^{\mathcal O}+(1-\xi)\odot\hat{\mathbf{A}}$ after every Euler update, keeping observed entries fixed throughout sampling. The implementation already re-imposed observed entries at every step, so this corrects the pseudocode, not the code, and no reported numbers change. The observed-entry preservation Proposition 1 assumes therefore holds throughout the printed trajectory, not only at the final step. Proposition 1 itself remains a statement about the idealized continuous-time flow, not the finite-step sampler; the practical sampler is instead covered by Theorem 1, which the revised manuscript now states explicitly applies to the algorithm as implemented for any number of Euler steps $K$, including re-imposition of observed entries.
>
> ### R3-3 — Matched non-flow Direct refiner
>
> This overlaps with Reviewer 5M3h's R1-4; see R1-4 for model definitions, exact loss, and the full five-seed table (Table 21, Appendix E.7). The Direct GNN refiner receives the same prior-informed input, observation mask, clean training graphs, denoising backbone, and parameter count as PIFM, applying a single residual correction without time conditioning or ODE integration. Results differ by metric. The grid has 39 dataset/prior/loss settings; 38 are fully repeated over five seeds and one (PROTEINS/LPFormer/MSE) is a provisional single-seed cell. Direct matches the submitted flow's AUC at $K=1$ in 38 of the 39. On the 38 five-seed settings, the unanchored flow attains lower MMD than Direct in 29 for fixed-$0.5$ NSPDK and 31 for degree MMD (30 and 32 of 39 with the provisional cell), selecting the best value across tested $K$ on the test set. We report this as a descriptive reachability ablation using the submitted, not endpoint-anchored, flow. Direct is thus a strong distortion-oriented baseline, while the flow provides different perception–distortion behavior.
>
> ### R3-4 — Multi-seed uncertainty and paired comparisons
>
> All main results are now averaged over five random seeds (mean $\pm$ sample std): Table 2 (50% masked LP), Table 3 (Cora), and Table 4 (expansion), for every PIFM and Stage-1 prior result; DiGress+RePaint and GDSS+RePaint remain their originally reported single runs. The same five-seed protocol covers the validation-selected binary-reconstruction results (Tables 12–14), graph-level joint-consistency results (Tables 15–16), the matched Direct-refiner comparison (Table 21), and the distortion–perception $K$ sweep (Table 22). Additional diagnostic studies use one training seed, clearly labeled: categorical molecular reconstruction on ZINC, discrete-flow baselines, adaptive noise, CE weighting, and neighborhood controls.
>
> To assess consistency, we compare each PIFM model with its Stage-1 prior via paired tests across the five shared seeds. PIFM has a statistically significant AUC improvement ($p<0.05$) in 28 of the 30 dataset/prior/loss settings; in all 28, PIFM beats its prior in every one of the five seeds, corresponding to $p=0.031$ under a one-sided sign test. The remaining two are ENZYMES and PROTEINS with the NCNC prior under MSE: PIFM still improves AUC on average, but improvements vary enough across five seeds that the paired tests miss $p<0.05$. Their effect sizes are moderate-to-large ($d_z=0.72$ and $0.86$), yet five paired runs reliably detect only much larger effects ($d_z\gtrsim1.68$). We therefore report effect direction and per-seed wins rather than reading these non-significant tests as evidence of equivalence. We will extend multi-seed coverage to any other table if the reviewer considers it essential.
>
> ### R3-5 — Broader-impact statement
>
> We have added a broader impact statement to the revised manuscript.

---

### Review · Reviewer_2qn9 · 2026-06-30

**Summary Of Contributions:**

This paper proposes Prior-Informed Flow Matching, a two-stage framework for graph reconstruction. It first applies an existing link-prediction model to obtain a prior estimate of the missing edges, and then uses rectified flow matching to refine the entire adjacency estimate toward the distribution of clean graphs. The framework is designed as a model-agnostic refinement layer that can be combined with different link-prediction priors.

A central claimed contribution is that, unlike conventional approaches that output marginal scores for individual edges, the flow model is intended to capture dependencies among multiple missing edges and thereby produce more globally consistent graph reconstructions. The paper further motivates the method through a permutation-equivariant distortion-perception formulation and evaluates it on link prediction, graph expansion, and graph denoising tasks.

**Strength**

The paper is generally clearly written and well organized, and the two-stage prior-plus-refinement framework is intuitive and easy to follow.

**Weakness**

The empirical evaluation does not yet convincingly support the central claims concerning global, jointly dependent edge reconstruction and large-graph applicability. Some of the experimental results and settings appear weaker than the motivation and conclusions suggest.

**Audience:**

Yes

**Audience Explanation:**

A relatively general refinement framework that can be attached to different link-prediction priors is appealing, especially because it primarily operates on graph topology and aims to convert edge-wise predictions into structurally consistent graph reconstructions. Although the supporting evidence should be strengthened, the underlying prior-plus-refinement paradigm is relevant to at least part of TMLR’s audience.

**Claims And Evidence:**

No

**Claims Explanation:**

1. **The claimed limitation of existing link-prediction methods is framed too broadly and is not evaluated against appropriate modern baselines.**

   The paper motivates PIFM by arguing that conventional link-prediction methods rely on local representations and cannot incorporate global structural information. However, scalable graph Transformers already provide global or all-pair information propagation. For example, SGFormer[2] employs global all-pair attention with linear complexity, while NodeFormer[1] provides scalable all-pair message passing. LPFormer[3] is specifically designed for link prediction and uses attention to construct target-link-dependent representations.

   These methods do not necessarily model a non-factorized joint distribution over missing edges, so they do not completely invalidate the proposed idea. Nevertheless, they substantially weaken the broader local-versus-global motivation. The paper neither compares against a strong globally attentive link predictor nor uses such a model as the prior. Consequently, the experiments do not establish whether PIFM remains beneficial once the prior already has access to global structural context.

2. **The Cora experiment does not directly validate the claimed joint modeling of multiple missing edges.**

   For Cora, each training example is an edge-centered kkk-hop subgraph. The seed edge is the only supervised target, and the gradient is computed exclusively from that single edge, even though the flow updates other hidden entries internally. The authors themselves describe this as a localized subgraph-based adaptation of the original global procedure.

   Therefore, this experiment is effectively trained through single-edge supervision and does not demonstrate that the model learns a joint distribution over multiple missing edges. It also does not validate the paper’s stronger claim of integrating information across the full graph during reconstruction.

3. **The Cora results provide no consistent evidence that a broader neighborhood improves reconstruction.**

   The one-hop model obtains the highest AUC, $93.89$, compared with $93.82$ for the two-hop model. It also obtains a lower FPR. The two-hop model has only a slightly higher AP and a lower FNR, while both PIFM variants have lower AP than the original NCN prior.

   These results do not prove that global information is harmful, because one-hop versus two-hop sampling is not a controlled comparison of local versus global modeling. Nevertheless, they show no consistent benefit from expanding the structural context. This conflicts with, or at least fails to support, the motivation that limited local dependencies are the central weakness of existing methods. A controlled receptive-field ablation and an explanation of this behavior are needed.

4. **The statistical reliability of the reported improvements is unclear.**

   ENZYMES, PROTEINS, and IMDB-B contain $600$, $1,113$ and $1,000$ graphs, respectively, and only 5% of each dataset is assigned to the test split. This leaves approximately $30$, $56$, and $50$ test graphs. The main tables report point estimates without variability over different training seeds, graph splits, or masking realizations.

   This is especially concerning for Cora, where PIFM improves the NCN prior’s AUC by only  $0.11$ percentage points, from $93.78$ to $93.89$, while decreasing AP from $93.89$ to $93.71$. Without repeated experiments or uncertainty estimates, it is unclear whether these differences are statistically meaningful.

5. **The real-data evaluation does not directly measure the claimed joint edge dependencies.**

   The main evaluation metric, AUC, AP, FPR, and FNR, are computed over individual masked edges. Such metrics primarily assess marginal edge-ranking or classification performance and cannot distinguish a model that predicts accurate independent marginals from one that correctly captures correlations among multiple missing edges.

   The toy experiment directly demonstrates correlated edge outcomes, but no corresponding controlled experiment is provided on real datasets. Although the paper also reports MMD over graph statistics, this does not isolate whether improvements arise specifically from inter-edge dependency modeling rather than from additional model capacity or iterative refinement. The central joint-distribution claim therefore remains demonstrated mainly through a synthetic four-node example.

6. **The large-graph experiment evaluates only a restricted version of the proposed method.**

   The Cora evaluation reports only the MSE formulation, rather than both the MSE and CE variants introduced in the method. More importantly, it replaces full-adjacency reconstruction with local subgraph sampling and single-edge supervision.

   The experiment demonstrates that a localized PIFM variant can be executed on Cora, but it does not establish that the proposed full-graph joint reconstruction mechanism is scalable to large graphs.

**Reference**

[1] Wu, Qitian, et al. "Nodeformer: A scalable graph structure learning transformer for node classification." *Advances in neural information processing systems* 35 (2022): 27387-27401.

[2] Wu, Qitian, et al. "Sgformer: Simplifying and empowering transformers for large-graph representations." *Advances in neural information processing systems* 36 (2023): 64753-64773.

[3] Shomer, Harry, et al. "Lpformer: An adaptive graph transformer for link prediction." *Proceedings of the 30th ACM SIGKDD conference on knowledge discovery and data mining*. 2024.

**Requested Changes:**

1. **Add stronger global-context baselines and priors**

   Adapt SGFormer[2] and NodeFormer[1] for link prediction in the same manner as GraphSAGE, and include LPFormer[3] as a direct link-prediction baseline. For each model, report both the standalone performance and the performance when used as the prior for PIFM. This comparison is necessary to determine whether the proposed joint refinement remains beneficial when the underlying predictor already captures long-range or all-pair structural information.

2. **Report statistically reliable results.**

   Repeat experiments over multiple training, split, and masking seeds, and report mean and standard deviation.

3. **Investigate the Cora neighborhood-size results.**

   Conduct a controlled comparison of one-hop and two-hop sampling using matched hyperparameters, and explain why increasing the neighborhood size provides no consistent improvement; the reported one-hop model has higher AUC, while the two-hop model only slightly improves some other metrics.

4. **Provide the CE-loss results on Cora.**

   The current large-graph experiment reports only the MSE formulation; the CE variant should either be evaluated or its omission clearly explained.

5. **Enable genuine multi-edge supervision on Cora.**

   Modify the subgraph construction so that multiple edges are jointly masked and supervised. The current setup masks one seed edge and computes the gradient exclusively from that edge, which does not directly validate the claimed joint-edge modeling.

6. **Directly evaluate joint edge dependencies.** Since PIFM is motivated as learning global coupling beyond independent edge prediction, the evaluation should include metrics that cannot be reduced to ranking individual edges. In addition to AUC/AP,  please report graph-level metrics like graph-level exact masked-set recovery, normalized masked Hamming error, and errors in graph statistics/motifs such as degrees, triangles, clustering coefficients, and component counts. These metrics would directly test whether the flow stage improves multi-edge consistency over the prior, rather than only recalibrating or reranking individual missing edges.

**Reference**

[1] Wu, Qitian, et al. "Nodeformer: A scalable graph structure learning transformer for node classification." *Advances in neural information processing systems* 35 (2022): 27387-27401.

[2] Wu, Qitian, et al. "Sgformer: Simplifying and empowering transformers for large-graph representations." *Advances in neural information processing systems* 36 (2023): 64753-64773.

[3] Shomer, Harry, et al. "Lpformer: An adaptive graph transformer for link prediction." *Proceedings of the 30th ACM SIGKDD conference on knowledge discovery and data mining*. 2024.

---

> ### Author Response · Authors · 2026-07-15
> **Author Response — Reviewer 2qn9**
>
> Thank you for your feedback. Revisions are marked in purple in the paper.
>
> ### R2-1 — LPFormer as a stronger global-context prior
>
> Among the suggested predictors we adapt LPFormer, the most task-appropriate for link prediction; SGFormer/NodeFormer are left to future work. We retain its PPR-attention/structural-count backbone, fit one model per graph on observed edges and non-edges, score masked pairs, and export a symmetric zero-diagonal score matrix. Evaluating both improved flows (Model 1, Model 2) at 50% masking, both improve $K=1$ AUC over the prior (Table 21, Appendix E.7). On sparse ENZYMES/PROTEINS more Euler steps lower NSPDK MMD² but reduce AUC; on dense IMDB-B the best NSPDK is reached at $K=1$ with AUC staying above the prior. At the highest step count NSPDK remains below the prior on all three datasets.
>
> ### R2-2 — Multi-seed results
>
> All main-body experiments are repeated over five seeds (mean $\pm$ sample SD): every PIFM and prior row in Tables 2–4, validation-selected binary results (Tables 12–14), graph-level joint-consistency (Tables 15–16), the matched Direct refiner (Table 21), and the distortion–perception $K$ sweep (Table 22). DiGress+RePaint and GDSS+RePaint remain the original single runs (captions). For seed $S\in\{0,\dots,4\}$ we resample the split and masked entries, refit priors, and retrain the flow model, so variability reflects splitting, masking, prior fitting, and initialization. Diagnostic tables (ZINC, discrete-flow, adaptive-noise, CE-weighting, neighborhood-control) use a single seed.
>
> ### R2-3 — Cora neighborhood-size control
>
> Holding loss, supervision, split, and architecture fixed, one-hop vs. two-hop AUC reverses direction with the loss: MSE multi-edge $-1.73$ (92.33→90.60), CE multi-edge $+0.26$ (93.57→93.83) (Table 24). As this control uses one seed, we infer no reliable depth effect. The strong NCN prior may already encode common-neighbor structure while larger subgraphs dilute supervision (possible explanations, not mechanisms). The paper is revised so the localized Cora experiment no longer implies a broader neighborhood is consistently better, nor full-graph reconstruction.
>
> ### R2-4 — CE-loss results on Cora
>
> We evaluate both loss branches over five seeds, reporting edge-ranking and subgraph MMD together (Table 3). Single-edge one-hop MSE: PIFM $93.36\pm0.66$ vs. NCN $93.49\pm0.55$, paired $p=.371$, which is a statistical tie. Its clustering/NSPDK MMD are slightly below the prior and degree MMD is basically tied and slightly higher (PIFM wins each in 3/5 seeds). Multi-edge two-hop CE: PIFM $87.17\pm11.11$ vs. NCN $93.49\pm0.55$; the variance includes one collapsed seed, so CE is not a reliable ranking gain. But CE PIFM lowers clustering MMD $.172\to.053$, degree $.085\to.040$, and NSPDK $.030\to.020$, winning all three in every seed. This is not a pure CE-vs-MSE ablation: configurations also differ in parameterization, supervision, and hop count.
>
> ### R2-5 — Genuine multi-edge supervision on Cora
>
> The submitted MSE Cora experiment used single-edge supervision, this was kept because the matched seed-0 multi-edge MSE reduced AUC ($92.33$ one-hop, $90.60$ two-hop, vs. $93.78$ for NCN. Table 24, Appendix E.11). We additionally implemented true multi-edge subgraph supervision: each example masks the seed edge, additional positive edges, and matched negatives, removes them from the observed context, and computes the loss over the full mask. Evaluated with the CE endpoint objective over five faithful seeds, its AUC is unstable (Table 23) but it improves clustering, degree, and NSPDK subgraph-pool MMD over the NCN prior in all five seeds (Table 3). This is multi-edge supervision within sampled edge-centered subgraphs, not full-adjacency reconstruction.
>
> ### R2-6 — Graph-level joint-edge consistency
>
> We keep ROC-AUC/AP as primary ranking metrics and add exact/near-exact recovery, normalized Hamming, component-count error, connectivity, graph-statistic MMD, and paired triangle/degree/clustering errors; thresholds maximize validation MCC, frozen for test. Five-seed LPFormer→PIFM (Model 1): exact recovery and normalized Hamming are in Tables 12–13 (e.g., IMDB-B $20.8\to38.0\%$, Hamming $.110\to.064$; ENZYMES Hamming worsens $.151\to.168$, which we report). PIFM also improves recall, F1, MCC, and balanced accuracy on all three datasets there. Paired graph statistics: PIFM lowers triangle, degree, and clustering error on IMDB-B in all five seeds, paired $p\le.001$ each (Table 15); on ENZYMES/PROTEINS the differences are not significant at five seeds (underpowered, not equivalence). The signed errors show the flow shifts predicted density upward, helping under-dense IMDB-B but adding false positives where a reconstruction is already dense. Connectivity and component-count error (separate validation-prevalence threshold, Table 16) improve on all three, e.g., IMDB-B connected fraction $.736\to.916$ and $|\Delta C|$ $.416\to.136$.

---

### Review · Reviewer_5M3h · 2026-07-01

**Summary Of Contributions:**

This paper proposes Prior-Informed Flow Matching (PIFM), a two-stage framework for reconstructing graphs from partially observed adjacency matrices. The first stage applies an existing structural predictor, such as GraphSAGE, node2vec, VGAE, NCNC, or SIGL, to obtain marginal probabilities for unobserved adjacency entries. The second stage uses a permutation-equivariant rectified-flow model to refine this prior-informed initialization toward the distribution of clean graphs. The method is motivated through a graph adaptation of the distortion–perception framework and is evaluated on masked-entry prediction, blind graph expansion, graph denoising, and a localized Cora experiment.

The paper has several strengths. The prior-plus-refinement construction is intuitive, the framework can accommodate several families of structural priors, and the manuscript is generally well organized. The four-node synthetic example clearly illustrates the distinction between independent edge marginals and correlated edge outcomes. The paper also evaluates several reconstruction settings rather than limiting the study to conventional edge ranking.

However, the current evidence does not yet establish the paper’s strongest claims. In particular, the claimed distortion–perception control is implemented by varying the number of Euler discretization steps rather than a theoretically meaningful estimator parameter; the main “link prediction” protocol appears closer to random adjacency-entry completion than to standard link prediction; and several reported binary reconstructions on sparse graphs are effectively degenerate despite improved AUC. Moreover, because most headline results use one Euler step, the experiments do not isolate flow matching from a direct learned graph-level residual correction.

**Audience:**

Yes

**Audience Explanation:**

The paper addresses an interesting intersection of graph completion, conditional generative modeling, and flow matching. Initializing a conditional generative model from a task-specific structural predictor rather than generic noise is a useful design pattern, and the framework could be relevant to researchers working on graph inverse problems, topology inference, and structured generative modeling. The synthetic correlated-edge example also identifies a real limitation of purely marginal edge prediction.

The current concerns are primarily about the interpretation and validation of the method rather than the underlying research direction. With a clearer task formulation, a theoretically sound distortion–perception experiment, and controls that isolate the value of flow matching, the work could provide useful findings for part of the TMLR audience.

**Broader Impact Concerns:**

The paper does not raise a major immediate ethical concern. However, graph reconstruction methods can potentially infer hidden or intentionally undisclosed relationships in social, communication, or organizational networks. A brief broader-impact discussion should acknowledge privacy and consent risks when the method is applied to human relational data, particularly when “missing” edges represent private rather than merely unobserved information. This does not affect my technical recommendation.

**Claims And Evidence:**

No

**Claims Explanation:**

The main concern is that the paper interprets the number of Euler steps K as controlling the distortion–perception trade-off, although K is formally a numerical integration parameter. Algorithm 1 integrates from t=0 to t=1 for every value of K, with step size 1/K. For a fixed learned velocity field, increasing K should provide a more accurate approximation to the same terminal flow map; it should not, by itself, select a different point on the theoretical distortion–perception curve. Consequently, the observation that K=1 gives higher AUC while larger K gives lower MMD may reflect discretization bias or implicit regularization from a coarse solver, rather than a principled perception–distortion interpolation. This issue affects a central theoretical interpretation of Sections 5.4 and E.6. A theoretically aligned experiment would fix a sufficiently accurate solver and vary the terminal integration time, or introduce an explicit interpolation parameter between the posterior-mean estimator and the terminal flow sample.

The experimental task definition also needs clarification. The inductive experiments appear to mask arbitrary upper-triangular adjacency entries, because the prior is trained using both observed positive and observed negative pairs, while evaluation is performed over masked upper-triangular entries. If this interpretation is correct, the setting reveals a substantial set of confirmed non-edges and is more accurately described as missing-at-random adjacency-matrix completion. This differs from standard link prediction, where a subset of positive edges is removed and the remaining non-edges are generally candidate unknowns rather than explicitly observed negatives. The distinction matters for both the available information and comparison with link-prediction literature. The blind expansion task is closer to the latter setting, but it is evaluated much less extensively.

The binary reconstruction results are also substantially weaker than the ranking metrics suggest. In the expansion experiment in Table 4, PIFM has an FNR of 100% on ENZYMES and 94.75% on PROTEINS at the stated threshold, meaning that it recovers essentially none of the hidden positive edges on these datasets. Similar behavior occurs for the MSE variants in Table 2, where FNR values on sparse graphs often lie between approximately 78% and 98%. The CE formulation reduces FNR, but sometimes produces FPR values as high as 45%−89%. Thus, the paper demonstrates improved pairwise ranking in several cases, but does not yet demonstrate a reliable operating point that produces useful reconstructed graphs. This is especially important because the stated task is graph reconstruction rather than only candidate-edge ranking.

Finally, the flow-specific contribution is not adequately isolated. Most headline results use K=1. The flow module is additionally trained on clean graphs from the dataset, while some standalone priors are fitted only to each partially observed graph. Therefore, improvements over those priors may arise from additional dataset-level supervision, capacity, or a learned graph denoiser, rather than specifically from continuous flow matching or learned multi-edge transport. A parameter- and information-matched direct refinement baseline is necessary before attributing the gains to flow matching.

**Requested Changes:**

1. The paper should not present the Euler step count K as a theoretically grounded distortion–perception control without further justification. Every value of K integrates to the same terminal time t=1, so K primarily controls discretization error.
2. The manuscript should state whether the 10% and 50% masks are sampled over all unordered node pairs or only over true edges. The numbers of observed positive entries, observed negative entries, masked positives, and masked negatives should be reported for every dataset.
3. AUC alone is insufficient when the default binary reconstruction predicts almost no positive edges. The authors should select thresholds exclusively on validation data and report precision, recall, F1, balanced accuracy or MCC, predicted edge-count error, normalized Hamming error, and graph-level exact or near-exact recovery.
4. The authors should train a baseline that receives exactly the same prior-informed matrix, observed graph information, masks, clean training graphs, and GNN backbone as PIFM, but directly predicts the corrected adjacency matrix without time conditioning or ODE integration.
5. The equivariance theorem is useful, but it mainly follows from composing an equivariant prior, equivariant velocity network, and entrywise operations. The main text should explicitly note that node2vec does not strictly satisfy the assumption, as acknowledged only later in the appendix. The novelty and practical implications of the theorem should also be separated from standard architectural equivariance.

---

> ### Author Response · Authors · 2026-07-15
> **Author Response — Reviewer 5M3h**
>
> We thank the reviewer and address each point below. Revisions are marked in purple in the paper.
>
> ### R1-1 — K as a discretization parameter
>
> We agree: $K$ is a numerical discretization parameter, not a theoretical control for the continuous distortion–perception trade-off. We removed that framing and now describe its intermediate effects as purely empirical. This step-count behavior is itself established in prior work: PMRF (Ohayon et al., 2025) and InDI (Delbracio and Milanfar) show that fewer integration steps approach the posterior mean while more steps behave like posterior sampling, trading distortion for perceptual quality; we cite them, as our $K$ observations align with theirs.
>
> We do maintain that the **endpoint $K=1$** is theoretically grounded: for the MSE rectified flow with population-optimal velocity $v^\star=\mathbb{E}[A_1-A_0\mid A_t]$, one Euler step from $t=0$ returns $\mathbb{E}[A_1\mid A_0]$: the posterior mean given the prior-informed source $A_0$, not the raw observation. Empirically (Tables 2 and 22), AUC is highest at $K=1$ in all 30 sweep cells, while larger $K$ improves distributional fidelity primarily for the MSE flow on sparse graphs (NSPDK improves in 10 of 15 cells, concentrated on ENZYMES/PROTEINS); intermediate $K$ forms no clean curve and metrics barely move past $K=50$. So $K$ has two principled endpoints but is not a solver-independent continuous control.
>
> ### R1-2 — Mask composition and protocol
>
> For ENZYMES, PROTEINS, and IMDB-B, the 10%/50% masks are sampled over **all unordered node pairs** (upper triangle), each dropped independently of edge status. So the setting is missing-at-random adjacency reconstruction, not positives-only link prediction. The manuscript adopts this wording (Appendix D.3; captions of Tables 2 and 17), with observed/masked counts (totaled over train/val/test) in Table 6. Cora instead uses a standard edge split (2,639/792/1,847 train/val/test positives, negatives 1:1).
>
> ### R1-3 — Binary metrics beyond AUC
>
> We add an operating-point evaluation: per seed we select the threshold $\tau^\star_{\mathrm{val}}$ maximizing validation MCC and apply it unchanged to test (Tables 12–14, LPFormer+MSE, five seeds). At $\tau^\star_{\mathrm{val}}$, endpoint-anchored PIFM (Model 1, $K=1$) improves recall, F1, and MCC over its LPFormer prior on all three datasets (e.g., IMDB-B MCC $0.784\to0.870$, exact recovery $20.8\to38.0\%$), while exact recovery stays 0% on ENZYMES, so strong ranking does not imply perfect graph recovery. Lowering the threshold trades false negatives for false positives, so Hamming and edge-count error can worsen even as recall/F1/MCC improve; we report these disagreements rather than treating any one thresholded metric as definitive. Figure 9 adds the threshold sweep as a sensitivity diagnostic.
>
> ### R1-4 — Matched Direct refiner
>
> We added a Direct GNN refiner matched to PIFM (input, mask, backbone, parameter count) but with time fixed at $t=0$, one residual correction, and no ODE integration (Table 21, Appendix E.7). Across 39 settings, Direct's AUC matches or exceeds the submitted flow at $K=1$ in 38/39. Diagnosing this exposed a $t=0$ train/inference mismatch (uniform $t\sim U(0,1)$ rarely samples the $K=1$ endpoint); we add two remedies (Appendix D.4): **Model 1 / Refinement A** adds a masked residual-MSE anchor at $t=0$ ($\lambda=5$) to the standard flow loss, directly supervising the velocity used by the first Euler update; **Model 2 / Refinement B** assigns 25% of training times to $t=0$. We adopt Model 1 as the main MSE model; the PIFM (LPFormer, MSE) row of Table 2 is this model at $K=1$ (73.16/74.12/94.99 AUC), matching Direct on all three. Moving it from $K=1$ to $K=100$ exposes the trade-off: on ENZYMES/PROTEINS NSPDK MMD² falls $3.1$–$3.5\times$ while AUC falls $0.10$–$0.12$. Direct is thus a strong distortion point, but the flow gives a tunable perception–distortion family.
>
> ### R1-5 — Equivariance theorem vs. architecture
>
> We agree. The theorem is a closure guarantee for the full reconstruction pipeline (masking, Euler updates, entrywise ops, restoring observed entries) given an equivariant prior and velocity field, not a new equivariant architecture. We clarify that Node2vec does not strictly satisfy the assumption, since its embeddings depend on random walks, initialization, and negative sampling. A quantitative equivariance-error bound for stochastic or non-equivariant priors is future work.
>
> ### R1-6 — Broader impact
>
> We agree that missing relational information is not benignly unobserved: a reconstructed edge may reveal a sensitive relationship. The manuscript adds an Impact Statement: PIFM should be used only when authorized, inferred edges are not established facts, and safeguards apply (data minimization, consent/purpose limitation, access controls, human review). Public-benchmark experiments do not establish suitability for surveillance or deanonymization.